# Learning Single-Index Models
# with Shallow Neural Networks

**Alberto Bietti**
New York University
Meta AI

**Joan Bruna**
New York University

**Clayton Sanford**
Columbia University

**Min Jae Song**
New York University

## Abstract

Single-index models are a class of functions given by an unknown univariate "link" function applied to an unknown one-dimensional projection of the input. These models are particularly relevant in high dimension, when the data might present low-dimensional structure that learning algorithms should adapt to. While several statistical aspects of this model, such as the sample complexity of recovering the relevant (one-dimensional) subspace, are well-understood, they rely on tailored algorithms that exploit the specific structure of the target function. In this work, we introduce a natural class of shallow neural networks and study its ability to learn single-index models via *gradient flow*. More precisely, we consider shallow networks in which biases of the neurons are frozen at random initialization. We show that the corresponding optimization landscape is benign, which in turn leads to generalization guarantees that match the near-optimal sample complexity of dedicated semi-parametric methods.

## 1 Introduction

High-dimensional learning with both computational and statistical guarantees, which is particularly relevant given the current scaling trends, remains an outstanding challenge. One important question which has received considerable attention is on understanding the advantages of using non-linear learning models, such as neural networks, over more mature (from a theoretical standpoint) counterparts, such as kernel methods [56, 57, 84, 22]. Perhaps surprisingly, the question remains largely open even for shallow neural networks.

While approximation benefits of shallow neural networks over non-adaptive kernels have been known for decades [9, 69], another important piece of the theoretical puzzle was provided by [5], whose analysis hinted at an inherent *statistical* advantage of neural networks for extracting information from high-dimensional data with a "hidden" low-dimensional structure. Providing *computational* guarantees, the remaining piece of this puzzle, is still mostly unresolved.

Several computational hardness results for learning functions that can be efficiently approximated by shallow neural networks have been established in the literature [30, 41, 28, 78, 20], ruling out positive results in the general setting. On the other hand, progress has been made on the positive side [1, 2, 73] by focusing on function classes with strong structural properties, thereby showcasing the adaptive representation learning capabilities of neural networks.

This work aligns with the latter effort, and focuses on the class of *single-index models*. Single-index models are high-dimensional functions $F : \mathbb{R}^d \to \mathbb{R}$ of the form $F_*(x) = f_*(\langle \theta^*, x \rangle)$, where *both* the univariate "link" function $f_* : \mathbb{R} \to \mathbb{R}$ and relevant (one-dimensional) subspace $\theta^* \in \mathbb{S}^{d-1}$ are unknown. These models have been extensively studied in the statistics literature [45, 48, 26, 46, 33], leading to dedicated algorithmic procedures, and can be provably approximated with shallow neural networks without incurring in a curse-of-dimensionality [5]. In contrast, the analysis

36th Conference on Neural Information Processing Systems (NeurIPS 2022).

of neural network learning using gradient-based methods has focused on the so-called "Teacher-Student" setup [43, 39, 85, 86, 10], where the link function $f_*$ is assumed to be known and used as the activation function for the student.

The mix of a high-dimensional parametric component (the hidden direction) with a non-parametric one in low dimension (the link function) in single-index models naturally suggests a shallow neural network architecture where the inner weights are shared and "active", while the biases are "lazy" [23]. We instantiate such an architecture by *freezing* the biases at random initialization, and analyze gradient descent on the free parameters in the continuous-time limit.

Our main results establish that as soon as the width $N$ of the network is larger than a quantity which depends solely on smoothness properties of the (univariate) link function $f_*$, gradient flow recovers the unknown direction $\theta^*$ with near optimal sample complexity $O(d^s)$, where $s$ is the so-called *information-exponent* of the link function [10] (at least when $s \geq 3$, see Theorem 6.1 for the formal result), and approximates the univariate link function $f_*$ near-optimally (see Corollary 6.4). The information exponent roughly captures the signal strength, which here refers to the alignment between the network direction $\theta$ and the hidden direction $\theta^*$, at typical initializations.

The success of gradient flow relies on the benign optimization landscape of the empirical loss, though the presence of degenerate saddles necessitates a careful analysis leveraging uniform convergence of the empirical landscape [60, 35]. We show that gradient flow over our proposed neural network architecture solves two distinct problems—univariate non-parametric kernel ridge regression and non-convex optimization in high dimension—*simultaneously* and *efficiently*, cementing its role as a versatile algorithm for high-dimensional learning. We illustrate our theoretical results with experiments in Section A.

## 2 Related Work

**Single-index and multi-index models.** A useful modeling assumption in high-dimensional regression is that the regression function $F(x) = \mathbb{E}[y|x]$ only depends on one or a few directions. This leads to the single-index model $F(x) = f_*(\langle \theta^*, x \rangle)$, and multi-index model $F(x) = f_*(\langle \theta_1^*, x \rangle, \ldots, \langle \theta_k^*, x \rangle)$, with $k$ typically much smaller than the dimension. Such models have a long history in the statistics literature and different methods exist for various estimation problems, including projection pursuit [36, 47], slicing [52], gradient-based estimators [53], and moment-based estimators [26]. When the function $f_*$ is also to be estimated, we face a semi-parametric problem involving parameter recovery of $\theta^*$ and non-parametric estimation of $f_*$. Our work is closely related to [33], which also characterizes the population landscape of certain objectives by leveraging Gaussian data. Multi-index models are also studied in [5] in the context of shallow neural networks, where it is shown that certain models of infinite-width shallow networks can adapt to such low-dimensional structure, though no tractable algorithms are introduced.

The works [8, 19, 66] show that certain neural networks trained close to initialization can learn certain sparse polynomials which take the form of multi-index models, but such networks do not directly aim to learn target directions. Recently, [1] studied the learnability of functions on the hypercube by shallow neural networks with stochastic gradient descent and introduces the *merged staircase property*, which provides necessary and sufficient conditions for learnability with linear sample complexity $n = O(d)$. While they learn a broader class of functions (including multi-index model) for a more efficient sample complexity regime ($O(d)$ vs $O(d^s)$), their setup is restricted to simple discrete data distributions, while our work captures the regime of semi-parametric estimation by considering Gaussian data without the sparsity requirements on $F$ implied by their merged staircase property.

Concurrently to our work, [4] and [27] studied the learnability of certain single and multi-index models on Gaussian data with shallow networks, by performing a single gradient step on the first layer before fitting the second layer. While the single step is sufficient to provide a separation from kernel methods in these works, we show that optimizing both layers jointly until convergence (for a more simplistic architecture) can significantly improve the rates, by fully decoupling the non-parametric learning part from the high-dimensional inference of the hidden direction. Finally, recently [64] studied the ability of shallow neural networks to learn certain single and multi-index models, showing in particular that SGD-trained ReLU networks can learn single-index functions with monotonic index function (corresponding in our setting to $s = 1$) with linear (up to logarithmic factors) sam-

ple complexity. Our results therefore extend such positive guarantees to a broader class of index functions with arbitrary information exponent.

**Teacher–student models.** In the context of neural networks, several works have considered the teacher–student setting [34], where the target function $F$ takes the form of a neural network with the same activation as the network used for learning [43, 39, 85, 77, 86, 10, 82]. In this case the problem does not involve non-parametric estimation as in our setup, but this line of work often involves studying optimization landscapes similar to ours for estimating hidden directions. In particular, the population landscape appearing in [10] is similar to ours, based on Hermite coefficients of link functions. The follow-up work [11] extends this to multiple student neuron directions, but still focuses on parametric rather than non-parametric statistical problems.

**Kernels and random features.** In order to obtain non-parametric estimation guarantees for learning the target function $f_*$ of the single-index model, our work builds on the kernel methods literature for approximation and non-parametric regression [76, 13, 7], their links with neural networks [24, 5], and in particular on random feature approximation [70, 6, 72, 61].

**Non-convex and non-smooth optimization landscapes.** There is a vast literature studying tractable non-convex optimization landscapes, arising from high-dimensional statistics and statistical physics [59, 58, 37, 12, 79, 17, 39, 71, 55]. A particular aspect of our setup is that the optimization landscape does *not* have the strict saddle property, which is often leveraged to establish global convergence [50]. [60, 35] study concentration properties of the empirical landscape to the population one for non-convex problems including generalized linear models. Our results rely on similar concentration analyses, but depart from these previous work by also allowing optimization of the link function, and by supporting the non-smoothness arising from ReLU activations. On the algorithmic side, we consider gradient flows on non-convex and non-smooth landscapes, which require careful technical treatment, but have been studied by previous works [32, 29, 49]. We refer the interested reader to Appendix F for more details on this technical issue.

## 3 Preliminaries

We focus on regression problems under a single-index model with Gaussian input data. Specifically, we assume $d$-dimensional inputs $x \sim \gamma_d := \mathcal{N}(0, I_d)$, and labels

$$y = F^*(x) + \xi = f_*(\langle \theta^*, x \rangle) + \xi \,,$$

where $\theta^* \in \mathbb{S}^{d-1}$ and $\xi \sim \mathcal{N}(0, \sigma^2)$ is an independent, additive Gaussian noise. The normalization $\theta^* \in \mathbb{S}^{d-1}$ is to ensure that both $f_*$ and $\theta^*$ are identifiable.

**Shallow networks and random features.** We consider learning algorithms based on shallow neural networks of the form

$$G(x; c, \theta) = c^\top \Phi(\langle \theta, x \rangle) = \frac{1}{\sqrt{N}} \sum_{i=1}^{N} c_i \phi(\varepsilon_i \langle \theta, x \rangle - b_i) \,,$$

with $\theta \in \mathbb{S}^{d-1}$, where $\phi(u) = \max\{0, u\}$ is the ReLU activation, $b_i \sim \mathcal{N}(0, \tau^2)$ (we assume $\tau > 1$) are random "bias" scalars that are frozen throughout training, and $\varepsilon_i$ are random signs with Rademacher distribution (i.e., uniform over $\{\pm 1\}$), independent from $b_i$, and also frozen during training. The resulting vector of random features is thus $\Phi(u) = \frac{1}{\sqrt{N}}(\phi(\varepsilon_i u - b_i))_{i \in [N]}$.

The choice of ReLU activation is motivated by its popularity among practitioners. As we shall see, the fact that $\phi$ is non-smooth introduces some technical challenges, but its piece-wise linear structure enables dedicated arguments both in terms of approximation as well as in the study of the optimization landscape. In Appendix G we discuss how our main results are affected when replacing the ReLU by a smooth activation, especially when choosing it such that $\phi'$ is Lipschitz.

**Empirical risk minimization.** The supervised learning task is to estimate $F^*$ (and therefore both $f_*$ and $\theta^*$) from samples $\{(x_i, y_i)\}_{i=1 \ldots n}$. We will focus on mean-squared error with Tychonov regularisation, determined by the following losses.

**Definition 3.1** (Population risk). *We define the $\ell^2$-regularized population loss by*

$$L(c,\theta) = \mathop{\mathbb{E}}_{x,y}[(y - G(x;c,\theta))^2] + \lambda\|c\|^2 = \mathop{\mathbb{E}}_{x\sim\gamma_d}[(F^*(x) - G(x;c,\theta))^2] + \sigma^2 + \lambda\|c\|^2 .$$

**Definition 3.2** (Empirical risk). *We define the $\ell^2$-regularized empirical loss by*

$$L_n(c,\theta) = \frac{1}{n}\sum_{i=1}^{n}\left(c^\top\Phi(\langle\theta,x_i\rangle) - y_i\right)^2 + \lambda\|c\|^2 . \tag{1}$$

**Hermite decomposition.** Given that our data is normally distributed, we consider the family of (normalized) Hermite polynomials $\{h_j\}_{j\in\mathbb{N}}$, which form an orthonormal basis of $L^2(\gamma)$, the space of squared-integrable function under the Gaussian measure $\gamma := \mathcal{N}(0,1)$. We will denote by $f_* = \sum_j \alpha_j h_j$ the Hermite decomposition of the target link function, which we assume is in $L^2(\gamma)$ henceforth.

We apply the following useful properties of Hermite polynomials [67, Chapter 11.2]:

$$h_j' = \sqrt{j}h_{j-1} \quad\text{and}\quad \langle h_j(\langle\theta,\cdot\rangle), h_{j'}(\langle\theta',\cdot\rangle)\rangle_{\gamma_d} = \delta_{j,j'}\langle\theta,\theta'\rangle^j,$$

where $\langle\cdot,\cdot\rangle_{\gamma_d}$ is the inner product in $L^2(\gamma_d)$ and $\delta$ the Kronecker delta. We will assume throughout that $\|f_*\|_\gamma^2 = \sum_j \alpha_j^2$, $\|f_*'\|_\gamma^2 = \sum_j j\alpha_j^2$, and $\|f_*''\|_\gamma^2 = \sum_j j(j-1)\alpha_j^2$ are all finite (see Assumption 5.2). We will also consider the weighted Sobolev space $H^2(\gamma)$, which contains functions $f = \sum_j \alpha_j h_j \in L^2(\gamma)$ such that $\sum_j j^2|\alpha_j|^2 < \infty$.

**Random features to Hermite coefficients.** To precisely characterize the landscape of $L(c,\theta)$, we introduce notation to represent each random feature function in terms of Hermite polynomials $h_j$. We define the linear integral operator $\mathcal{T} : L^2(\gamma) \to \mathbb{R}^N$ by

$$(\mathcal{T}f)_i := \langle f, \phi_{b_i}^{\epsilon_i}\rangle_\gamma := \frac{1}{\sqrt{N}}\mathop{\mathbb{E}}_{z\sim\gamma}[f(z)\phi(\varepsilon_i z - b_i)] , \ i \in [N] . \tag{2}$$

Note that $\mathcal{T}$ has rank $N$ almost surely.

The operator $\mathcal{T}$ has an adjoint $\mathcal{T}^* : \mathbb{R}^N \to L^2(\gamma)$ defined as $(\mathcal{T}^*c)(u) = \frac{1}{\sqrt{N}}\sum_{i=1}^N c_i\phi(\varepsilon_i u - b_i) = c^\top\Phi(u)$. Finally, for any $j \in \mathbb{N}$, let $\mathcal{T}_j \in \mathbb{R}^N$ defined as $\mathcal{T}_j = \mathcal{T}h_j$. We can then write down the Hermite expansion of the student network:

$$G(x;c,\theta) = c^\top\Phi(\langle\theta,x\rangle) = \sum_{j\geq 0}\langle c, \mathcal{T}_j\rangle h_j(\langle\theta,x\rangle).$$

Denoting $m = \langle\theta,\theta^*\rangle$, the regularized population objective can be expressed as

$$L(c,\theta) = \sum_j \alpha_j^2 + \sum_j\langle c, \mathcal{T}_j\rangle^2 - 2\sum_j \alpha_j\langle c, \mathcal{T}_j\rangle m^j + \lambda\|c\|^2, \tag{3}$$

where the term $\sum_j \alpha_j^2$ is a constant that can be ignored. Let $Q := \mathcal{T}\mathcal{T}^* \in \mathbb{R}^{N\times N}$ be a feature covariance matrix and $Q_\lambda = Q + \lambda I$. Note that $Q_\lambda$ is positive definite for $\lambda > 0$. We define the regularized projection $\hat{P}_\lambda = \hat{\Sigma}(\hat{\Sigma} + \lambda I)^{-1}$ onto the random feature space for $\hat{\Sigma} = \mathcal{T}^*\mathcal{T}$. Observe that $(\hat{P}_\lambda f)(u) = c^{*\top}\Phi(u)$, where $c^* \in \mathbb{R}^N$ is the solution to the following objective:

$$\min_{c\in\mathbb{R}^N}\left\|f - c^\top\Phi\right\|_\gamma^2 + \lambda\|c\|_2^2 . \tag{4}$$

**Geometry on the sphere.** Because the direction $\theta$ is constrained to lie on the sphere, our optimization algorithms rely on spherical (Riemannian) gradients, which are defined as follows:

$$\nabla_\theta^{\mathbb{S}^{d-1}}L(c,\theta) = \Pi_{\theta^\perp}\nabla_\theta L(c,\theta) ,$$

where $\Pi_{\theta^\perp}v = v - \langle\theta,v\rangle\theta$. We say that $(c,\theta)$ is a *critical point* of $L$ if $\nabla_\theta^{\mathbb{S}^{d-1}}L(c,\theta) = 0$ and $\nabla_c L(c,\theta) = 0$.

# 4 Univariate Approximation using Random Features

Before addressing the high-dimensionality of the learning problem, we first focus on the non-parametric approximation aspects of the univariate link function. As usual, we start by deriving approximation rates of the infinitely-wide model, given by a RKHS, and then establish approximation rates for our random feature model.

**Univariate RKHS.** If we fix the direction $\theta$, learning $c$ alone may be seen as a random feature model [70] that approximates a kernel method with the following kernel.

$$\kappa(u,v) = \mathbb{E}_{b \sim \gamma_\tau, \varepsilon \sim \text{Rad}}[\phi(\varepsilon u - b)\phi(\varepsilon v - b)] , \ u, v \in \mathbb{R} , \tag{5}$$

where $\gamma_\tau = \mathcal{N}(0, \tau^2)$ is the Gaussian measure on $\mathbb{R}$ with mean zero and variance $\tau^2$, and $\varepsilon$ is a random sign with a Rademacher distribution. The kernel $\kappa$ corresponds to an RKHS $\mathcal{H}$, given by

$$\mathcal{H} := \left\{ f : \mathbb{R} \to \mathbb{R} \mid f(u) = \frac{1}{\sqrt{2}} \int [c_+(b)\phi(u-b) + c_-(b)\phi(-u-b)] \, d\gamma_\tau(b), \ c_+, c_- \in L^2(\gamma_\tau) \right\} .$$

The following lemma characterizes the corresponding RKHS norm $\| \cdot \|_{\mathcal{H}}$, and follows from Theorem B.8, by noting that $\kappa(u,v) = \langle \psi(u), \psi(v) \rangle_{L^2(\gamma_\tau)^2}$, with $\psi(u) = \frac{1}{\sqrt{2}}(\phi(u - \cdot), \phi(-u - \cdot))$.

**Lemma 4.1** (RKHS norm). *The RKHS norm in $\mathcal{H}$ is given by*

$$\|f\|_{\mathcal{H}}^2 = \inf \left\{ \|c_+\|_{\gamma_\tau}^2 + \|c_-\|_{\gamma_\tau}^2; \ f(u) = \frac{1}{\sqrt{2}} \int [c_+(b)\phi(u-b) + c_-(b)\phi(-u-b)] \, d\gamma_\tau(b) \right\} . \tag{6}$$

The choice of ReLU for the activation function gives us more explicit control over the RKHS norm, based on Sobolev representations, as already exploited by several works [68, 5, 75].

**Lemma 4.2** (RKHS norm bound). *Let $f \in H^2(\gamma) \cap C^2(\mathbb{R})$ and $\tau > 1$. If $f$ and $f'$ both have polynomial growth and $\int \frac{|f''(t)|^2}{\gamma_\tau(t)} dt < \infty$, then $f \in \mathcal{H}$ with*

$$\|f\|_{\mathcal{H}}^2 \leq 6\tau \left( \int \frac{|f''(t)|^2}{\gamma_\tau(t)} dt + \|f\|_\gamma^2 + 6\|f'\|_\gamma^2 + 2\langle f, f'' \rangle_\gamma \right) . \tag{7}$$

The proof is in Appendix C.1.

**RKHS approximation properties.** Let $A(f, \lambda)$ be the (regularized) $L^2$ approximation error for functions in the space $\mathcal{H}$ with respect to the target function $f$ and measure $\gamma$. Formally,

$$A(f, \lambda) := \min_{g \in \mathcal{H}} \|f - g\|_\gamma^2 + \lambda \|g\|_{\mathcal{H}}^2 .$$

We will now show that the approximation error of the RKHS corresponding to an infinite number of random features can be bounded in terms of the regularization $\lambda$ and the $\gamma$-norm of the second derivative of the target function. For that purpose, we consider the following 'source' condition to ensure a polynomial approximation error in $\lambda$.

**Assumption 4.3** (Containment in $L^4(\gamma)$). *Let $\mathcal{F} = \{f \in H^2(\gamma) \mid f'' \in L^4(\gamma)\}$. We assume $f \in \mathcal{F}$ and define $K := \inf \{B \geq 1 \mid \|f''\|_{L^4(\gamma)} \leq B\|f''\|_{L^2(\gamma)}\}$.*

Assumption 4.3 provides a sufficient condition for approximating $f$ with functions in the RKHS. The family of approximants $\{h_M \in \mathcal{H} \mid M > 0\}$ we use in Lemma 4.4 are exactly equal to $f$ on $[-M, M]$ and are linear outside of $[-M, M]$. The $L^4$ assumption on $f''$ ensures control over the RKHS norm of $h_M$. Note that by Jensen's inequality, $L^4(\gamma) \subset L^2(\gamma)$, so $K$ is always well-defined for $f'' \in L^4(\gamma)$. Sigmoidal functions, compactly supported smooth functions, and, more generally, functions with polynomial growth satisfy Assumption 4.3.

**Lemma 4.4** (RKHS approximation error). *Let $\lambda \in (0, 1)$ and $f \in \mathcal{F}$. Then, there exists a universal constant $C > 0$ such that*

$$A(f, \lambda) \leq C\left( \tau^{1+\beta} \|f''\|_4^2 \cdot \lambda^\beta + \lambda C_f^2 \right) , \tag{8}$$

*where $\beta = \frac{1 - 1/\tau^2}{3 + 1/\tau^2}$ and $C_f = \max\{\|f\|_\gamma, \|f'\|_\gamma, \|f''\|_\gamma\}$.*

The proof appears in Appendix C.2. This lemma allows us to control the RKHS approximation error of a target function in terms of their Hermite decompositions. The main technical difficulty is that the RKHS integral operator $\Sigma$ does not diagonalise in the Hermite basis; we address this with a dedicated argument exploiting the RKHS Sobolev representation of Lemma 4.2. The assumption that $f'' \in L^4(\gamma)$ (Assumption 4.3) is sufficient for our purposes but not necessary for polynomial-in-$\lambda$ approximation rates. In Section H, we show that the ReLU function $\phi(t) = \max(0, t)$, which is Lipschitz but not in $H^2(\gamma)$, as the *target* satisfies $A(\phi, \lambda) \lesssim \tau^2 \lambda^{2/3}$ using a direct argument. Extending the class of functions approximable by $\mathcal{H}$ with polynomial-in-$\lambda$ rate is an interesting future direction.

**Random feature approximation.** We now consider (finite) random feature approximations to functions in the RKHS. Lemma C.2 shows that the best possible loss of a linear combination of sufficiently many finite features is bounded above by the best approximation with infinitely many features with high probability. More specifically, as long as $N \gtrsim \lambda^{-1}$, the random feature approximation error behaves like the RKHS approximation error. The proof leverages the 'degrees of freedom' of the kernel and closely tracks [6].

## 5 Population Landscape under Frozen Random Biases

To understand optimization and generalization properties of gradient flow on the empirical loss $L_n(c, \theta)$, we first the study optimization landscape of the population loss $L(c, \theta)$. We characterize the critical points of the population loss and show that gradient flow on a shallow neural network of sufficient width $N$ converges only when its direction vector $\theta$ is either parallel or orthogonal to the target direction $\theta^*$. Importantly, the sufficient number of random features $N$ depends on the $\ell^2$-regularization parameter $\lambda \in (0, 1)$, but *not* on the input dimension $d$. In Section 6, we further show that sufficiently large $n$, the number of training samples, guarantees similar properties for the *empirical* landscape and thus has favorable generalization properties for most initializations.

One of the main measures of complexity for the target link function $f_*$ is its information exponent (see e.g., [10]), defined as follows.

**Definition 5.1** (Information exponent). *Let* $f : \mathbb{R} \to \mathbb{R}$ *be any function such that* $f \in L^2(\gamma)$. *The* information exponent *of* $f$, *which we denote by* $s$, *is the index of the first non-zero Hermite coefficient. That is,* $s := \min\{j \in \mathbb{N} : \alpha_j \neq 0\}$.

We make the following regularity assumptions on the target link function $f_*$ to ensure small approximation error by random features, and benign population and empirical landscape.

**Assumption 5.2** (Regularity of $f_*$). *We consider* $f_* \in L^2(\gamma)$, *with* $f_* = \sum_j \alpha_j h_j$. *Assume 1)* $f_*$ *is Lipschitz, 2)* $\sum_j j^4 |\alpha_j|^2 < \infty$, *and 3)* $f_*''(z) := \sum_j \sqrt{(j+2)(j+1)}\alpha_{j+2} h_j(z)$ *is in* $L^4(\gamma)$ *(Assumption 4.3)*

We also suppose w.l.o.g. that $f_*$ is normalized so that $\|f_*\|_\gamma = 1$. To analyze the critical points of $L(c, \theta)$, we introduce the *projected population loss* $\bar{L}(\theta)$, which can be seen as a semiparametric least squares (SLS) objective [48].

$$\bar{L}(\theta) := \min_c L(c, \theta) .$$

**Theorem 5.3** (Critical points of the population loss). *Assume* $f_*$ *satisfies Assumption 5.2 and has information exponent* $s \geq 1$. *For* $\tau > 1$, *and* $\delta \in (0, 1)$, *there exists* $\lambda^* \leq 1$ *depending only on* $\tau$ *and the target link function* $f_*$ *and a universal constant* $C > 0$ *such that if*

$$\lambda < \lambda^* \qquad and \qquad N \geq \frac{C}{\lambda} \log\left(\frac{1}{\lambda\delta}\right) \tag{9}$$

*then with probability* $1 - \delta$ *over the random biases* $b_j$ *and signs* $\varepsilon_j$, $j = 1 \ldots N$, *the set of first-order critical points* $\Omega := \{(c, \theta) : \nabla_\theta^{\mathbb{S}^{d-1}} L(c, \theta) = 0, \ \nabla_c L(c, \theta) = 0\}$ *satisfies:*

1. *(orientation relative to* $\theta^*$*) if* $(c, \theta) \in \Omega$, *then either* $\theta \in \{\pm\theta^*\}$ *or* $\langle\theta, \theta^*\rangle = 0$.

2. *(existence and uniqueness of c) if* $\nabla_\theta^{\mathbb{S}^{d-1}} \bar{L}(\theta) = 0$, *then there exists a unique* $c \in \mathbb{R}^N$ *such that* $(c, \theta) \in \Omega$.

Theorem 5.3 thus establishes a benign optimization landscape in the population limit, rejoining several known non-convex objectives with similar behavior, such as tensor decomposition [40] or matrix completion [38]. Importantly, this optimization landscape has the same topology as the one that arises from using the Hermite basis, the tailored choice for data generated by a single-index model in Gaussian space [33, Theorem 5], instead of random scalar features. We view this as an interesting robustness property of shallow neural networks, at least in the regime where biases are randomly frozen.

## 6 Empirical Landscape and Generalization Guarantees

Section 5 shows that the population landscape has a relatively simple structure given $N = \Omega_d(1)$ random features. We now study the optimization properties of its finite-sample counterpart.

We consider the estimator $\hat{F}(x) := \hat{f}(\langle x, \hat{\theta}\rangle)$, where $(\hat{f}, \hat{\theta})$ are obtained by running gradient flow on $c$ and $\theta$ to minimize $L_n(c, \theta)$. Such strategy appears to be reasonable in light of the properties of the population landscape, since its local minimizers are also global and correspond to $\tilde{f} = \hat{P}_\lambda f^*$ and $\tilde{\theta} = \theta^*$ (Theorem 5.3). For a sufficiently large sample size $n$, one expects the empirical landscape $L_n$ to concentrate around its expectation $L$ and inherit its benign optimization properties. However, the presence of a degenerate saddle at $(c, m) = (0, 0)$ for $m = \langle \theta, \theta^* \rangle$ flattens the landscape around the "equator" $\{\theta : \langle \theta, \theta^* \rangle = 0\}$. Thus, more samples are required to ensure that gradient flow escapes from the equatorial region despite its dangerously close random initialization $|\langle \theta_0, \theta^* \rangle| = \Theta(1/\sqrt{d})$.

Prior works have obtained sample complexity of $n = O(d^s)$, where we recall that $s$ is the information exponent of the target function $f_*$, for recovering $\theta^*$ either by employing a learning algorithm that explicitly learns individual Hermite polynomials [33] or by assuming that $f_*$ is known a priori [10].[1] The intuition behind this sample complexity is roughly as follows.

- The empirical optimization landscape (when regarded as a function only of the direction $\theta$) near the equator ($|m| \ll 1$) is of the form $L(\theta) \asymp m^s$.

- In order to certify that the optimization algorithm does *not* converge to a suboptimal critical point (i.e., $\|\nabla L(\theta)\| \le \epsilon$) on the equator, one requires that $m \ge \epsilon^{1/(s-1)}$.

- A uniform gradient convergence bound of the form $\|\nabla L(\theta) - \nabla L_n(\theta)\| = O(\sqrt{d/n})$ and the fact that $m = \Theta(1/\sqrt{d})$ at initialization together imply that $n = O(d^s)$ samples are sufficient to escape from the "influence" of the equator.

In order to repurpose these arguments to our setting, the relative scaling of the top-layer weights $c$ relative to the direction vector $\theta$ is crucial, as has also been observed in the literature on *lazy*-vs-*rich* regimes [23, 84] in the context of overparametrized neural networks.

We consider an idealized version of Gradient Descent over the empirical loss $L_n$ in the infinitesimally small learning rate regime. This results in a gradient flow ODE of the form:

$$\dot{c}(t) = -\zeta(t)\nabla_c L_n(c, \theta)$$
$$\dot{\theta}(t) = -\nabla_\theta^{\mathbb{S}^{d-1}} L_n(c, \theta) \, , \tag{10}$$

where $\zeta$ is the relative scale between $c$ and $\theta$, and $\nabla_\theta^{\mathbb{S}^{d-1}}$ is the Riemannian gradient.

Specifically, we study a setting where $\zeta(t) = \mathbf{1}(t > T_0)$ for an appropriately chosen time $T_0$. This choice produces a two-stage gradient-flow. During the first phase, up until time $T_0$, we only optimize the first-layer parameter $\theta$ from a random initialization. In the second phase, the parameters $c$ and $\theta$ are jointly optimized. Additionally, the first phase only utilizes a small fraction $\frac{N_0}{N} \ll 1$ of the random features employed in the second phase. Our procedure implements this by randomly initializing $c(0) \in \mathbb{R}^N$ as a sparse vector with $N_0$ non-zero components. The overall approach is described in Procedure 1.

---

[1]Actually, in [10] the authors obtain a slightly improved sample complexity of $\widetilde{O}(d^{s-1})$ for $s \ge 3$ by directly analyzing SGD with fixed step-size, as well as a matching lower bound (for SGD in the small step-size regime) up to polylogarithmic factors.

Our main result, proved in Appendix E, establishes that this gradient flow efficiently finds an approximate minimizer of the *population* loss, with an error (explicitly quantified as a function of $n$) that reveals the fundamental role of the information exponent $s$ of $f_*$. On top of the regularity conditions on the target link function of Assumption 5.2, the upper bound on $\lambda$, and the lower bound on $N$ from Eq. (9) of Theorem 5.3, the main result imposes a (compatible) upper bound on $N$ and an appropriate choice of initial norm ($\rho$) and sparsity ($N_0/N$) for $c(0)$. In this section, we are interested in behavior as $n$, $d$, and $N$ grow asymptotically and hence treat the target function $f_*$ and terms derived from it (including Hermite coefficients $\alpha_j$ and information exponent $s$), along with the bias parameter $\tau$ and regularity parameter $\beta$, as constants and omit them from asymptotic notation.

**Theorem 6.1** (Gradient flow finds approximate minimizers). *For $\delta \in (0, 1/4)$ and $f_*$ satisfying Assumption 5.2, suppose the following are true: (i) $\lambda = O(1)$ and $\lambda = \Omega(\sqrt{\Delta_{crit}})$, where $\Delta_{crit} :=$* $\max\{\sqrt{\frac{d+N}{n}}, (\frac{d^2}{n})^{2s/(2s-1)}\}$, *(ii) $n = \widetilde{\Omega}(\max\{\frac{(d+N)d^{s-1}}{\lambda^4}, \frac{d^{(s+3)/2}}{\lambda^2}\})$, *(iii)* $N = \Omega(\frac{1}{\lambda}\log\frac{1}{\lambda\delta})$ & $N = \widetilde{O}(\lambda\Delta_{crit}^{-1})$, *(iv)* $N_0 = \Theta(\log(\frac{1}{\delta}))$, *(v)* $\rho = \Theta(\sqrt{N}N_0^{-(2+s)/2}(\tau^2 + \lambda N/N_0)^{-1})$, *(vi)* $T_0 = \widetilde{\Theta}(d^{s/2-1})$, *and (vii)* $T_1 = \widetilde{\Theta}(\frac{\lambda^4 n}{d+N})$. *Then, if we run Procedure 1 for $T = T_0 + T_1$ time steps with the above parameters, with probability at least $\frac{1}{2} - \delta$ we have*

$$1 - |\langle\theta_T, \theta^*\rangle| = \widetilde{O}\left(\lambda^{-4}\max\left\{\frac{d+N}{n}, \frac{d^4}{n^2}\right\}\right) . \tag{11}$$

The empirical gradient flow therefore escapes the influence of the degenerate saddle with sample complexity $n = \tilde{\Theta}(d^s)$ when $\lambda = \Theta(1)$ and $s > 2$. This is an instance of gradient flow successfully optimizing a non-convex objective *without the strict saddle property* as soon as $s > 2$, building from the simpler optimization landscapes of [33, 10]. This is in contrast, for example, with spiked tensor recovery problems [12] where the signal strength is substantially weaker, leading to complexity in the optimization landscape. This sample complexity nearly matches the tight lower bound $n \gg d^{s-1}$ of [10], obtained in the case $s > 2$ and applies to SGD rather than batch gradient descent, as is our case. For $s \in \{1, 2\}$, the sample complexity becomes $d^2$ and $d^{2.5}$, respectively, but we note that these may be improved to $d^s$ when using a smooth activation (see Appendix G). For $s = 2$, this is comparable to [27], which requires $\Omega(d^2)$ samples, but still above the $n \gg d\log d$ of [10].

We emphasize that the "near-optimality" of our sample complexity $n = \tilde{\Theta}(d^s)$ only pertains to gradient-based methods in small learning rate regimes [10]. In fact, alternative methods have been shown to achieve a better sample complexity of $\widetilde{O}(d^{\lceil s/2 \rceil})$ in the setting where $f_*$ is a certain degree-$s$ polynomial with information exponent $s$ [19], leveraging tensor factorization tools. We leave it as an interesting open question to further understand the nature of this gap.

We note that the dependence on $d + N$ in the recovery guarantee can likely be improved to $d$ using a more refined norm-based landscape concentration analysis. We also remark that if we chose the number of random features $N = \Theta(\frac{1}{\lambda}\log\frac{1}{\lambda\delta})$, then the requirement $\lambda = \Omega(\sqrt{\Delta_{crit}})$ for large enough sample size $n$ imposes $\lambda \gg n^{-1/5}$. This lower bound on $\lambda$ guarantees that critical points near initialization can be escaped, but may slow down learning. Nevertheless, this is sufficient to obtain an excess risk that vanishes with $n$, with a rate independent of $d$, as we now show.

**Corollary 6.2** (Excess risk of Algorithm 1). *Under the assumptions of Theorem 6.1, and further assuming $n \gtrsim d^3$, an appropriate choice of $\lambda$ yields an excess risk guarantee of the form*

$$\|\hat{F} - F^*\|_{\gamma_d}^2 = \widetilde{O}\left(\left(\frac{d}{n}\right)^{\frac{\beta}{\beta+4}} + \left(\frac{1}{n}\right)^{\frac{\beta}{\beta+5}}\right), \tag{12}$$

*where $\beta$ is defined as in Lemma 4.4.*

This result indicates that the joint training is consistent, with excess risk that vanishes with a rate with explicit dependence on the ambient dimension $d$ and the non-parametric exponent $\beta$. However, it requires a regularisation strength $\lambda = \Theta\left(\max\left\{\left(\frac{1}{n}\right)^{\frac{1}{\beta+5}}, \left(\frac{d}{n}\right)^{\frac{1}{\beta+4}}\right\}\right)$ to ensure enough gradient concentration, which happens to be larger than the optimal regularisation of the univariate kernel ridge regression associated with learning $f_*$. We are thus 'over-regularising' as a consequence of the joint training, resulting in a slower rate than what would be dictated by estimating $\theta_*$ and $f_*$ separately. A simple mechanism to break this inefficiency is by considering a fine-tuning step of the second-layer terms.

---

**Procedure 1** Gradient Flow

---

**Require:** $N_0, \rho, T_0, T_1, N$, and $\lambda$.
    Initialize $\theta(0) \sim \mathrm{Unif}(\mathbb{S}^{d-1})$, $c(0) \sim \mathrm{Unif}(\{c \in \mathbb{R}^N; \|c\|_2 = \rho; \|c\|_0 = N_0\})$.
    Run Gradient Flow (10) with $\zeta(t) = \mathbf{1}(t > T_0)$ up to time $T = T_0 + T_1$.
    Set $\hat{\theta} = \theta(T)$, $\hat{c} = c(T)$.

---

**Procedure 2** Fine-Tuning

---

**Require:** $\hat{\theta}$ from Procedure 1, and $\lambda_{n'}$.
    Set $\hat{c} = \arg\min_c L'_{n'}(c, \hat{\theta})$ as in (13).

---

**Fine-tuning the second layer.** After running Algorithm 1, we may include a final fine-tuning phase of training for second layer weights $c$ alone, using a separate training sample $(x'_i, y'_i)$, $i = 1, \dots, n'$ and a possibly different regularization parameter $\lambda_{n'}$. More precisely, we set

$$\hat{c} = \arg\min_c \left\{ L'_{n'}(c, \hat{\theta}) := \frac{1}{n'} \sum_{i=1}^{n'} (c^\top \Phi(\langle \hat{\theta}, x'_i \rangle) - y'_i)^2 + \lambda_{n'} \|c\|^2 \right\}, \tag{13}$$

where $\hat{\theta}$ denotes the output of the previous gradient descent phase. Note that this is a strongly convex optimization problem, and can thus be optimized efficiently using gradient methods or by solving a linear system. While this may not be needed in practice, we use a different training sample for technical reasons, namely to break the dependence between the data and the kernel, which depends on the initial training sample through $\hat{\theta}$. We note that such sample splitting strategies are commonly used in other contexts in the statistics literature (e.g., [15, 21]). We obtain the following guarantee.

**Proposition 6.3** (Excess risk of fine-tuning). *Let* $\delta \in (0, 1/4)$. *Let* $m = \langle \theta^*, \hat{\theta} \rangle$, *where* $\hat{\theta}$ *is obtained from the previous gradient descent phase, and let* $\hat{c}$ *be the ridge regression estimator obtained from a fresh dataset* $\mathcal{D}'$ *of* $n'$ *samples,* $N$ *random features, and regularization parameter* $\lambda_{n'} := (\sigma^2 \tau^2 / \|f''_*\|_\gamma^2 n')^{1/(\beta+1)}$, *and let* $\hat{F}(x) = \hat{c}^\top \Phi(\langle \hat{\theta}, x \rangle)$. *Assume*

$$n' \gtrsim \max \left\{ \sigma^2 \tau^2 / \|f''_*\|_\gamma^2, (\|f''_*\|_\gamma^2 / \sigma^2 \tau^2)^{1/\beta}, \|f_*\|_\infty^2 / (\sigma^2 \tau^2)^{\beta/(\beta+1)} \right\}, \text{ and}$$

$$N \gtrsim C_\tau \left( n' \|f''_*\|_\gamma^2 / \sigma^2 \tau^2 \right)^{\frac{1}{\beta+1}} \log \left( n'^{1/(\beta+1)} \delta^{-1} \right).$$

*Then with probability at least* $1 - \delta$ *over the random features, we have*

$$\mathop{\mathbb{E}}_{\mathcal{D}'} [\|\hat{F} - F^*\|_{\gamma_d}^2 | \hat{\theta}] \lesssim \|f''_*\|_\gamma^{\frac{2}{\beta+1}} \left( \frac{\sigma^2 \tau^2}{n'} \right)^{\frac{\beta}{\beta+1}} + \|f'_*\|_\gamma^2 (1 - |m|), \tag{14}$$

*where the expectation is over the* $n'$ *fresh samples, and is conditioned on the previously obtained* $\hat{\theta}$.

Decoupling the regularization parameters of the two phases (along with number of random features $N$) allows us to keep a large $\lambda$ in the first phase, leading to fast recovery as per Theorem 6.1, while obtaining vanishing excess risk through a decreasing $\lambda_{n'}$. This is illustrated in the result on the excess risk for Algorithm 2.

**Corollary 6.4** (Excess risk of Algorithm 2). *Let* $\delta \in (0, 1/4)$. *As in Theorem 6.1, let* $\mu_s = \langle h_s, \Sigma h_s \rangle > 0$, *and let* $f_*$ *satisfy Assumption 5.2. Let* $\lambda = \Theta(1)$, *and assume the following on the sample sizes and number of random features for the first phase* $(n, N, N_0)$ *and fine-tuning phase* $(n', N')$:

$$N = N_0 = \Theta \left( \frac{1}{\lambda} \log \frac{1}{\lambda \delta} \right), \quad n = \widetilde{\Omega} \left( \max\{d^s, d^{(s+3)/2}\} \right), \quad N' = \widetilde{\Omega} \left( n'^{\frac{1}{\beta+1}} \right).$$

*and let* $\rho$ *be as in Theorem 6.1. With probability at least* $1/2 - 2\delta$ *over the initial* $n$ *samples, initialization, random features, we have*

$$\mathop{\mathbb{E}}_{\mathcal{D}'} [\|\hat{F} - F_{\theta^*}\|_{\gamma_d}^2] \leq \widetilde{O} \left( \max \left\{ \frac{d}{n}, \frac{d^4}{n^2} \right\} + \left( \frac{1}{n'} \right)^{\frac{\beta}{\beta+1}} \right), \tag{15}$$

*where the constants in* $\widetilde{O}$ *do not depend on* $d$ *other than through logarithmic factors.*

Comparing Corollaries 6.2 and 6.4, we observe that the fine-tuning stage recovers the optimal sample complexity, where the non-parametric rate is fully independent of the ambient dimension $d$, while in Corollary 6.2 there is still a dependence in the constants. We make the following additional remarks:

- The time-scale separation schedule for $\zeta$ in Theorem 6.1 is sufficient but possibly not necessary. The analysis of vanilla dynamics ($\zeta(t) \equiv 1$) is challenging, since during the initial phase of training there may be adverse interaction between $c$ and $\theta$, which under naive analysis lead to sub-optimal sample complexity of $n \geq O(d^{2s})$. Observe that this separate analysis of 'weak' and 'strong' recovery phases of learning appears in most contemporary related work [27, 1, 4, 10, 11].

- The time discretization to turn Procedure 1 into a proper algorithm should follow from standard time discretization arguments, although the case where $\phi = \text{ReLU}$ requires special care due to the non-smoothness of the loss (see Appendix F for further discussion). In such setting, such discretization arguments do not hold for vanilla gradient descent in the worst-case [51], although these may be recovered by appropriately smoothing the objective prior to computing the gradient, or by using instead a smooth activation function (see Appendix G).

## 7   Conclusion

This work studies the ability of shallow neural networks to learn single-index models with gradient descent. Our main results are positive, and demonstrate their ability to solve a semi-parametric problem with nearly optimal guarantees. Interestingly, this success story combines elements from the feature-learning regime, i.e., the ability to efficiently identify the hidden direction in high-dimensions under a non-convex objective, with ingredients from the lazy-regime. Our technical analysis leverages tools from high-dimensional probability (such as uniform gradient concentration) and RKHS approximation, and complements the growing body of theoretical work on the efficacy of gradient methods for non-convex objectives. We have followed the standard approach of first establishing benign topological properties of the population loss, and then extending them to the empirical loss. There are nonetheless several unanswered questions that our work has not addressed.

**Weaker regularity and discrete-time analysis.** Our approximation rate for ReLU as the *target* (see Appendix H) suggests that the polynomial-in-$\lambda$ approximation rate may be extended to function classes beyond $\mathcal{F} \subset H^2(\gamma)$, such as Lipschitz functions with smooth tail behavior. Thus, it would be interesting to extend our empirical landscape concentration results to such functions satisfying weaker regularity assumptions, which currently rely on certain polynomial decay of the Hermite coefficients (see Assumption 5.2). Additionally, by using a smooth activation function (see Appendix G), our GF dynamics can be discretized and turned into GD with analogous sample and time complexity. In that context, a natural goal is to compare quantitatively the differences between GD with multiple passes over the training data and SGD by adapting tools from [10, 11].

**Trainable biases and untied directions.** Our proposed neural network architecture is non-standard, in the sense that its biases are frozen at initialization and all neurons share the same inner weight. For the purposes of learning single-index models, removing these restrictions would not bring any statistical benefits. However, it would be interesting to extend our analysis to the general setting where the first layer weights are not tied and biases are not frozen.

**Extension to multi-index models.** Multi-index models are natural extensions in which the hidden direction $\theta^*$ is replaced by a hidden low-dimensional *subspace*. Typically, multi-index models enjoy similar statistical guarantees as single-index models [33, 5], and thus a natural question is whether the same algorithmic tools developed here extend to the multi-index setting.

**Gradient dynamics without warm-start.** An unsatisfactory aspect of our results is the requirement that the algorithm starts by only optimizing $\theta$ for $t < T_0$. It would be interesting to understand whether the vanilla dynamics can also succeed provably.

**Acknowledgments.**   We are thankful to Enric Boix-Adserà, Alex Damian, Cédric Gerbelot, Daniel Hsu, Jason Lee, Theodor Misiakiewicz, Matus Telgarsky, Eric Vanden-Eijnden, and Denny Wu for useful discussions. We also thank the anonymous NeurIPS reviewers and area chair for helpful feedback. JB, AB and MJ are partially supported by NSF RI-1816753, NSF CAREER CIF 1845360, NSF CHS-1901091, NSF Scale MoDL DMS 2134216, Capital One and Samsung Electronics. CS is supported by an NSF GRFP and by NSF grants CCF-1814873 and IIS-1838154.

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
