# Appendices

# A   Numerical Experiments

In this section, we illustrate our results by running Algorithm 1 on a simple synthetic dataset. We consider a piecewise linear target link function $f_*$ that is compactly supported, illustrated in Figure 1, and data generated by the model described in Section 3, with $\sigma = 0.001$. To understand behavior when varying the information exponent $s$, we also consider teachers $f_*^s = f_* - \sum_{j<s} <f_*, h_j> h_j$ with the low-order Hermite components are removed. We initialize the direction $\theta$ randomly on the sphere, and the parameters $c_k$ i.i.d. with variance 1. We run gradient descent on the empirical loss for 10 000 iterations using a step-size that is 100 times larger on $\theta$ than on $c$, with projections of $\theta$ on the sphere after each step, and with $\ell_2$ regularization on $c$. We start optimizing $c$ only after 500 steps to simulate the warm-start phase. We show the loss obtained on 10K held-out test samples. For fine-tuning (denoted "ridge" in the plots), we re-optimize the output layer $c$ exactly on the training data using ridge regression with a possibly different regularization parameter $\lambda'$, We fix $N = 100$ random features, and optimize hyperparameters ($\lambda$, $\lambda'$, and the step-size) on the test data. All experiments were run on CPUs. Each experiment was repeated 10 times, and the figures report either the mean and standard deviation over the 10 runs, or the best performing model out of the 10 runs.

Our experiment results are shown in Figure 2. For $s \geq 3$, only some of the 10 runs were successful in recovering the target direction, and we thus show the best performing run for such curves (indeed, our theory suggests that there may be a non-negligible probability of failure). We observe that full recovery ($|m| \to 1$) requires more samples when the dimension $d$ increases, while the excess risk curves have approximately the same rate for large enough $n$, regardless of the dimension or information exponent, as predicted by our theory. The bottom plots for $d = 50$ suggest that $s = 3$ requires more samples than smaller $s$ for perfect recovery, while the remaining curves are somewhat comparable. This similarity between $s = 1$ and $s = 2$ is reminiscent of the situation in [10], where the rates for these two cases only differ by a logarithmic factor, and suggests that it may be possible to improve the $O(d^s)$ rates in our results for $s \geq 2$.

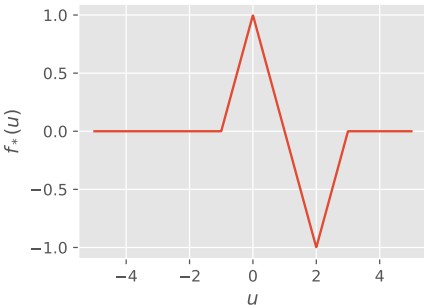

Figure 1: Piecewise linear teacher link function $f_*$.

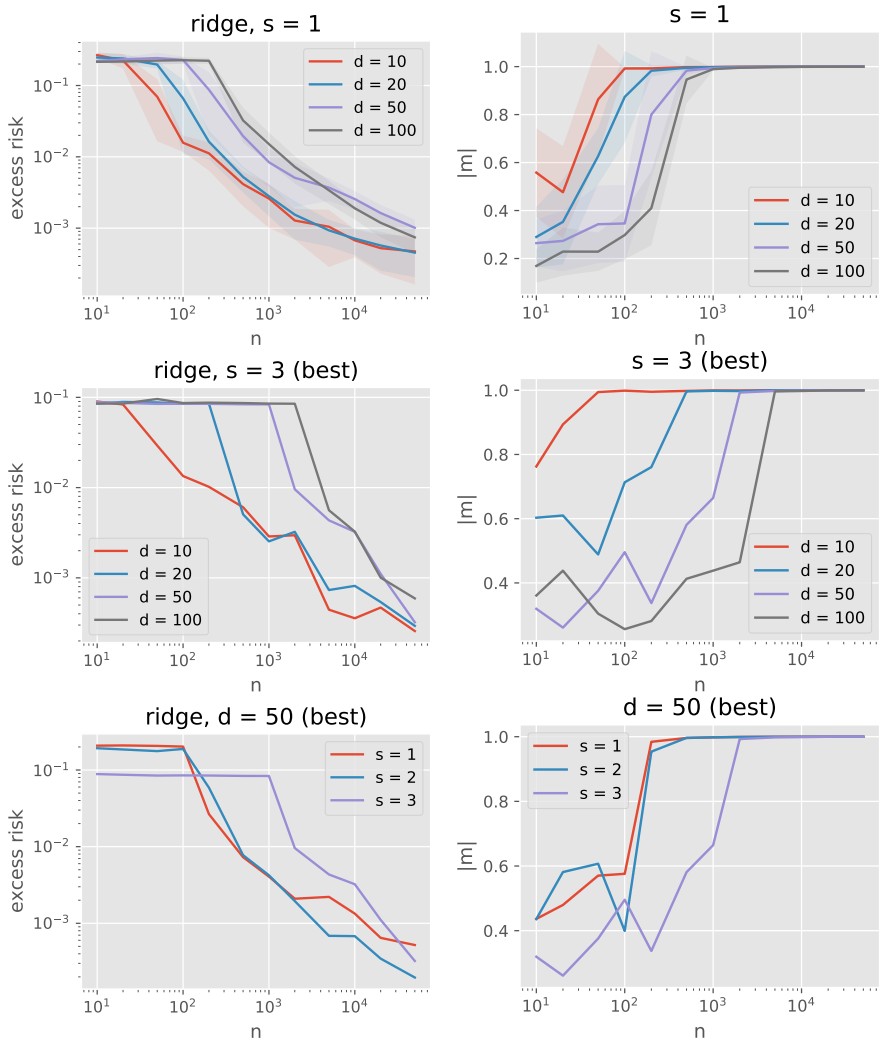

Figure 2: Excess risk $\|\hat{F} - F^*\|_{\gamma_d}^2$ with final ridge/fine-tuning step (left), and correlation $|m|$ (right) as a function of sample size $n$.

## B  Additional Preliminaries and Concentration Bounds

We introduce several well-known concentration bounds that we apply throughout the appendix. Borrowing notation from [83], we first introduce notation of sub-gaussian and sub-exponential random variables, vectors, and matrices.

**Definition B.1.** *A real-valued random variable $z$ is $\gamma^2$-sub-gaussian $\|z\|_{\psi_2} := \inf\{t > 0 : \mathbb{E}[\exp(z^2/t^2)] \le 2\} \le \gamma$. Likewise, a random vector $x \in \mathbb{R}^d$ is $\gamma^2$-sub-gaussian and we denote $\|x\|_{\psi_2} \le \gamma$ if $\|w \cdot x\|_{\psi_2} \le \gamma$ for any fixed $w \in \mathbb{S}^{d-1}$.*

**Definition B.2.** *A real-valued random variable $y$ is $\gamma$-sub-exponential $\|y\|_{\psi_1} := \inf\{t > 0 : \mathbb{E}[\exp(|y|/t)] \le 2\} \le \gamma$. Likewise, a random vector $u \in \mathbb{R}^d$ is $\gamma$-sub-exponential and we denote $\|u\|_{\psi_1} \le \gamma$ if $\|w \cdot u\|_{\psi_1} \le \gamma$ for any fixed $w \in \mathbb{S}^{d-1}$.*

We note several key properties of sub-gaussian and sub-exponential random variables that we repeatedly rely on.

**Fact B.3.** *Let $z_1, \ldots, z_N$ and $y_1, \ldots, y_N$ be sub-gaussian and sub-exponential random variables respectively. Then the following hold for some universal constant $C$.*

1. *Centering preserves sub-gaussianity and sub-exponentiality:* $\|z_1 - \mathbb{E}[z_1]\|_{\psi_2} \leq C \|z_1\|_{\psi_2}$ *and* $\|y_1 - \mathbb{E}[y_1]\|_{\psi_1} \leq C \|y_1\|_{\psi_2}$ *[83, Lemma 2.6.8 and Exercise 2.7.10].*

2. *Products of sub-gaussian random variables are sub-exponential:* $z_1 z_2$ *is sub-exponential and* $\|z_1 z_2\|_{\psi_1} \leq \|z_1\|_{\psi_2} \|z_2\|_{\psi_2}$ *[83, Lemma 2.7.7].*

3. *Sums of independent sub-gaussian random variables are sub-gaussian. If* $z_1, \ldots, z_N$ *are independent, then,*
$$\left\| \sum_{i=1}^N z_i \right\|_{\psi_2}^2 \leq C \sum_{i=1}^n \|z_i\|_{\psi_2}^2$$
*[83, Proposition 2.6.1].*

4. *Sums of pairs of sub-exponential random variables are sub-exponential.* $\|y_1 + y_2\|_{\psi_1} \leq C(\|y_1\|_{\psi_1} + \|y_2\|_{\psi_1})$ *[60, Lemma 2].*

5. *Lipschitz functions preserve sub-gaussianity. For Lipschitz-continuous* $f : \mathbb{R} \to \mathbb{R}$, $\|f(z_1) - \mathbb{E}[f(z_1)]\|_{\psi_2} \leq C \mathrm{Lip}(f) \|z_1\|_{\psi_2}$.

6. *If* $\beta$ *is a bounded random variable with* $|\beta| \leq s$, *then* $\|\beta(z_1 - \mathbb{E}[z_1])\|_{\psi_2} \leq Cs \|z_1\|_{\psi_2}$ *[60, Lemma 1].*

**Theorem B.4** (Bernstein's inequality, [83, Corollary 2.8.3])**.** *For independent, mean-zero, sub-exponential random variables* $x_1, \ldots, x_n$ *and any* $t \geq 0$,
$$\mathbb{P}\left[ \left| \frac{1}{n} \sum_{i=1}^n x_i \right| \geq t \right] \leq 2 \exp\left( -Cn \min\left( \frac{t^2}{K^2}, \frac{t}{K} \right) \right)$$
*for universal* $C$ *and* $K = \max_i \|x_i\|_{\psi_1}$.

We also include several basic facts about $\epsilon$-covers, which are useful in several proofs.

**Definition B.5.** *For some compact* $S \subset \mathbb{R}^d$, $\mathcal{N}_\epsilon \subseteq S$ *is an* $\epsilon$-*covering if for all* $x \in S$, *there exists* $\tilde{x} \in \mathcal{N}_\epsilon$ *such that* $\|x - \tilde{x}\| \leq \epsilon$.

**Fact B.6** ([83, Corollary 4.2.13, Lemma 4.4.1])**.** *For all* $\epsilon \in (0, 1]$, *there exists an* $\epsilon$-*net* $\mathcal{N}_\epsilon$ *for* $\mathbb{S}^{d-1}$ *with* $|\mathcal{N}_\epsilon| \leq (\frac{3}{\epsilon})^d$. *Moreover, for any* $a \in \mathbb{R}^d$,
$$\|a\| \leq \frac{1}{1-\epsilon} \max_{x \in \mathcal{N}_\epsilon} \langle a, x \rangle.$$

**Lemma B.7** (Anticoncentration on the unit sphere)**.** *Let* $d \in \mathbb{N}$. *Let* $\theta \in \mathbb{S}^{d-1}$ *be any fixed unit vector and let* $u$ *be a random vector drawn uniformly from* $\mathbb{S}^{d-1}$. *Then, for any* $\epsilon > 0$,
$$\mathbb{P}[\langle \theta, u \rangle \leq \epsilon] \leq 4\sqrt{d}\epsilon .$$

*Proof.* We show this using an elementary argument. Let $\epsilon \in [0, 1]$. Define
$$G(\epsilon) := \int_0^\epsilon (1 - t^2)^{(d-1)/2} dt \leq \epsilon .$$

By Gautschi's inequality [31, Eq. 5.6.4] for the Gamma function, we have
$$G(1) = \frac{\sqrt{\pi}}{2} \cdot \frac{\Gamma((d+1)/2)}{\Gamma((d+2)/2)} \geq \frac{\sqrt{\pi}}{4} \cdot \frac{1}{\sqrt{d}} .$$

Thus we have the following anti-concentration bound on $\mathbb{S}^{d-1}$.
$$\mathbb{P}[|\langle \theta, u \rangle| \leq \epsilon] = G(\epsilon)/G(1) \leq \frac{4}{\sqrt{\pi}} \cdot \sqrt{d}\epsilon .$$

$\square$

Finally, we recall the following result on reproducing kernel Hilbert spaces, which describes the RKHS for kernels defined from explicit features maps.

**Theorem B.8** ([74, §2.1]). *Let $\psi : \mathcal{X} \to \mathcal{F}$ be a mapping into a Hilbert space $\mathcal{F}$, and for $x, x' \in \mathcal{X}$, define the kernel $\kappa(x, x') = \langle \psi(x), \psi(x') \rangle_{\mathcal{F}}$. The RKHS $\mathcal{H}$ of $\kappa$ consists of functions of the form $f(x) = \langle g, \psi(x) \rangle_{\mathcal{F}}$, and for any $f \in \mathcal{H}$, the RKHS norm of $f$ is defined by*

$$\|f\|_{\mathcal{H}}^2 := \inf \left\{ \|g\|_{\mathcal{F}}^2 \ : \ g \in \mathcal{F} \text{ s.t. } f = \langle g, \psi(\cdot) \rangle_{\mathcal{F}} \right\} \ .$$

# C  Proofs of Section 4

## C.1  Proof of Lemma 4.2

**Claim C.1** (RKHS Sobolev representation). *Let $f \in H^2(\gamma) \cap C^2(\mathbb{R})$ be a function such that $\lim_{t \to -\infty} f(t) = \lim_{t \to -\infty} f'(t) = 0$ and $\int_{-\infty}^{\infty} (f''(u)^2/\gamma_\tau(u)) du < \infty$. Then,*

$$f(t) = \int_{-\infty}^{\infty} \frac{f''(u)}{\gamma_\tau(u)} \phi(t - u) d\gamma_\tau(u) \ . \tag{16}$$

*Moreover, the RKHS norm of $f$ is upper bounded as follows.*

$$\|f\|_{\mathcal{H}}^2 \le \int_{-\infty}^{\infty} \frac{f''(u)^2}{\gamma_\tau(u)} du \ .$$

*Proof of Claim C.1.* By the Fundamental Theorem of Calculus and Fubini's Theorem,

$$f(t) = \int_{-\infty}^{t} f'(w) dw = \int_{-\infty}^{\infty} f'(w) \mathbf{1}[w \le t] dw = \int_{-\infty}^{\infty} \left( \int_{-\infty}^{\infty} f''(u) \mathbf{1}[u \le w] du \right) \mathbf{1}[w \le t] dw$$

$$= \int_{-\infty}^{\infty} f''(u) \left( \int_{-\infty}^{\infty} \mathbf{1}[u \le w] \mathbf{1}[w \le t] dw \right) du$$

$$= \int_{-\infty}^{\infty} f''(u) \phi(t - u) du$$

$$= \int_{-\infty}^{\infty} \frac{f''(u)}{\gamma_\tau(u)} \phi(t - u) d\gamma_\tau(u) \ .$$

The upper bound on the RKHS norm follows from the above representation and Lemma 4.1. $\square$

**Lemma 4.2** (RKHS norm bound). *Let $f \in H^2(\gamma) \cap C^2(\mathbb{R})$ and $\tau > 1$. If $f$ and $f'$ both have polynomial growth and $\int \frac{|f''(t)|^2}{\gamma_\tau(t)} dt < \infty$, then $f \in \mathcal{H}$ with*

$$\|f\|_{\mathcal{H}}^2 \le 6\tau \left( \int \frac{|f''(t)|^2}{\gamma_\tau(t)} dt + \|f\|_\gamma^2 + 6\|f'\|_\gamma^2 + 2\langle f, f'' \rangle_\gamma \right) \ . \tag{7}$$

*Proof of Lemma 4.2.* For general $f$, the boundary conditions of Claim C.1, i.e., $\lim_{t \to -\infty} f(t) = \lim_{t \to -\infty} f'(t) = 0$, do not hold. However, we can reduce to the case considered in Claim C.1 by decomposing $f$ into 2 parts, i.e., $f = f_1 + f_2$, where $f_1(t)$ and $f_2(-t)$ individually satisfy the assumptions of Claim C.1.

Let $\varphi(t) = \int_{-\infty}^{t} \gamma(u) du$. We decompose

$$f = f \cdot \varphi + f \cdot (1 - \varphi) := f_1 + f_2 \ .$$

For any $t \in \mathbb{R}$, $0 \le \varphi(t) \le 1$, and since we assume $f$ has polynomial growth, $\lim_{t \to -\infty} f_1(t) = \lim_{t \to -\infty} f_1'(t) = 0$ and $\lim_{t \to \infty} f_2(t) = \lim_{t \to \infty} f_2'(t) = 0$. By Lemma 4.1, it holds that

$$\|f\|_{\mathcal{H}}^2 \le \int \left( \left( \frac{f_1''(t)}{\gamma_\tau(t)} \right)^2 + \left( \frac{f_2''(t)}{\gamma_\tau(t)} \right)^2 \right) d\gamma_\tau(t) = \int \frac{f_1''(t)^2 + f_2''(t)^2}{\gamma_\tau(t)} dt \ . \tag{17}$$

To upper bound the RHS of Eq. (17) in terms of $f$ and its derivatives, we derive explicit expressions for $f_1''$ and $f_2''$. Since $\varphi'(t) = \gamma(t)$ and $\varphi''(t) = -t\gamma(t)$, we have

$$f_1''(t) = f''(t) \cdot \varphi(t) + 2f'(t) \cdot \gamma(t) - f \cdot (t\gamma(t))$$
$$f_2''(t) = f''(t) \cdot (1 - \varphi(t)) - 2f'(t) \cdot \gamma(t) + f \cdot (t\gamma(t)) .$$

Using the elementary inequality $(a_1 + a_2 + a_3)^2 \le 3(a_1^2 + a_2^2 + a_3^2)$ (from Cauchy-Schwarz) and the fact that $0 \le \varphi(t) \le 1$ and $0 \le \gamma(t) \le 1$ for all $t \in \mathbb{R}$,

$$\int \frac{f_1''(t)^2}{\gamma_\tau(t)} dt \le \int \frac{3}{\gamma_\tau(t)} \left( f''(t)^2 + 4f'(t)^2 \gamma(t)^2 + f(t)^2 (t\gamma(t))^2 \right) dt$$

$$\le 3 \left( \int \frac{f''(t)^2}{\gamma_\tau(t)} dt + 4\tau \int f'(t)^2 d\gamma(t) + \tau \int f(t)^2 t^2 d\gamma(t) \right)$$

$$= 3 \left( \int \frac{f''(t)^2}{\gamma_\tau(t)} dt + 4\tau \|f'\|_\gamma^2 + \tau \|f \cdot t\|_\gamma^2 \right) .$$

The same upper bound holds for $f_2$. Thus, from Eq. (17) and the fact that $\tau > 1$, we have

$$\|f\|_{\mathcal{H}}^2 \le 6\tau \left( \int \frac{|f''(t)|^2}{\gamma_\tau(t)} dt + 4\|f'\|_\gamma^2 + \|f \cdot t\|_\gamma^2 \right) .$$

It remains to upper bound $\|f \cdot t\|_\gamma$ purely in terms of $f$ and its higher-order derivatives. Using integration by parts, we have that for any differentiable $F : \mathbb{R} \to \mathbb{R}$ with polynomial growth,

$$\int tF(t)\gamma(t) dt = -F(u)\gamma(u)\big|_{u=-\infty}^{\infty} + \int F'(t)\gamma(t) dt = \int F'(t)\gamma(t) dt . \tag{18}$$

Hence, by the assumption that $f, f'$ have polynomial growth and applying the identity Eq. (18) twice, we have

$$\|f \cdot t\|_\gamma^2 = \int t \cdot (tf(t)^2)\gamma(t) dt = \int (tf(t)^2)'\gamma(t) dt = \|f\|_\gamma^2 + 2\int t \cdot (f(t)f'(t))\gamma(t) dt$$

$$= \|f\|_\gamma^2 + 2\int (f'(t)^2 + f(t)f''(t))\gamma(t) dt$$

$$= \|f\|_\gamma^2 + 2\langle f, f''\rangle_\gamma + 2\|f'\|_\gamma^2 ,$$

In conclusion, we have

$$\|f\|_{\mathcal{H}}^2 \le 6\tau \left( \int \frac{|f''(t)|^2}{\gamma_\tau(t)} dt + \|f\|_\gamma^2 + 6\|f'\|_\gamma^2 + 2\langle f, f''\rangle_\gamma \right) . \tag{19}$$

$\square$

## C.2 Proof of Lemma 4.4

**Lemma 4.4** (RKHS approximation error). *Let $\lambda \in (0,1)$ and $f \in \mathcal{F}$. Then, there exists a universal constant $C > 0$ such that*

$$A(f, \lambda) \le C \left( \tau^{1+\beta} \|f''\|_4^2 \cdot \lambda^\beta + \lambda C_f^2 \right) , \tag{8}$$

*where $\beta = \frac{1 - 1/\tau^2}{3 + 1/\tau^2}$ and $C_f = \max\{\|f\|_\gamma, \|f'\|_\gamma, \|f''\|_\gamma\}$.*

*Proof.* We first prove the result for $f \in C^2(\mathbb{R})$ and then extend it to general $f \in H^2(\gamma)$ s.t. $f'' \in L^4(\gamma)$ using a density argument. Define the $\lambda$-regularized approximation error of $h \in \mathcal{H}$ by

$$\mathcal{E}(h) = \|f - h\|_\gamma^2 + \lambda\|h\|_{\mathcal{H}}^2 ,$$

and recall that $A(f, \lambda) = \min_{h \in \mathcal{H}} \mathcal{E}(h)$.

We approximate $f$ with a single-parameter family of functions $\{h_M \mid M > 0\}$, defined by $h_M''(t) = f''(t) \cdot \mathbb{1}[|t| \le M]$. By the Fundamental Theorem of Calculus, we have

$$
h_M(t) = \begin{cases} f(t) & \text{if } |t| \le M \\ f(M) + f'(M)t & \text{if } t > M \\ f(-M) + f'(-M)t & \text{if } t < -M \ . \end{cases}
$$

Thus, $h_M$ matches $f$ exactly on $[-M, M]$ and is linear with slope $f'(M)$ (resp. $f'(-M)$) for $t \ge M$ (resp. $t \le -M$). We first show that $h_M \in \mathcal{H}$, which implies $A(f, \lambda) \le \inf_{M>0} \mathcal{E}(h_M)$, and then show that for an explicit choice of $M$, $\mathcal{E}(h_M)$ has the desired upper bound.

For any finite $M > 0$, $h_M$ satisfies the assumptions of Lemma 4.2; both $f$ and $f'$ have polynomial growth (linear and zero growth, respectively) and

$$
\int_{\mathbb{R}} h_M''(u)^2/\gamma_\tau(u) = \int_{|u| \le M} f''(u)^2/\gamma_\tau(u) \le \|f''\|_\gamma^2/(\gamma_\tau(M)\gamma(M)) < \infty \ .
$$

Hence,

$$
\begin{aligned}
\|h_M\|_{\mathcal{H}}^2 &\le 6\tau \left( \int_{-\infty}^{\infty} \frac{h_M''(u)^2}{\gamma_\tau(u)} du + \|h_M\|_\gamma^2 + 6\|h_M'\|_\gamma^2 + 2\langle h_M, h_M''\rangle_\gamma \right) \\
&\le 6\tau \left( \int_{-M}^{M} \frac{f''(u)^2}{\gamma_\tau(u)\gamma(u)} d\gamma(u) + \|h_M\|_\gamma^2 + 6\|h_M'\|_\gamma^2 + 2\|f\|_\gamma\|f''\|_\gamma \right) \\
&\le 6\tau \left( \frac{\|f''\|_\gamma^2}{\gamma_\tau(M)\gamma(M)} + 2(\|f\|_\gamma^2 + \|r_M\|_\gamma^2) + 12(\|f'\|_\gamma^2 + \|r_M'\|_\gamma^2) + \|f\|_\gamma^2 + \|f''\|_\gamma^2 \right) ,
\end{aligned}
\tag{20}
$$

where $r_M = f - h_M$ and we used the triangle inequality and $2ab \le a^2 + b^2$ in Eq. (20). Note that since $\tau > 1$, the first term of Eq. (20) is upper bounded by $\lesssim \tau^2 \|f''\|_\gamma^2 e^{(\tau^2+1)M^2/(2\tau^2)}$.

We now upper bound $\|r_M\|_\gamma$ and $\|r_M'\|_\gamma$. Note that both $r_M$ and its derivative $r_M'$ are identically zero on $[-M, M]$ and that $r_M''(t) = f''(t)$ for $|t| > M$. Thus, for $t > M$ (same holds for $t > -M$),

$$
r_M'(t) = \int_M^t r_M''(u)du = \int_M^t f''(u)du \ , \text{ and } r_M(t) = \int_M^t r_M'(u)du \ .
$$

Next, we decompose $\|r_M'\|_\gamma^2$ into two terms, the positive part $\|r_M'\|_{\gamma,+}^2$ and the negative part $\|r_M'\|_{\gamma,-}^2$. That is,

$$
\|r_M'\|_\gamma^2 = \int_{-\infty}^{\infty} r_M'(u)^2 du = \int_0^{\infty} r_M'(u)^2 du + \int_{-\infty}^0 r_M'(u)^2 du := \|r_M'\|_{\gamma,+}^2 + \|r_M'\|_{\gamma,-}^2 \ .
$$

We show an upper bound for $\|r_M'\|_{\gamma,+}^2$ with the understanding that the same upper bound applies to $\|r_M'\|_{\gamma,-}^2$. By repeated applications of Fubini's Theorem and an upper bound on the complementary

error function $\mathrm{erfc}(t) = \int_t^\infty e^{-u^2/2}du \le 2e^{-t^2/2}$ [18, Theorem 1],

$$
\begin{aligned}
\|r_M'\|_{\gamma,+}^2 &= \int r_M'(t)^2 \gamma(t)dt \\
&= \int \left( \int_M^\infty f''(u)\mathbb{1}[0 \le u \le t]du \int_M^\infty f''(w)\mathbb{1}[0 \le w \le t]dw \right)\gamma(t)dt \\
&\le \int_M^\infty \int_M^\infty |f''(u)||f''(w)| \left( \int \mathbb{1}[0 \le u \le t]\mathbb{1}[0 \le w \le t]\gamma(t)dt \right)dudw \\
&\le \int_M^\infty \int_M^\infty |f''(u)||f''(w)|\gamma(\max\{u,w\})dudw \qquad\qquad \text{[18, Theorem 1]} \\
&\le \sqrt{8\pi}\left( \int_M^\infty |f''(u)|\gamma_{\sqrt{2}}(u)du \right)^2 \\
&\lesssim \left( \int_M^\infty \gamma_2(u)^{4/3}du \right)^{3/2} \cdot \left( \int (f''(u)\gamma_2(u))^4 du \right)^{1/2} \qquad\qquad \text{[Hölder's inequality]} \\
&\lesssim \gamma_{\sqrt{3}}(M)^{3/2}\cdot\|f''\|_4^2 . \qquad\qquad\qquad\qquad\qquad\qquad\qquad\qquad (21)
\end{aligned}
$$

An upper bound on $\|r_M\|_{\gamma,+}^2$ follows from similar calculations.

$$
\begin{aligned}
\|r_M\|_{\gamma,+}^2 &= \int_{\mathbb{R}_+} r_M^2(t)\gamma(t)dt \\
&= \int_M^\infty \int_M^\infty r_M'(u)r_M'(w)\left( \int_{\mathbb{R}_+} \mathbb{1}[0 \le u \le t]\cdot\mathbb{1}[0 \le w \le t]\gamma(t)dt \right)dudw \\
&\le 2\int_M^\infty \int_M^\infty r_M'(u)r_M'(w)\gamma(\max\{u,w\})dudw \\
&\le 2\int_M^\infty \int_M^\infty \left( \int_M^\infty |f''(\tilde{u})|\,\mathbb{1}[0 \le \tilde{u} \le u]d\tilde{u} \right)\left( \int_M^\infty |f''(\tilde{w})|\,\mathbb{1}[0 \le \tilde{w} \le w]d\tilde{w} \right)\cdot\gamma(\max\{u,w\})dudw \\
&\le 2\int_M^\infty \int_M^\infty |f''(\tilde{u})||f''(\tilde{w})|\left( \int\int \mathbb{1}[0 \le \tilde{u} \le u]\,\mathbb{1}[0 \le \tilde{w} \le w]\gamma(\max\{u,w\})dudw \right)d\tilde{u}d\tilde{w} \\
&\lesssim \int_M^\infty \int_M^\infty |f''(\tilde{u})||f''(\tilde{w})|\left( \int \mathbb{1}[0 \le \tilde{u} \le u]\gamma_{\sqrt{2}}(u)du \right)\left( \int \mathbb{1}[0 \le \tilde{w} \le w]\gamma_{\sqrt{2}}(w))dw \right)d\tilde{u}d\tilde{w} \\
&\lesssim \left( \int_M^\infty |f''(\tilde{u})|\gamma_{\sqrt{2}}(\tilde{u})d\tilde{u} \right)^2 \\
&\lesssim \left( \int_M^\infty \gamma_2(\tilde{u})^{4/3}d\tilde{u} \right)^{3/2}\cdot\left( \int(f''(\tilde{u})\gamma_2(\tilde{u}))^4 d\tilde{u} \right)^{1/2} \\
&\lesssim \gamma_{\sqrt{3}}(M)^{3/2}\cdot\|f''\|_4^2 ,
\end{aligned}
$$

Putting everything together in Eq. (20),

$$
\begin{aligned}
\|h_M\|_{\mathcal{H}}^2 &\lesssim \tau\Big( \frac{\|f''\|_\gamma^2}{\gamma_\tau(M)\gamma(M)} + (\|f\|_\gamma^2 + \|r_M\|_\gamma^2) + (\|f'\|_\gamma^2 + \|r_M'\|_\gamma^2) + \|f\|_\gamma^2 + \|f''\|_\gamma^2 \Big) \\
&\lesssim \tau\Big( \tau\|f''\|_\gamma^2\cdot e^{\frac{1+\tau^2}{2\tau^2}\cdot M^2} + \|f\|_\gamma^2 + \|f'\|_\gamma^2 + \|f''\|_\gamma^2 + \|f''\|_4^2\cdot e^{-\frac{M^2}{4}} \Big) .
\end{aligned}
$$

Thus,

$$
\begin{aligned}
\mathcal{E}(h_M) &= \|r_M\|_\gamma^2 + \lambda\|h_M\|_{\mathcal{H}}^2 \\
&\lesssim \|f''\|_4^2\cdot e^{-\frac{M^2}{4}} + \lambda\tau\Big( \tau\|f''\|_\gamma^2\cdot e^{\frac{1+\tau^2}{2\tau^2}\cdot M^2} + \|f\|_\gamma^2 + \|f'\|_\gamma^2 + \|f''\|_\gamma^2 + \|f''\|_4^2\cdot e^{-\frac{M^2}{4}} \Big) \\
&\lesssim (1+\lambda\tau)\|f''\|_4^2\cdot e^{-\frac{M^2}{4}} + \lambda\tau\Big( \tau\|f''\|_\gamma^2\cdot e^{\frac{1+\tau^2}{2\tau^2}\cdot M^2} + \|f\|_\gamma^2 + \|f'\|_\gamma^2 + \|f''\|_\gamma^2 \Big) \\
&\lesssim \tau\|f''\|_4^2\cdot e^{-\frac{M^2}{4}} + \lambda\tau\Big( \tau\|f''\|_\gamma^2\cdot e^{\frac{1+\tau^2}{2\tau^2}\cdot M^2} + \|f\|_\gamma^2 + \|f'\|_\gamma^2 + \|f''\|_\gamma^2 \Big) , \qquad (22)
\end{aligned}
$$

where we used the fact that $\tau > \max\{1, \lambda\tau\}$ in Eq. (22).

It remains to balance the terms in Eq. (22) by choosing an appropriate value for $M > 0$. We choose $M$ by balancing $\tau\|f''\|_4^2 \cdot e^{-\frac{M^2}{4}}$ and $\lambda\tau^2\|f''\|_\gamma^2 \cdot e^{\frac{1+\tau^2}{2\tau^2}\cdot M^2}$.

$$\tau\|f''\|_4^2 \cdot e^{-\frac{M^2}{4}} = \lambda\tau^2\|f''\|_\gamma^2 \cdot e^{\frac{1+\tau^2}{2\tau^2}\cdot M^2} \Rightarrow M^2 = \frac{4\tau^2}{1+3\tau^2}\log\left(\frac{\|f''\|_4^2}{\lambda\tau\|f''\|_\gamma^2}\right).$$

Let $\beta = 1 - \frac{1+\tau^2}{2\tau^2}\cdot\frac{4\tau^2}{1+3\tau^2} = \frac{\tau^2-1}{1+3\tau^2} = \frac{1-1/\tau^2}{3+1/\tau^2}$. Plugging the above value of $M$ into Eq. (22),

$$\mathcal{E}(h_M) \lesssim \tau^{1+\beta}\|f''\|_4^2(\lambda\|f''\|_\gamma^2/\|f''\|_4^2)^\beta + \lambda C_f^2 \leq \tau^{1+\beta}\|f''\|_4^2\lambda^\beta + \lambda C_f^2,$$

where $C_f = \max\{\|f\|_\gamma, \|f'\|_\gamma, \|f''\|_\gamma\}$.

Finally, let us use a density argument to extend the result to general $f \in H^2(\gamma)$ such that $f'' \in L^4(\gamma)$. We can consider a sequence $f_\rho = U_\rho f$ with $\rho \to 1$, where $U_\rho$ is the Ornstein-Uhlenbeck semigroup given by $U_\rho f(x) = \mathbb{E}_z[f(\rho x + \sqrt{1-\rho^2}z)]$. We verify that $f_\rho \in C^2(\mathbb{R})$ for any $\rho < 1$. Moreover, from $f_\rho' = \rho U_\rho[f'], f_\rho'' = \rho^2 U_\rho[f'']$ and the fact that the semigroup is strongly continuous in $L^p(\gamma)$ for any $p \geq 1$, we obtain that $f_\rho \to f, f_\rho' \to f'$ and $f_\rho'' \to f''$ in $L^2(\gamma)$ as well as in $L^4(\gamma)$, as $\rho \to 1$. As a result,

$$\begin{aligned}
A(f,\lambda) &\leq \|f - f_\rho\|_\gamma^2 + A(f_\rho,\lambda) \\
&\leq \|f - f_\rho\|_\gamma^2 + C\tau^{1+\beta}\|f_\rho''\|_4^2\lambda^\beta + \lambda C_{f_\rho}^2 \to C\tau^{1+\beta}\|f''\|_4^2\lambda^\beta + \lambda C_f^2 \qquad \text{(as } \rho \to 1\text{)}.
\end{aligned}$$

$\square$

## C.3  Random feature approximation

**Lemma C.2** (Random features approximation, adapted from [6, Prop 1]). *Let $\delta \in (0,1)$, $\tau > 1$, $\hat{\Sigma} = \mathcal{T}^*\mathcal{T}$, and let $\hat{P}_\lambda = \hat{\Sigma}(\hat{\Sigma} + \lambda I)^{-1}$ be the regularized projection onto the random feature space $\hat{\mathcal{H}}$. There exists a universal constant $C > 0$ such that if $N \geq \frac{C}{\lambda}\log\frac{1}{\lambda\delta}$, then with probability at least $1 - \delta$, for any $f \in L^2(\gamma)$ with $\mathbb{E}[f] = 0$, the following holds.*

$$\|(I - \hat{P}_\lambda)f\|_\gamma^2 \leq 4A(f,\lambda). \tag{23}$$

*Proof.* The lemma follows from Lemmas C.3 and C.4. Note that the zero-mean assumption $\mathbb{E}[f] = 0$ is necessary to obtain a tight enough bound of these degrees of freedom. $\square$

Before stating and proving the two supporting lemmas, we introduce several terms are used to study the similarity of a finite random feature model to its infinite counterpart. Let $\hat{\kappa}(u,v)$ be the random empirical kernel associated with $N$ random features:

$$\hat{\kappa}(u,v) = \frac{1}{N}\sum_{i=1}^N \phi_{b_i}^{\varepsilon_i}(u)\phi_{b_i}^{\varepsilon_i}(v), \tag{24}$$

where $\phi_b^\varepsilon(u) := \phi(\varepsilon u - b)$, $b_i, \varepsilon_i$ are i.i.d. random variables drawn from $\gamma_\tau \otimes \text{Rad}$. Its associated integral operator in $L_2(\gamma)$ is given by $\hat{\Sigma}$.

In the following, we consider $\Sigma$ the integral operator corresponding to the kernel $\kappa$:

$$\Sigma f(u) = \int f(v)\kappa(u,v)d\gamma(v).$$

By a technical lemma adapted from [6], the approximation error of the random feature model is controlled via the regularization parameter $\lambda > 0$.

**Lemma C.3** (Random features approximation, adapted from [6]). *Let $\delta \in (0, 1)$, and let $\mathbb{P}$ be the orthogonal projection operator on $span\{h_j, j \geq 1\}$ in $L^2(\gamma)$. Define*

$$d_{\max}(\lambda) := \sup_{b \in \mathbb{R}, \varepsilon \in \{\pm 1\}} \langle \mathbb{P}\phi_b^\varepsilon, (\mathbb{P}\Sigma\mathbb{P} + \lambda I)^{-1}\mathbb{P}\phi_b^\varepsilon\rangle_\gamma. \tag{25}$$

*There exists a constant $C > 0$ such that if $N \geq Cd_{\max}(\lambda) \log(d_{\max}(\lambda)/\delta)$, we have, with probability at least $1 - \delta$, for any $f \in L^2(\gamma)$,*

$$\|(I - \hat{P}_\lambda)\mathbb{P}f\|_\gamma^2 \leq 4A(\mathbb{P}f, \lambda) . \tag{26}$$

*Proof.* Assume for now that $f \in \mathcal{H}$ and denote $\tilde{f} := \mathbb{P}f$. Note that we have

$$\begin{aligned}
\|(I - \hat{P}_\lambda)\tilde{f}\|_\gamma^2 &= \|(I - \hat{\Sigma}(\hat{\Sigma} + \lambda I)^{-1})\tilde{f}\|_\gamma^2 \\
&= \|((\hat{\Sigma} + \lambda I) - \hat{\Sigma})(\hat{\Sigma} + \lambda I)^{-1}\tilde{f}\|_\gamma^2 \\
&= \|\lambda(\hat{\Sigma} + \lambda I)^{-1}\tilde{f}\|_\gamma^2 \\
&= \lambda^2\langle \tilde{f}, (\hat{\Sigma} + \lambda I)^{-2}\tilde{f}\rangle_\gamma \\
&\leq \lambda\langle \tilde{f}, (\hat{\Sigma} + \lambda I)^{-1}\tilde{f}\rangle_\gamma = \lambda\langle f, \mathbb{P}(\hat{\Sigma} + \lambda I)^{-1}\mathbb{P}f\rangle_\gamma \\
&= \lambda\langle f, \mathbb{P}(\mathbb{P}\hat{\Sigma}\mathbb{P} + \lambda I)^{-1}\mathbb{P}f\rangle_\gamma = \lambda\langle \tilde{f}, (\mathbb{P}\hat{\Sigma}\mathbb{P} + \lambda I)^{-1}\tilde{f}\rangle_\gamma,
\end{aligned}$$

where the last line uses the fact that $\mathbb{P}$ is a projection operator. Note that $\mathbb{P}\hat{\Sigma}\mathbb{P}$ is the integral operator of the random feature kernel with features $\mathbb{P}\phi_{b_i}^{\varepsilon_i}$. We now apply [6] to this projected kernel to control random feature approximation. From the proof of [6, Prop. 1], the following holds with probability $1 - \delta$ (see [6, end of p.37]): for any $g \in L^2(\gamma)$,

$$\langle g, (\mathbb{P}\hat{\Sigma}\mathbb{P} + \lambda I)^{-1}g\rangle_\gamma \leq 4\langle g, (\mathbb{P}\Sigma\mathbb{P} + \lambda I)^{-1}g\rangle_\gamma,$$

as long as $N \geq Cd_{\max}(\lambda) \log(d_{\max}(\lambda)/\delta)$. Now note that we have

$$\lambda\langle \mathbb{P}f, (\mathbb{P}\Sigma\mathbb{P} + \lambda I)^{-1}\mathbb{P}f\rangle_\gamma = \lambda\langle \mathbb{P}f, (\Sigma + \lambda I)^{-1}\mathbb{P}f\rangle_\gamma = A(\mathbb{P}f, \lambda),$$

where the last equality follows from [7, Lemma 7.2]. Thus, we have proved the result for $f \in \mathcal{H}$. Given that (23) does not require $f$ to be in $\mathcal{H}$, we may conclude by limiting arguments that the result holds for any $f$ in the closure of $\mathcal{H}$, which includes $L^2(\gamma)$, since the kernel is universal, given that its associated RKHS is a weighted Sobolev Space, which is dense in $L^2(\gamma)$. $\square$

**Lemma C.4** (Degrees of freedom). *We have $d_{\max}(\lambda) \leq C/\lambda$, for an absolute constant $C > 0$.*

*Proof.* Recall that $d_{\max}(\lambda) := \sup_{b \in \mathbb{R}, \varepsilon \in \{\pm 1\}} \langle \mathbb{P}\phi_b^\varepsilon, (\mathbb{P}\Sigma\mathbb{P} + \lambda I)^{-1}\mathbb{P}\phi_b^\varepsilon\rangle_\gamma$. We consider two cases separately.

**If** $b \geq 0$, then we have $|\phi_b^\varepsilon(u)| \leq |u|$ for all $u$, thus

$$\langle \mathbb{P}\phi_b^\varepsilon, (\mathbb{P}\Sigma\mathbb{P} + \lambda I)^{-1}\mathbb{P}\phi_b^\varepsilon\rangle_\gamma \leq \frac{1}{\lambda}\|\mathbb{P}\phi_b^\varepsilon\|_\gamma^2 \leq \frac{1}{\lambda}\|\phi_b^\varepsilon\|_\gamma^2 \leq \frac{C}{\lambda},$$

with $C = 2\int_0^\infty u^2\gamma(u)du$.

**If** $b \leq 0$, we may write

$$\phi_b^\varepsilon(u) = \phi_{-b}^{-\varepsilon}(u) + g(u),$$

with $g(u) = \epsilon u - b$ a linear function, using the relation $\max(0, u) = u + \max(0, -u)$. Then we have

$$\begin{aligned}
\langle \mathbb{P}\phi_b^\varepsilon, (\mathbb{P}\Sigma\mathbb{P} + \lambda I)^{-1}\mathbb{P}\phi_b^\varepsilon\rangle_\gamma &\leq \frac{1}{\lambda}\|\mathbb{P}\phi_b^\varepsilon\|_\gamma^2 \\
&\leq \frac{2}{\lambda}(\|\phi_{-b}^{-\varepsilon}\|_\gamma^2 + \|\mathbb{P}g\|_\gamma^2) \leq \frac{4C}{\lambda},
\end{aligned}$$

with the same constant $C$ as in the previous case, since both $\phi_{-b}^{-\varepsilon}$ and $\mathbb{P}g(u) = (g - \mathbb{E}[g])(u) = \varepsilon u$ are controlled by $u \mapsto |u|$ in absolute value. $\square$

# D  Proofs for Section 5

## D.1  Exact expression of first-order critical points

We first derive critical point equations for the regularized population loss.

**Claim D.1** (Critical point equations). *Let $(\alpha_j)_{j \in \mathbb{N}}$ be the Hermite coefficients of $f_* \in L_2(\gamma)$, let $m \in [-1, 1]$, and let $g_m, \bar{g}_m \in L^2(\gamma)$ be defined by*

$$g_m(z) := \sum_{j=s}^{\infty} \alpha_j m^j h_j(z) \ , \ \bar{g}_m(z) := \sum_{j=s}^{\infty} \alpha_j j m^{j-1} h_j(z) \ . \tag{27}$$

*Then, denoting $m = \langle \theta^*, \theta \rangle$, we have*

$$\bar{L}(\theta) = -\langle \hat{P}_\lambda g_m, g_m \rangle_\gamma + (1 + \sigma^2) = -\|g_m\|_\gamma^2 + \langle (I - \hat{P}_\lambda) g_m, g_m \rangle_\gamma + (1 + \sigma^2) \ . \tag{28}$$

*Furthermore, the critical points of $L(c, \theta)$ satisfy the following equations:*

$$c = Q_\lambda^{-1} \sum_{j=s}^{\infty} \alpha_j m^j \mathcal{T}_j \ ,$$

$$0 = -\sum_{j=s}^{\infty} \alpha_j^2 j m^{2j-1} + \langle (I - \hat{P}_\lambda) g_m, \bar{g}_m \rangle_\gamma \ .$$

We prove Claim D.1 in Appendix D.3 by analyzing the population gradient and relating the critical points of $\bar{L}(\theta)$ to those of $L(c, \theta)$. Note that the function $g_m$ corresponds to the minimizer of $\|f_*(\langle \theta^*, \cdot \rangle) - g(\langle \theta, \cdot \rangle)\|_{\gamma_d}$, which is essentially the optimal function we may learn from fitting the second layer $c$ with no regularization when $\theta$ is fixed.

## D.2  Proof of Theorem 5.3

The intuition behind Theorem 5.3 is as follows. We first observe that for fixed $\theta \in \mathbb{S}^{d-1}$ and $\lambda > 0$, the population loss $L(c, \theta)$ is strictly convex in $c$. Hence, if $(c, \theta)$ is a critical point of $L(c, \theta)$, then $L(c, \theta) = \bar{L}(\theta)$. Since $\nabla_\theta^{\mathbb{S}^{d-1}} L(c, \theta) = 0$, it follows that $\theta$ must also be a critical point of the projected population loss $\bar{L}(\theta)$. Now consider the idealized (and impossible) setting in which $N$ is large enough to exactly express any function in $L^2(\gamma)$ and there is no $\ell^2$ regularization (i.e., $\lambda = 0$). Then, the projected population loss is

$$\bar{L}(\theta) = \min_{g \in L^2(\gamma)} \|f_*(\langle \theta^*, \cdot \rangle) - g(\langle \theta, \cdot \rangle)\|_{\gamma_d}^2 + \sigma^2 = -\sum_{j=s}^{\infty} \alpha_j^2 m^{2j} + (1 + \sigma^2) \ ,$$

where $m = \langle \theta^*, \theta \rangle$. $\bar{L}(\theta)$ in the ideal case is strictly decreasing in $|m| \in (0, 1]$. Using the expression for $\bar{L}(\theta)$ in Eq. (28), we observe that by setting $\lambda > 0$ sufficiently small (and $N$ proportional to $1/\lambda$), the projection $\hat{P}_\lambda$ onto the subspace $\hat{\mathcal{H}}$ spanned by random features approximates the identity map in the operator norm, thereby preserving the strict monotonicity of $\bar{L}(\theta)$ with respect to $|m|$. We formalize this intuition in the following proof.

We first establish the following lemma (proved in Section D.4) which controls the approximation error for the functions $g_m$ defined in (27).

**Lemma D.2** (Uniform approximation error for $g_m$). *Under the regularity assumptions on $f_*$ (Assumption 5.2), there exists a universal constant $C > 0$ and a constant $\tilde{K} \geq 1$ depending only on $f_*$ and* not *on $m$, such that for all $|m| \leq 1$,*

$$A(g_m, \lambda) \leq C(\tau^{1+\beta} \tilde{K}^2 \|g_m''\|_\gamma^2 \lambda^\beta + \lambda C_{g_m}^2) \ , \tag{29}$$

*where $\beta = \frac{1 - 1/\tau^2}{3 + 1/\tau^2}$ and $C_{g_m} = \max\{\|g_m\|_\gamma, \|g_m'\|_\gamma, \|g_m''\|_\gamma\}$.*

*Proof of Theorem 5.3.* For $(c, \theta)$ to be a critical point of $L$, it must satisfy $\nabla_\theta^{\mathbb{S}^{d-1}} L(c, \theta) = 0$ and $\nabla_c L(\theta, c) = 0$. By Claim D.1, for any $m = \langle \theta^*, \theta \rangle \in [-1, 1]$, there is a unique $c \in \mathbb{R}^N$ such that $\nabla_c L(\theta, c) = 0$ and $\nabla_\theta^{\mathbb{S}^{d-1}} L(c, \theta) = 0$ if and only if $\theta \in \{-\theta^*, \theta^*\}$, or

$$\sum_{j=s}^\infty \alpha_j^2 j m^{2j-1} = \langle (I - \hat{P}_\lambda) g_m, \bar{g}_m \rangle_\gamma. \tag{30}$$

We show that the latter condition is true only if $m = 0$ by deriving a contradiction whenever $m \neq 0$. Note that by the regularity assumption on $f_*$ (Assumption 5.2), it also holds that $\|f_*'\|_\gamma, \|f_*''\|_\gamma < \infty$. We contradict the equality in (30) with probability at least $1 - \delta$ over the randomly sampled biases $b_1, \dots, b_N$ and signs. Let $\tilde{K} \geq 1$ be the constant from Lemma D.2 and $C_{f_*} = \max\{\|f_*\|_\gamma, \|f_*'\|_\gamma, \|f_*''\|_\gamma\}$. Define the threshold for $\lambda$ by

$$\lambda^* := \left( \frac{4\sqrt{C \tau^{1+\beta}} \tilde{K} C_{f_*}^2}{\alpha_s^2 s} \right)^{-2/\beta}. \tag{31}$$

If $\lambda < \lambda^*$ and $N \geq \frac{C}{\lambda} \log \frac{1}{\lambda \delta}$, then with probability greater than $1 - \delta$ it holds

$$
\begin{aligned}
\left| \langle (I - \hat{P}_\lambda) g_m, \bar{g}_m \rangle_\gamma \right| &\leq \|(I - \hat{P}_\lambda) g_m\|_\gamma \|\bar{g}_m\|_\gamma \\
&\leq 2\sqrt{A(g_m, \lambda)} \|\bar{g}_m\|_\gamma && \text{[Lemma C.2]} \\
&\leq 2\sqrt{C(\tau^{1+\beta} \tilde{K}^2 \|g_m''\|_\gamma^2 \lambda^\beta + \lambda C_{g_m}^2)} \|\bar{g}_m\|_\gamma && \text{[Lemma D.2]} \\
&\leq 2\lambda^{\beta/2} C_{g_m} \sqrt{2C\tau^{1+\beta}} \tilde{K} \|\bar{g}_m\|_\gamma \\
&\leq 4\lambda^{\beta/2} |m|^{2s-1} \sqrt{C\tau^{1+\beta}} \tilde{K} C_{f_*}^2 && \text{[Lemma D.8]} \\
&< \alpha_s^2 s |m|^{2s-1} \leq \sum_{j=0}^\infty \alpha_j^2 j |m|^{2j-1} = \left| \sum_{j=0}^\infty \alpha_j^2 j m^{2j-1} \right|, && \text{[Eq (31)]},
\end{aligned}
$$

which contradicts (30). Therefore, the existence of critical points satisfying $m \notin \{-1, 0, 1\}$ is ruled out with probability at least $1 - \delta$ over the random features. $\qquad \square$

**Remark D.3** (Robust version of Theorem 5.3)**.** *The proof of Theorem 5.3 in fact implies a stronger result. It implies that if $\|\nabla^{\mathbb{S}^{d-1}} \bar{L}(\theta)\| \approx 0$, i.e., if $\theta$ is nearly a critical point for the projected population loss $\bar{L}$, then $|m| \approx 0$ or $|m| \approx 1$. This is formally stated in Lemma E.9.*

### D.3 First-order critical points of the population loss and proof of Claim D.1

To characterize the critical points of $L(c, \theta)$ for fixed random features and prove Claim D.1, we derive exact expressions for $L$ and its gradients. We observe that the population loss depends on the student direction $\theta$ only via its angle to the teacher direction $\theta^*$.

**Proposition D.4.** *The $\ell^2$-regularized population loss is given by*

$$L(c, \theta) = 1 + c^\top Q_\lambda c - 2\langle c, \sum_{j=s}^\infty \alpha_j m^j \mathcal{T}_j \rangle + \sigma^2. \tag{32}$$

*Proof.* Recall the decomposition $c^\top \Phi(z) = \sum_{j=0}^\infty \langle \mathcal{T}_j, c \rangle h_j(z)$. Straightforward calculation gives

$$
\begin{aligned}
L(c, \theta) &= \|(f(\langle \theta^*, \cdot \rangle) + \xi) - c^\top \Phi(\langle \theta, \cdot \rangle)\|_{\gamma_d}^2 + \lambda \|c\|^2 \\
&= \|f\|_\gamma^2 + \|c^\top \Phi\|_\gamma^2 - 2\langle f(\langle \theta^*, \cdot \rangle), c^\top \Phi(\langle \theta, \cdot \rangle) \rangle_{\gamma_d} + \sigma^2 + \lambda \|c\|^2 \\
&= 1 + \sum_{j=0}^\infty \langle \mathcal{T}_j, c \rangle^2 - 2\sum_{j=s}^\infty \langle \mathcal{T}_j, c \rangle \alpha_j \langle \theta, \theta^* \rangle^j + \sigma^2 + \lambda \|c\|^2 \\
&= 1 + c^\top Q_\lambda c - 2\langle c, \sum_{j=s}^\infty \alpha_j m^j \mathcal{T}_j \rangle + \sigma^2. && \square
\end{aligned}
$$

Eq. (32) gives us an expression for gradients with respect to $\theta \in \mathbb{S}^{d-1}$ and $c \in \mathbb{R}^N$.

**Corollary D.5** (Gradient of population loss). *The (non-spherical) gradient with respect to the student direction $\theta \in \mathbb{S}^{d-1}$ and $c \in \mathbb{R}^N$ are given by*

$$\nabla_\theta L = -\left( \sum_{j=s}^\infty \langle \mathcal{T}_j, c \rangle j \alpha_j m^{j-1} \right) \theta^* \quad and \quad \nabla_c L = 2 \left( Q_\lambda c - \sum_{j=s}^\infty \alpha_j m^j \mathcal{T}_j \right) . \tag{33}$$

Recall that the criticality of $\theta$ depends on the *spherical* gradient being zero, not the standard one. Since the gradient $\nabla_\theta L$ is colinear with $\theta^*$, we stipulate necessary and sufficent conditions for $\theta$ to be critical.

**Corollary D.6** (Projection onto the sphere). $\nabla_\theta^{\mathbb{S}^{d-1}} L = 0$ *if and only if either (i)* $\theta = \theta^*$ *(i.e., $m = 1$) or (ii)* $\sum_{j=s}^\infty \langle \mathcal{T}_j, c \rangle j \alpha_j m^{j-1} = 0$.

Identifying the critical values of $\theta$ is sufficient because the *unique* critical point for $c$ exists as the solution to a linear system. To elaborate, $Q_\lambda \in \mathbb{R}^{N \times N}$ is non-singular for $\lambda > 0$, so if $\theta \in \mathbb{S}^{d-1}$ is fixed, then $c$ is given as the solution to a linear system of equations: $c = Q_\lambda^{-1}(\sum_j \alpha_j m^j \mathcal{T}_j)$.

By Proposition D.4, $L(c, \theta)$ is strictly convex with respect to $c \in \mathbb{R}^N$ for any fixed $\theta \in \mathbb{S}^{d-1}$. Hence, if $(c, \theta)$ is a critical point of $L$, then $\theta$ must be a critical point of $\bar{L}$.

**Lemma D.7** (Critical points of the *projected* population loss). *Recall that* $g_m(z) = \sum_{j=s}^\infty \alpha_j m^j h_j(z)$. *Then, the projected population loss $\bar{L}$ is given by*

$$\bar{L}(\theta) = -\sum_{j,j'} \alpha_j \alpha_{j'} m^{j+j'} \langle h_j, \hat{P}_\lambda h_{j'} \rangle_\gamma + (1 + \sigma^2) \tag{34}$$

$$= -\langle g_m, \hat{P}_\lambda g_m \rangle_\gamma + (1 + \sigma^2) .$$

*Furthermore, critical points of $\bar{L}$ satisfy the following equation.*

$$\sum_{j,j'} \alpha_j \alpha_{j'} (j + j') m^{j+j'-1} \langle h_j, P_\lambda h_{j'} \rangle_\gamma = 0 . \tag{35}$$

*Proof of Lemma D.7.* Because $\mathcal{T}$ is full-rank, let $\mathcal{T} = U\Lambda\mathcal{V}$ be its SVD for some $U \in \mathbb{R}^{N \times N}$, diagonal $\Lambda \in \mathbb{R}^{N \times N}$, and $\mathcal{V} : L^2(\mu) \to \mathbb{R}^N$. Then, $\mathcal{T}_j = U\Lambda\mathcal{V}h_j$, $Q_\lambda = U^\top(\Lambda^2 + \lambda I_N)U$, and $Q_\lambda^{-1} = U^\top(\Lambda^2 + \lambda I_N)^{-1}U$. Similarly, $\hat{P}_\lambda = \mathcal{V}^*\Lambda(\Lambda^2 + \lambda I_N)^{-1}\Lambda\mathcal{V}$. As a result,

$$\langle \mathcal{T}_j, Q_\lambda^{-1} \mathcal{T}_{j'} \rangle = \langle h_j, \mathcal{V}^*\Lambda(\Lambda^2 + \lambda I_N)^{-1}\Lambda\mathcal{V}h_{j'} \rangle_\gamma = \langle h_j, \hat{P}_\lambda h_{j'} \rangle_\gamma.$$

Now we plug in $c = Q_\lambda^{-1}(\sum_j \alpha_j m^j \mathcal{T}_j)$ into Eq. (32). Then, we have

$$\bar{L}(\theta) - (1 + \sigma^2) = -\left\langle \sum_j \alpha_j m^j \mathcal{T}_j, Q_\lambda^{-1} \sum_{j'} \alpha_{j'} m^{j'} \mathcal{T}_{j'} \right\rangle$$

$$= -\sum_{j,j'} \alpha_j \alpha_{j'} m^j m^{j'} \left\langle \mathcal{T}_j, Q_\lambda^{-1} \sum_{j'} \mathcal{T}_{j'} \right\rangle$$

$$= -\sum_{j,j'} \alpha_j \alpha_{j'} m^j m^{j'} \langle h_j, \hat{P}_\lambda h_{j'} \rangle_\gamma .$$

Differentiating $\bar{L}(\theta)$ with respect to $m = \langle \theta, \theta^* \rangle$, we obtain the following critical point equation.

$$\sum_{j,j'} \alpha_j \alpha_{j'} (j + j') m^{j+j'-1} \langle h_j, P_\lambda h_{j'} \rangle_\gamma = 0 .$$

$\square$

**Claim D.1** (Critical point equations). *Let $(\alpha_j)_{j\in\mathbb{N}}$ be the Hermite coefficients of $f_* \in L_2(\gamma)$, let $m \in [-1, 1]$, and let $g_m, \bar{g}_m \in L^2(\gamma)$ be defined by*

$$g_m(z) := \sum_{j=s}^{\infty} \alpha_j m^j h_j(z) \ , \quad \bar{g}_m(z) := \sum_{j=s}^{\infty} \alpha_j j m^{j-1} h_j(z) \ . \tag{27}$$

*Then, denoting $m = \langle \theta^*, \theta \rangle$, we have*

$$\bar{L}(\theta) = -\langle \hat{P}_\lambda g_m, g_m \rangle_\gamma + (1 + \sigma^2) = -\|g_m\|_\gamma^2 + \langle (I - \hat{P}_\lambda) g_m, g_m \rangle_\gamma + (1 + \sigma^2) \ . \tag{28}$$

*Furthermore, the critical points of $L(c, \theta)$ satisfy the following equations:*

$$c = Q_\lambda^{-1} \sum_{j=s}^{\infty} \alpha_j m^j \mathcal{T}_j \ ,$$

$$0 = -\sum_{j=s}^{\infty} \alpha_j^2 j m^{2j-1} + \langle (I - \hat{P}_\lambda) g_m, \bar{g}_m \rangle_\gamma \ .$$

*Proof of Claim D.1.* As discussed above $(c, \theta)$ is a critical point of $L$ if and only if $\theta$ is a critical point of $\bar{L}$ and $c = Q_\lambda^{-1}(\sum_j \alpha_j m^j \mathcal{T}_j)$. By applying Lemma D.7, we separate the diagonal and off-diagonal terms to rewrite Eq. (35) as

$$0 = \sum_j \alpha_j^2 (2j) m^{2j-1} - 2\langle (I - \hat{P}_\lambda) g_m, \bar{g}_m \rangle_\gamma \ .$$

Dividing both sides by 2 gives the claim. □

## D.4 Proof of Lemma D.2

**Lemma D.2** (Uniform approximation error for $g_m$). *Under the regularity assumptions on $f_*$ (Assumption 5.2), there exists a universal constant $C > 0$ and a constant $\tilde{K} \geq 1$ depending only on $f_*$ and not on $m$, such that for all $|m| \leq 1$,*

$$A(g_m, \lambda) \leq C(\tau^{1+\beta} \tilde{K}^2 \|g_m''\|_\gamma^2 \lambda^\beta + \lambda C_{g_m}^2) \ , \tag{29}$$

*where $\beta = \frac{1 - 1/\tau^2}{3 + 1/\tau^2}$ and $C_{g_m} = \max\{\|g_m\|_\gamma, \|g_m'\|_\gamma, \|g_m''\|_\gamma\}$.*

*Proof.* For simplicity, we denote the $L^p(\gamma)$ norms by $\| \cdot \|_p$. For any $\rho \in [0, 1]$, we define the noise operator $U_\rho$ by

$$U_\rho f(x) = \mathbb{E}_{z\sim\gamma}[f(\rho x + \sqrt{1 - \rho^2} z)]$$

This is a reparametrisation of the *Ornstein-Uhlenbeck* semigroup,[2] and we have from [67, Prop 11.33] that $U_\rho h_j = \rho^j h_j$. In other words, the Hermite polynomials are eigenfunctions of the semigroup. As a consequence, we have from (27) that $g_m = U_m f_*$.

Let us verify that if $f_*'' \in L^4(\gamma)$ and $K = \|f_*''\|_4 / \|f_*''\|_2$, then there exists a constant $\tilde{K} \geq 1$, depending only on $f_*$ and *not* $m$, such that $\|g_m''\|_4 \leq \tilde{K} \|g_m''\|_2$ for any $m \in [-1, 1]$. If $f_*'' \equiv 0$, then $g_m'' \equiv 0$ for any $m$, and thus $\tilde{K} = 1$. Otherwise, let $\tilde{s}$ denote the information exponent of $f_*'' \neq 0$. That is, if $s \geq 2$, then $\tilde{s} = s - 2$, and if $s = 1$, then $\tilde{s} = s_2 - 2$ where $s_2$ is the second non-zero harmonic of $f_*$. For $|m| \leq 1/\sqrt{3}$, we have

$$g_m'' = m^2 U_m[f_*''] = m^2 U_{\sqrt{1/3}} U_{m\sqrt{3}}[f_*''] \ .$$

---

[2]The Ornstein–Uhlenbeck semigroup $\mathcal{P}_t$ is given by $\mathcal{P}_t f(x) = \int f(e^{-t}x + \sqrt{1 - e^{-2t}} z) d\gamma(z)$ . We thus have $\mathcal{P}_t = U_{e^{-t}}$

Since $|m\sqrt{3}| \le 1$, we have

$$\left\|U_{m\sqrt{3}}[f_*'']\right\|_2^2 = \sum_{j=\tilde{s}}^{\infty}(j+2)(j+1)\alpha_{j+2}^2(m\sqrt{3})^{2j} \le (m\sqrt{3})^{2\tilde{s}}\sum_{j=\tilde{s}}^{\infty}(j+2)(j+1)\alpha_{j+2}^2 = (m\sqrt{3})^{2\tilde{s}}\|f_*''\|_2^2 \ .$$

By Nelson's Gaussian hypercontractivity [65] (reproduced in [67, Theorem 11.23]),

$$\|g_m''\|_4 = m^2\left\|U_{\sqrt{1/3}}\Big(U_{m\sqrt{3}}[f_*'']\Big)\right\|_4 \le m^2\|U_{m\sqrt{3}}[f_*'']\|_2 \le |m|^{\tilde{s}+2}\cdot 3^{\tilde{s}/2}\|f_*''\|_2$$

Since $\|g_m''\|_2 \ge |m|^{\tilde{s}+2}\sqrt{(\tilde{s}+2)(\tilde{s}+1)}|\alpha_{\tilde{s}+2}|$, we obtain that

$$\sup_{|m|\le 1/\sqrt{3}}\frac{\|g_m''\|_4}{\|g_m''\|_2} \le \frac{3^{\tilde{s}/2}}{\sqrt{(\tilde{s}+2)(\tilde{s}+1)}|\alpha_{\tilde{s}+2}|}\cdot\|f_*''\|_2 \ .$$

Let us now consider $m \ge 1/\sqrt{3}$. Since $U_\rho$ is an averaging operator for all $\rho \le 1$, from Jensen's inequality (reproduced in [67, Proposition 11.15]) it holds that $\|U_\rho f\|_p \le \|f\|_p$ for any $p \ge 1$. We thus have

$$\|g_m''\|_4 \le m^2\|f_*''\|_4 \ .$$

and $\|g_m''\|_2 \ge 3^{-(\tilde{s}+2)/2}\sqrt{(\tilde{s}+2)(\tilde{s}+1)}|\alpha_{\tilde{s}+2}|$, therefore

$$\sup_{|m|\ge 1/\sqrt{3}}\frac{\|g_m''\|_4}{\|g_m''\|_2} \le \frac{3^{(\tilde{s}+2)/2}\|f_*''\|_4}{\sqrt{(\tilde{s}+2)(\tilde{s}+1)}|\alpha_{\tilde{s}+2}|} \le \frac{3^{(\tilde{s}+2)/2}K\|f_*''\|_2}{\sqrt{(\tilde{s}+2)(\tilde{s}+1)}|\alpha_{\tilde{s}+2}|} \ .$$

Hence, we set

$$\tilde{K} = \frac{3^{(\tilde{s}+2)/2}\|f_*''\|\cdot\max\{K,1/3\}}{\sqrt{(\tilde{s}+2)(\tilde{s}+1)}|\alpha_{\tilde{s}+2}|} = \frac{3^{s/2}K\|f_*''\|}{\sqrt{(\tilde{s}+2)(\tilde{s}+1)}|\alpha_{\tilde{s}+2}|} \ .$$

By Lemma 4.4, we therefore have

$$\forall\, m \in [-1,1], \ A(g_m,\lambda) \le C(\tau^{1+\beta}\tilde{K}^2\|g_m''\|_\gamma^2\lambda^\beta + C_{g_m}^2\lambda) \ , \tag{36}$$

where $\beta = \frac{1-1/\tau^2}{3+1/\tau^2}$, $C > 0$ is a universal constant, and $C_{g_m} = \max\{\|g_m\|_\gamma, \|g_m'\|_\gamma, \|g_m''\|_\gamma\}$.

$\square$

## D.5 Other lemmas for the proof of Theorem 5.3

**Lemma D.8** ($\gamma$-norm of $g_m$ and $\bar{g}_m$). *Let $f \in H^2(\gamma)$ be such that $f', f'' \in L^2(\gamma)$ and let $s \ge 1$ be its information exponent. Furthermore, let $f_* = \sum_j \alpha_j h_j$ be the Hermite expansion of $f_*$, and let $g_m$ and $\bar{g}_m$ be defined as in Theorem 5.3. Then,*

$$\|g_m\|_\gamma^2 \le \|f_*\|_\gamma^2 m^{2s},$$
$$\|g_m''\|_\gamma^2 \le \|f_*''\|_\gamma^2 m^{2s},$$
$$\|\bar{g}_m\|_\gamma^2 \le \left(\|f_*''\|_\gamma^2 + \|f_*'\|_\gamma^2\right)m^{2(s-1)} \ .$$

*Proof.* By definition of $g_m$ and Holder's inequality,

$$\|g_m\|_\gamma^2 = \sum_{j=s}^{\infty}\alpha_j^2 m^{2j} \le \|\alpha\|_2^2\max_{j\ge s}m^{2j} = \|f_*\|_\gamma^2 m^{2s},$$

$$\|g_m''\|_\gamma^2 = \sum_{j=2}^{\infty}j(j-1)\cdot\alpha_j^2 m^{2j} \le \left(\sum_{j=2}^{\infty}j(j-1)\alpha_j^2\right)\max_{j\ge s}m^{2j} = \|f_*''\|_\gamma^2 m^{2s},$$

$$\|\bar{g}_m\|_\gamma^2 = \sum_{j=1}^{\infty}j^2\alpha_j^2 m^{2(j-1)} = \sum_{j=s}^{\infty}j(j-1)\alpha_j^2 m^{2(j-1)} + \sum_{j=s}^{\infty}j\alpha_j^2 m^{2(j-1)} \le m^{2(s-1)}\left(\|f_*''\|_\gamma^2 + \|f_*'\|_\gamma^2\right).$$

$\square$

**Corollary D.9.** *Let $f \in H^2(\gamma)$ be a function satisfying assumptions of Lemma D.8, let $g_m$ and $\bar{g}_m$ be defined as in Theorem 5.3, and let $C_{g_m} = \max\{\|g_m\|, \|g'_m\|_\gamma, \|g''_m\|_\gamma\}$. Then,*

$$C_{g_m}^2 \leq \sum_{j=s}^\infty j^2 \alpha_j^2 m^{2j} \leq 2C_{g_m}^2 \quad and \quad C_{g_m}\|\bar{g}_m\|_\gamma \leq \sqrt{2}\sum_{j=s}^\infty j^2\alpha_j^2 m^{2j-1} \; .$$

*Proof.* The first inequality follows from the following.

$$C_{g_m}^2 \leq \sum_{j=s}^\infty j^2\alpha_j^2 m^{2j} = \|g''_m\|_\gamma^2 + \|g'_m\|_\gamma^2 \leq 2C_{g_m}^2 \; .$$

The proof of the second inequality is via straightforward algebraic manipulation.

$$
\begin{aligned}
C_{g_m}\|\tilde{g}_m\|_\gamma &\leq \left(\sum_{j=s}^\infty j^2\alpha_j^2 m^{2j}\right)^{1/2}\left(\sum_{j=s}^\infty j^2\alpha_j^2 m^{2(j-1)}\right)^{1/2}\\
&= \left(\sum_{j=s}^{\tilde{s}-1} j^2\alpha_j^2 m^{2j} + \sum_{j=\tilde{s}}^\infty j^2\alpha_j^2 m^{2j}\right)^{1/2}\left(\sum_{j=s}^{\tilde{s}-1} j^2\alpha_j^2 m^{2(j-1)} + \sum_{j=\tilde{s}}^\infty j^2\alpha_j^2 m^{2(j-1)}\right)^{1/2}\\
&= \left(\left(\sum_{j=s}^{\tilde{s}-1} j^2\alpha_j^2 |m|^{2j-1}\right)^2 + \left(\sum_{j=s}^{\tilde{s}-1} j^2\alpha_j^2 m^{2j}\right)\left(\sum_{j=\tilde{s}}^\infty j^2\alpha_j^2 m^{2(j-1)}\right) + \left(\sum_{j=\tilde{s}}^\infty j^2\alpha_j^2 |m|^{2j-1}\right)^2\right)^{1/2}\\
&= \left(\left(\sum_{j=s}^{\tilde{s}-1} j^2\alpha_j^2 |m|^{2j-1}\right)^2 + \left(\sum_{j=s}^{\tilde{s}-1} j^2\alpha_j^2 |m|^{2j-1}\right)\left(\sum_{j=\tilde{s}}^\infty j^2\alpha_j^2 |m|^{2j-1}\right) + \left(\sum_{j=\tilde{s}}^\infty j^2\alpha_j^2 |m|^{2j-1}\right)^2\right)^{1/2}\\
&\leq \left(\left(\sum_{j=s}^{\tilde{s}-1} j^2\alpha_j^2 |m|^{2j-1}\right)^2 + \left(\sum_{j=s}^{\tilde{s}-1} j^2\alpha_j^2 |m|^{2j-1}\right)^2 + \left(\sum_{j=\tilde{s}}^\infty j^2\alpha_j^2 |m|^{2j-1}\right)^2 + \left(\sum_{j=\tilde{s}}^\infty j^2\alpha_j^2 |m|^{2j-1}\right)^2\right)^{1/2}\\
&\leq \left(2\left(\sum_{j=s}^{\tilde{s}-1} j^2\alpha_j^2 |m|^{2j-1}\right)^2 + 2\left(\sum_{j=\tilde{s}}^\infty j^2\alpha_j^2 |m|^{2j-1}\right)^2\right)^{1/2}\\
&\leq \sqrt{2}\sum_{j=s}^\infty j^2\alpha_j^2 |m|^{2j-1} \; .
\end{aligned}
$$

where we used the inequalities $ab \leq a^2 + b^2$, and $\sqrt{a^2+b^2} \leq a+b$ which apply to any $a, b \geq 0$. $\quad\square$

# E    Proofs for Section 6

## E.1    Proof of Theorem 6.1

**Theorem 6.1** (Gradient flow finds approximate minimizers). *For $\delta \in (0, 1/4)$ and $f_*$ satisfying Assumption 5.2, suppose the following are true: (i) $\lambda = O(1)$ and $\lambda = \Omega(\sqrt{\Delta_{crit}})$, where $\Delta_{crit} := \max\{\sqrt{\frac{d+N}{n}}, (\frac{d^2}{n})^{2s/(2s-1)}\}$, (ii) $n = \widetilde{\Omega}(\max\{\frac{(d+N)d^{s-1}}{\lambda^4}, \frac{d^{(s+3)/2}}{\lambda^2}\})$, (iii) $N = \Omega(\frac{1}{\lambda}\log\frac{1}{\lambda\delta})$ & $N = \widetilde{O}(\lambda\Delta_{crit}^{-1})$, (iv) $N_0 = \Theta(\log(\frac{1}{\delta}))$, (v) $\rho = \Theta(\sqrt{N}N_0^{-(2+s)/2}(\tau^2 + \lambda N/N_0)^{-1})$, (vi) $T_0 = \tilde{\Theta}(d^{s/2-1})$, and (vii) $T_1 = \tilde{\Theta}(\frac{\lambda^4 n}{d+N})$. Then, if we run Procedure 1 for $T = T_0 + T_1$ time steps with the above parameters, with probability at least $\frac{1}{2} - \delta$ we have*

$$1 - |\langle \theta_T, \theta^* \rangle| \;\; = \;\; \widetilde{O}\left(\lambda^{-4}\max\left\{\frac{d+N}{n}, \frac{d^4}{n^2}\right\}\right) \; . \tag{11}$$

*Proof.* The proof of this theorem has two separate parts: we first prove that our gradient flow procedure escapes the neighborhood of the equator, and then show that it converges to a neighborhood

of the north pole. Define the set of approximate-first-order critical points of the empirical landscape in the sublevel set $L_n(c, \theta) \le \nu$.

$$\Omega_n(\epsilon_\theta, \epsilon_c, \nu) := \{(c, \theta); \|\nabla_\theta L_n(c, \theta)\| \le \epsilon_\theta, \|\nabla_c L_n(c, \theta)\| \le \epsilon_c; L_n(c, \theta) \le \nu\}$$

Recall from Theorem 5.3 that the structure of critical points of the population landscape $\Omega_\infty(0, 0, \infty)$ has two distinct components: the equator $\mathrm{E} = \{\theta; \langle\theta, \theta^*\rangle = 0\}$ and the poles $\theta = \pm\theta^*$, leading to

$$\Omega_\infty(0, 0, \infty) = \Omega_\infty(0, 0, 0) = \underbrace{\{(0, \theta); \theta \in \mathrm{E}\}}_{:=\Omega_\infty^b} \sqcup \underbrace{\{(zc^*, z\theta^*); z = \{-1, +1\}\}}_{:=\Omega_\infty^g},$$

with $c^* = \hat{P}_\lambda f_*$. One would expect that for $n$ sufficiently large, these topological properties should be transferred to the empirical landscape. This intuition is indeed correct, and relies on the following uniform convergence result, proved in Appendix E.2.

**Lemma E.1** (Uniform convergence of the empirical landscape). *Let $d, n, N \in \mathbb{N}$ be such that $d \le n$, let $D = \max\{d, N\}$, let $\delta \in (0, 1/4)$, $r \ge 1$, and let $\sigma^2 > 0, \tau^2 > 1$ be the variance of label noise and random feature biases, respectively. Under Assumption 5.2, there exists a universal constant $C_0 > 0$ such that the following holds with probability at least $1 - \delta$ over the samples and random features.*

$$\sup_{\theta \in \mathbb{S}^{d-1}, \|c\| \le r} \|\nabla_\theta L_n(c, \theta) - \nabla_\theta L(c, \theta)\| \le C_{f_*} C r^2 \cdot \max\left(\sqrt{\frac{D \log(nN/\delta)}{n}}, \frac{(d \log(nN/\delta))^2}{n}\right)$$

$$\sup_{\theta \in \mathbb{S}^{d-1}, \|c\| \le r} \|\nabla_c L_n(c, \theta) - \nabla_c L(c, \theta)\| \le C^2 r \cdot \sqrt{\frac{D \log(n/\delta)}{n}},$$

*where $C = C_0 \cdot \max\{\mathrm{Lip}(f_*), \tau\sqrt{\log(1/\delta)}, \sigma\}$, and $C_{f_*} = \max\{\|f_*^{(1)}\|_\gamma, \dots, \|f_*^{(4)}\|_\gamma, 1\}$.*

Equipped with this uniform gradient concentration, we can first establish the analogous classification of first-order critical points for the empirical landscape (proof in Section E.3):

**Lemma E.2** (Local sharpness of the empirical landscape). *Let $d, n \in \mathbb{N}$ be such that $d \le n$, let $\delta \in (0, 1/4)$, let $\tilde{s} \in \mathbb{N}$ be such that $\tilde{s} \ge s$, where $s \ge 1$ is the information exponent of $f_*$, let $\lambda \in (0, \lambda_{\tilde{s}}^*)$ where $\lambda_{\tilde{s}}^* \le 1$ depends only on $\tilde{s}$, $f_*$, and $\tau$, and let $N \in \mathbb{N}$ be such that $N \ge \frac{C_0}{\lambda} \log \frac{1}{\lambda\delta}$, where $C_0 > 0$ is a universal constant. Furthermore, let $D = \max\{d, N\}$ and let $\epsilon \in (0, 1)$ be such that $\epsilon \le \lambda^{-2}\sqrt{d/n}$. Then, there exists a universal constant $C_1 > 0$ such that for $C = C_1 \cdot \max\{\mathrm{Lip}(f_*), \tau\sqrt{\log(1/\delta)}, \sigma\}$ and $\Delta = \max\left\{\sqrt{\frac{D \log(n/\delta)}{n}}, \frac{(d \log(n/\delta))^2}{n}\right\}$ the following holds with probability at least $1 - \delta$ over the samples and random features.*

$$\Omega_n(\epsilon, \epsilon) = \Omega_n^{\mathrm{bad}} \sqcup \Omega_n^{\mathrm{good}}, \text{where} \tag{37}$$

$$\Omega_n^{\mathrm{bad}} \subset \left\{(c, \theta) \,\middle|\, |m| \le \left(\frac{C_2 C^7}{\lambda^2} \cdot \Delta\right)^{\frac{1}{2\tilde{s}-1}} \wedge \|c\| \le \frac{C^3}{\lambda}\right\},$$

$$\Omega_n^{\mathrm{good}} \subset \left\{(c, \theta) \,\middle|\, 1 - |m| \le \frac{C_3 C^{14}}{\lambda^4} \cdot \Delta^2 \wedge \|c\| \le \frac{C^3}{\lambda}\right\},$$

*and $C_2 = C_{f_*}/\tilde{s}\alpha_{\tilde{s}}^2$ and $C_3 = (2^{2\tilde{s}-1} C_2)^2$ in the above display. Moreover,*

$$\min_{(c, \theta) \in \Omega_n^{\mathrm{bad}}} L_n(c, \theta) \ge \sigma^2 + \|f_*\|_\gamma^2 - 2\max\{\|f_*\|_\gamma^2 C_2^2 C^{14}, C^8\} \cdot \Delta_{\mathrm{crit}}/\lambda^2, \tag{38}$$

*where*

$$\Delta_{\mathrm{crit}} = \max\left\{\sqrt{\frac{D \log(n/\delta)}{n}}, \left(\frac{d^2 \log^2(n/\delta)}{n}\right)^{\frac{2\tilde{s}}{2\tilde{s}-1}}\right\}. \tag{39}$$

We consider the gradient flow procedure of Algorithm 1, that we restate here for convenience:

We establish the following fact, proved in Appendix E.4:

---

**Procedure 3** Gradient Flow with time-scale scheduling (Restated)

---

**Require:** $N_0, \rho, T_0, T_1$
    Initialise $\theta(0) \sim \mathrm{Unif}(\mathbb{S}^{d-1})$, $c(0) \sim \mathrm{Unif}(\{c \in \mathbb{R}^N; \|c\|_2 = \rho; \|c\|_0 = N_0\})$.
    Run Gradient Flow (10) with $\zeta(t) = \mathbf{1}(t > T_0)$ up to time $T = T_0 + T_1$.

---

**Lemma E.3** (Gradient flow escapes the equator). *Assume $\sum_j (j+A)^k \alpha_j^2 \le C$ for $A \le s$ and $k \le 3$. With probability at least $1/2 - 2\delta$ over the initial condition, the draw of the data, and the draw of the random features, if $n = \tilde{\Omega}(\max\{\lambda^{-4}(d+N)d^{s-1}, \lambda^{-2}d^{(s+3)/2}\})$ and $N = \Theta\left(\lambda^{-1}\log(\lambda^{-1}\delta^{-1})\right)$ then the first phase of gradient flow with a randomly initialised $c(0) \sim \mathrm{Unif}\{c \in \rho\mathbb{S}^{N-1}; \|c\|_0 = N_0\}$ with $\rho = \Theta(\sqrt{N}N_0^{-(2+s)/2}(\tau^2 + \lambda N/N_0)^{-1})$ and $N_0 = \Theta\left(\log\frac{1}{\delta}\right)$ escapes the equator in time $T_0 = \widetilde{O}\left(d^{s/2-1}\right)$.*

Therefore, for any $\epsilon > 0$, Gradient Flow converges to an $\epsilon$-approximate first-order critical point *at energy lower than* $B_{\mathrm{crit}} = \tilde{\Theta}\left(\lambda^{-2}\Delta_{crit}\right)$ in time $\widetilde{O}(\epsilon^{-2}) = \widetilde{O}(\frac{\lambda^4 n}{D})$, since gradient flow is a descent curve under our settings (see Appendix F), and therefore satisfies

$$L_n(c(T), \theta(T)) - L_n(c(0)) = -\int_0^T \|\nabla L_n(c(t), \theta(t))\|^2 dt . \tag{40}$$

By Lemma E.2, such critical points can only be in $\Omega_n^{\mathrm{good}}$, which yields the result. $\qquad\square$

## E.2    Proof of Lemma E.1

**Lemma E.1** (Uniform convergence of the empirical landscape). *Let $d, n, N \in \mathbb{N}$ be such that $d \le n$, let $D = \max\{d, N\}$, let $\delta \in (0, 1/4)$, $r \ge 1$, and let $\sigma^2 > 0, \tau^2 > 1$ be the variance of label noise and random feature biases, respectively. Under Assumption 5.2, there exists a universal constant $C_0 > 0$ such that the following holds with probability at least $1 - \delta$ over the samples and random features.*

$$\sup_{\theta \in \mathbb{S}^{d-1}, \|c\| \le r} \|\nabla_\theta L_n(c, \theta) - \nabla_\theta L(c, \theta)\| \le C_{f_*}Cr^2 \cdot \max\left(\sqrt{\frac{D\log(nN/\delta)}{n}}, \frac{(d\log(nN/\delta))^2}{n}\right)$$

$$\sup_{\theta \in \mathbb{S}^{d-1}, \|c\| \le r} \|\nabla_c L_n(c, \theta) - \nabla_c L(c, \theta)\| \le C^2 r \cdot \sqrt{\frac{D\log(n/\delta)}{n}} ,$$

*where $C = C_0 \cdot \max\{\mathrm{Lip}(f_*), \tau\sqrt{\log(1/\delta)}, \sigma\}$, and $C_{f_*} = \max\{\|f_*^{(1)}\|_\gamma, \ldots, \|f_*^{(4)}\|_\gamma, 1\}$.*

*Proof.* Recall the empirical loss of equation (1) for $c \in \mathbb{R}^N$ and $\theta \in \mathbb{S}^{d-1}$:

$$L_n(c, \theta) = \frac{1}{n}\sum_{i=1}^n \ell(c, \theta; x_i, y_i) + \lambda\|c\|^2 ,$$

where

$$\ell(c, \theta; x, y) = \left(c^\top \Phi(\langle x, \theta \rangle) - y\right)^2 .$$

**Uniform convergence of $\nabla_\theta L_n$.**    Our proof tracks closely the argument in [33, Section G.1] and [60, Theorem 1], but takes additional steps to handle the non-Lipschitzness of the sample gradient around 0. For simplicity, we write $C = \max\{\mathrm{Lip}(f_*), \tau\sqrt{\log(1/\delta)}, \sigma\}$, i.e., omit the universal constant $C_0 > 0$, with the understanding that $C_0$ is implicitly determined by accounting for all occurrences of $\lesssim$ in our analysis. We also repeatedly use the following elementary fact: If $C_1, C_2 \ge 1$, then $C_1 + C_2 \le 2\max\{C_1, C_2\} \le 2C_1 C_2$. We first recall basic concentration properties of standard Gaussian random variables.

**Fact E.4** (Niceness of Gaussian random variables). *Let $\delta \in (0, 1/4)$, $N \in \mathbb{N}$, and let $b_1, \ldots, b_N$ be i.i.d. random variables drawn from $\mathcal{N}(0, \tau^2)$. Then, there exists a universal constant $C' > 0$ such*

*that the following two events hold simultaneously with probability at least $1 - \delta$.*

$$\max_j |b_j| \le C'\tau\sqrt{\log(N/\delta)}\,,$$

$$\sum_j b_j^2 \le N\tau^2 + C'\tau^2 \max\left\{\log(1/\delta), \sqrt{N\log(1/\delta)}\right\}\,.$$

**Corollary E.5** ($\ell_2$-norm of random features)**.** *Let $\delta \in (0, 1/4)$ and let $b_1, \ldots, b_N$ be i.i.d. random variables drawn from $\mathcal{N}(0, \tau^2)$. Then, there exists a universal constant $C' > 0$ such that the following holds for all $z \in \mathbb{R}$ with probability at least $1 - \delta$ over the random features,*

$$\|\Phi(z)\| \le |z| + C'\tau(1 + \sqrt{\log(1/\delta)/N}) \le |z| + 2C'\tau\sqrt{\log(1/\delta)}\,.$$

*Proof.* Let $C_1 = \max\{C, 1\}$, where $C > 0$ is the constant from Fact E.4. Then, with probability at least $1 - \delta$ over the random features,

$$
\begin{aligned}
\|\Phi(z)\| &= \sqrt{\frac{1}{N}\sum_{j=1}^N \phi(\zeta_j z - b_j)^2} \\
&\le \sqrt{\frac{1}{N}\sum_{j=1}^N (\zeta_j z - b_j)^2} \\
&= \sqrt{z^2 - \frac{2z}{N}\sum_{j=1}^N \zeta_j b_j + \frac{1}{N}\sum_{j=1}^N b_j^2} \\
&\le \sqrt{z^2 + 2|z|C_2\tau\sqrt{\log(1/\delta)/N} + \tau^2(1 + C_1\max\{\log(1/\delta)/N, \sqrt{\log(1/\delta)/N}\})} \\
&\le |z| + \max\{C_2^2, 2C_1\}\cdot\tau(1 + \sqrt{\log(1/\delta)/N}))\,.
\end{aligned}
$$

$\square$

Fix $c \in \mathbb{R}^N$ such that $\|c\| \le r$. For $x \sim \mathcal{N}(0, I_d)$ consider the random vector corresponding to the samplewise gradient with respect to $\theta \in \mathbb{S}^{d-1}$.

$$\nabla_\theta \ell(c, \theta; x, f_*(\langle x, \theta_* \rangle)) = (c^\top \Phi'(\langle \theta, x \rangle))\left(c^\top \Phi(\langle \theta, x \rangle) - f_*(\langle x, \theta_* \rangle) - \xi\right)x\,.$$

The tail of this random vector is subexponential, as stated in the following lemma.

**Lemma E.6** (Sub-exponential gradients)**.** *Let $f_* : \mathbb{R} \to \mathbb{R}$ be a Lipschitz function, let $\delta \in (0, 1/4)$, let $r \ge 1$, and let $\tau^2 > 1$ be the variance of random feature biases. Then, there exists a universal constant $C' > 0$ such that the following holds with probability at least $1 - \delta$ over the random features.*

$$\|\nabla_\theta \ell(c, \theta; x, f_*(\langle x, \theta^* \rangle) + \xi)\|_{\psi_1} \le C'\|c\|(\mathrm{Lip}(f_*) + \sigma + \|c\|\tau(1 + \sqrt{\log(1/\delta)/N})) \le 2C'Cr^2\,,$$
(41)

*where $C = \max\{\mathrm{Lip}(f_*), \tau\sqrt{\log(1/\delta)}, \sigma\}$.*

*Proof of Lemma E.6.* Define $W = c^\top \Phi(\langle \theta, x \rangle) - f_*(\langle \theta^*, x \rangle) - \xi$. Using Fact B.3,

$$
\begin{aligned}
\|\nabla_\theta \ell(c, \theta; x, f_*(\langle x, \theta^* \rangle))\|_{\psi_1} &\le \sup_{v \in \mathbb{S}^{d-1}} \left\|(c^\top \Phi'(\langle \theta, x \rangle))W\langle v, x \rangle\right\|_{\psi_1} \\
&\le \left\|(c^\top \Phi'(\langle \theta, x \rangle))W\right\|_{\psi_2} \\
&\le \|c\|\,\|W\|_{\psi_2}\,,
\end{aligned}
$$

where the last inequality follows from the Cauchy-Schwartz inequality $\|c\|_1 \le \sqrt{N}\|c\|_2$ and the fact that $\Phi'(y) \in \{0, 1/\sqrt{N}\}^N$. Now denote the (correlated) Gaussian variables by $Z = \langle \theta, x \rangle$ and

$Z_* = \langle \theta^*, x \rangle$. Recalling Fact B.3 and our assumption that $\mathbb{E}[f_*(Z)] = 0$,

$$
\begin{aligned}
\|W\|_{\psi_2} &\leq \left\|c^\top \Phi(Z)\right\|_{\psi_2} + \|f_*(Z_*)\|_{\psi_2} + \|\xi\|_{\psi_2} \\
&\leq \left\|c^\top \Phi(Z) - \mathbb{E}[c^\top \Phi(Z)]\right\|_{\psi_2} + \left\|\mathbb{E}[c^\top \Phi(Z)]\right\|_{\psi_2} + \|f_*(Z_*)\|_{\psi_2} + \|\xi\|_{\psi_2} \\
&\lesssim \|c\| + |\mathbb{E}[c^\top \Phi(Z)]| + \mathrm{Lip}(f_*) + \sigma \\
&\leq \|c\|(1 + \|\mathbb{E}[\Phi(Z)]\|) + \mathrm{Lip}(f_*) + \sigma \ .
\end{aligned}
$$

By Corollary E.5, the following holds with probability at least $1 - \delta$ over the random features.

$$
\|\mathbb{E}[\Phi(Z)]\| \leq \mathbb{E}[\|\Phi(Z)\|] \lesssim \tau(1 + \sqrt{\log(1/\delta)/N}) \ .
$$

$\square$

We now prove the uniform convergence of $\nabla_\theta L_n(c, \theta)$. Let $\epsilon_\theta, \epsilon_c < 1$ be small positive integers. To make things concrete, we set $\epsilon_\theta = 1/(4n^2 N)$ and $\epsilon_c = r/n^2$. We consider $\epsilon$-nets of $\theta \in \mathbb{S}^{d-1}$ and $c \in \mathbb{R}^N$ with $\|c\| \leq r$. We denote these sets by $\mathcal{N}_\theta$ and $\mathcal{N}_c$ respectively. By Fact B.6, there exist such sets with $|\mathcal{N}_\theta| \leq (3/\epsilon_\theta)^d$ and $|\mathcal{N}_c| \leq (3r/\epsilon_c)^N$. We use these $\epsilon$-nets to decompose gradient error into three terms, which we bound individually. We abuse notation and denote by $\tilde{\theta}$ the element in $\mathcal{N}_\theta$ closest to $\theta$ in the $\ell_2$ norm and $\tilde{c}$ the closest element in $\mathcal{N}_c$ to $c$. Then, the gradient deviation term can be decomposed into

$$
\begin{aligned}
\sup_{\theta \in \mathbb{S}^{d-1}, \|c\| \leq r} \|\nabla_\theta L_n(c, \theta) - \nabla_\theta L(c, \theta)\| &\leq \sup_{\theta \in \mathbb{S}^{d-1}, \|c\| \leq r} \left\|\nabla_\theta L_n(c, \theta) - \nabla_\theta L_n(\tilde{c}, \tilde{\theta})\right\| \\
&+ \sup_{\tilde{\theta} \in \mathcal{N}_\theta, \tilde{c} \in \mathcal{N}_c} \left\|\nabla_\theta L_n(\tilde{c}, \tilde{\theta}) - \nabla_\theta L(\tilde{c}, \tilde{\theta})\right\| \\
&+ \sup_{\theta \in \mathbb{S}^{d-1}, \|c\| \leq r} \left\|\nabla_\theta L(c, \theta) - \nabla_\theta L(\tilde{c}, \tilde{\theta})\right\| .
\end{aligned}
$$

As mentioned previously, bounding the first term (with high probability) requires some care due to the non-differentiability of ReLU at the origin. The other two terms can be bounded using standard techniques. We first look at the sample gradient $\nabla_\theta \ell(c, \theta; x_i, y_i) := \nabla_\theta (y_i - c^\top \Phi(\langle x_i, \theta \rangle))^2$ to bound the "worst-case" discretization error incurred by discontinuities in the sample gradient. Then, we show that with high probability over the samples, for any $\theta \in \mathbb{S}^{d-1}$ and $c$, this "worst-case" discretization error occurs only for a few samples $x_i$. Thus, the contribution from the worst-case discretization error get averaged out in $\nabla_\theta L_n(c, \theta) - \nabla_\theta L_n(\tilde{c}, \tilde{\theta}) = \frac{1}{n} \sum_{i=1}^n (\nabla_\theta \ell(c, \theta; x_i, y_i) - \nabla_\theta \ell(\tilde{c}, \tilde{\theta}; x_i, y_i))$.

For simplicity, we set $\mathrm{ReLU}'(0) = 0$.[3] Note that $\Phi'(z) \in \{0, 1/\sqrt{N}\}^N$, so $\|\Phi'(z)\|_2 \leq 1$ for any $z \in \mathbb{R}$. By Corollary E.5, the sample gradient and an upper bound for its $\ell_2$ norm is given by

$$
\nabla_\theta \ell_i(c, \theta; x_i, y_i) = -(c^\top \Phi'(\langle x_i, \theta \rangle))(y_i - c^\top \Phi(\langle x_i, \theta \rangle))x_i \ .
$$

$$
\|\nabla_\theta \ell_i(c, \theta; x_i, y_i)\| \leq \|c\| \left(\mathrm{Lip}(f_*)\|x_i\| + \|c\|(\|x_i\| + C'\tau\sqrt{\log(1/\delta)})\right) \|x_i\| \lesssim Cr^2(\|x_i\|^2 + \|x_i\|) \ .
$$

When discretizing $\mathbb{S}^{d-1}$, the "best-case" is when $\Phi'(\langle x_i, \theta \rangle) = \Phi'(\langle x_i, \tilde{\theta} \rangle)$ in which case the discretization error is 1-Lipschitz in $\theta - \tilde{\theta}$. On the other hand, the discretization error is not Lipschitz when $\Phi'(\langle x_i, \theta \rangle) \neq \Phi'(\langle x_i, \tilde{\theta} \rangle)$, i.e., when the ReLU' sign patterns change after projecting $\theta$ onto $\mathcal{N}_\theta$. The "worst-case" discretization error corresponding to this case is upper bounded by

$$
\|\nabla_\theta \ell_i(c, \theta; x_i, y_i) - \nabla_\theta \ell_i(c, \tilde{\theta}; x_i, y_i)\| \leq \|\nabla_\theta \ell_i(c, \theta; x_i, y_i)\| + \|\nabla_\theta \ell_i(c, \tilde{\theta}; x_i, y_i)\| \lesssim Cr^2(\|x_i\|^2 + \|x_i\|) \ .
$$

We now show that for any fixed $\tilde{\theta} \in \mathbb{S}^{d-1}$, bottom-layer weights $b_1, \dots, b_N$ and signs $\varepsilon_1, \dots, \varepsilon_N$, with high probability only a small fraction of the sample gradients $\nabla_\theta \ell_i$ change ReLU' sign patterns $\Phi'(\langle x_i, \tilde{\theta} \rangle)$ within a ball of radius $\epsilon_\theta$ from $\tilde{\theta}$. As a consequence, the worst-case discretization error,

---

[3]The particular value of $\mathrm{ReLU}'(0)$ does not matter for our proof it is $[0, 1]$. In practice, however, the particular value assigned to $\mathrm{ReLU}'(0)$ may have non-trivial implications [14].

which comes from the non-differentiability of ReLU, does not accumulate too much for the averaged gradient $\nabla_\theta L_n(c, \theta) - \nabla_\theta L_n(c, \tilde{\theta})$. To this end, fix an arbitrary $\theta \in \mathbb{S}^{d-1}$ and let $Z_{\theta,i}$ be the indicator that projecting $\theta$ on to the $\epsilon$-net $\mathcal{N}_\theta$ changes the sign pattern for $x_i$. Formally,

$$Z_{\theta,i} = \mathbb{1}\left[ \exists \theta, j \in [N] \text{ s.t. } \|\tilde{\theta} - \theta\| \le \epsilon_\theta, \ \text{sign}(\varepsilon_j \langle \tilde{\theta}, x_i \rangle - b_j) \ne \text{sign}(\varepsilon_j \langle \theta, x_i \rangle - b_j) \right] .$$

By the union bound (over $j \in [N]$) and basic properties of Gaussian random variables,

$$\begin{aligned}
\mathbb{E}[Z_{\theta,i}] &\le N \, \mathbb{P}[\exists \theta \text{ s.t. } \|\tilde{\theta} - \theta\| \le \epsilon_\theta, \ \text{sign}(\varepsilon_1 \langle \tilde{\theta}, x_i \rangle - b_1) \ne \text{sign}(\varepsilon_1 \langle \theta, x_i \rangle - b_1)] \\
&\le N \, \mathbb{P}[|\langle \tilde{\theta}, x_i \rangle - b_1| \le \|x_i\| \, \epsilon_\theta] \\
&= N \, \mathbb{P}[|\langle \tilde{\theta}, x_i/\|x_i\| \rangle - b_1/\|x_i\| | \le \epsilon_\theta] \\
&\le N \, \mathbb{E}_t [\mathbb{P}[|\langle \tilde{\theta}, u \rangle - b_1/t| \le \epsilon_\theta |t|]] \quad \text{(denoting } t = \|x_i\| \text{ and } u = x_i/\|x_i\|) \\
&\le N \, \mathbb{P}[|\langle \tilde{\theta}, u \rangle| \le \epsilon_\theta] \, ,
\end{aligned}$$

where $u$ is a random vector drawn uniformly from $\mathbb{S}^{d-1}$, and we used the fact that $u$ and $r$ are independent, along with the fact that the density of $\langle \tilde{\theta}, u \rangle$ is peaked around 0. The last line can be upper bounded by the surface area of an $\epsilon$-thick strip around the equator in $\mathbb{S}^{d-1}$. This is given by the following anticoncentration bound from Section B.

**Lemma B.7** (Anticoncentration on the unit sphere). *Let $d \in \mathbb{N}$. Let $\theta \in \mathbb{S}^{d-1}$ be any fixed unit vector and let $u$ be a random vector drawn uniformly from $\mathbb{S}^{d-1}$. Then, for any $\epsilon > 0$,*

$$\mathbb{P}[\langle \theta, u \rangle \le \epsilon] \le 4\sqrt{d}\epsilon \, .$$

By Lemma B.7, if $\epsilon_\theta = 1/(4n^2 N)$, then $\mathbb{E}[Z_{\theta,i}] \le 1/n$ and thus $\sum_{i=1}^n \mathbb{E}[Z_{\theta,i}] \le 1$. By the independence of the samples $x_i$ and a Chernoff bound, it holds that the probability that at least $q \ge 6$ inputs $x_i$ satisfy the above event is upper bounded by $2^{-q}$ [63, Theorem 4.4]. As a result, with probability at least $1 - (3/\epsilon_\theta)^d \cdot 2^{-q}$, no more than $q$ inputs can change sign pattern for each fixed $\tilde{\theta}$. It suffices to set $q = 4d \log(1/(\epsilon_\theta \delta)) \lesssim d \log(nN/\delta)$.

We assume that this event holds and establish the first inequality with the help of the $\epsilon$-nets. We assume without loss of generality that for fixed $\tilde{\theta}$, the inputs that may change ReLU signs for any $\theta$ in an $\epsilon_\theta$-ball of $\tilde{\theta}$ are contained in $x_1, \ldots, x_q$. Recall that for any given $\theta$ and $c$, we denote by $\tilde{\theta}$ and $\tilde{c}$ the closest elements in the $\epsilon$-nets.

$$\begin{aligned}
&\left\| \nabla_\theta L_n(c, \theta) - \nabla_\theta L_n(\tilde{c}, \tilde{\theta}) \right\| \\
&\le \frac{1}{n} \sum_{i=1}^n \left\| \nabla_\theta \ell(c, \theta; x_i, y_i) - \nabla_\theta \ell(\tilde{c}, \tilde{\theta}; x_i, y_i) \right\| \\
&\le \frac{1}{n} \sum_{i=1}^n \left( \| \nabla_\theta \ell(c, \theta; x_i, y_i) - \nabla_\theta \ell(\tilde{c}, \theta; x_i, y_i) \| + \left\| \nabla_\theta \ell(\tilde{c}, \theta; x_i, y_i) - \nabla_\theta \ell(\tilde{c}, \tilde{\theta}; x_i, y_i) \right\| \right)
\end{aligned}$$

We first bound the terms involving differences between $c$ and $\tilde{c}$. Note that by Fact E.4, $\max_i |\xi_i| \lesssim \sigma \sqrt{\log(n/\delta)}$ and $\max_i \|x_i\| \lesssim \sqrt{d \log(n/\delta)}$ with probability at least $1 - \delta/6$. Thus, we consider events for which these conditions hold.

$$\begin{aligned}
&\| \nabla_\theta \ell(c, \theta; x, y) - \nabla_\theta \ell(\tilde{c}, \theta; x, y) \| \\
&= \left\| (c^\top \Phi'(\langle \theta, x \rangle))(c^\top \Phi(\langle \theta, x \rangle) - y)x - (\tilde{c}^\top \Phi'(\langle \theta, x \rangle))(\tilde{c}^\top \Phi(\langle \theta, x \rangle) - y)x \right\| \\
&\le \left\| (c^\top \Phi'(\langle \theta, x \rangle))((c - \tilde{c})^\top \Phi(\langle \theta, x \rangle))x \right\| + \left\| (c - \tilde{c})^\top \Phi'(\langle \theta, x \rangle)(\tilde{c}^\top \Phi(\langle \theta, x \rangle) - f_*(\langle x, \theta_* \rangle - \xi))x \right\| \\
&\le \|c\| \epsilon_c \|\Phi(\langle \theta, x \rangle)\| \|x\| + \epsilon_c \left( \|c\| \|\Phi(\langle \theta, x \rangle)\| + \text{Lip}(f_*)\|x\| + |\xi| \right) \|x\| \\
&\lesssim \epsilon_c \|x\| \left( \|c\|(\|x\| + C) + C\|x\| + |\xi| \right) \\
&\lesssim C r \epsilon_c \left( \|x\|^2 + \|x\|(1 + |\xi|/\sigma) \right) \\
&\lesssim C r \epsilon_c \left( \|x\|^2 + \|x\|\sqrt{\log(n/\delta)} \right) \, .
\end{aligned} \tag{42}$$

Next, we consider the first $q$ terms of the differences between $\theta$ and $\tilde{\theta}$. For some $i \in [\ell]$

$$\left\| \nabla_\theta \ell(\tilde{c}, \theta; x_i, y_i) - \nabla_\theta \ell(\tilde{c}, \tilde{\theta}; x_i, y_i) \right\|$$

$$\leq \left\| (\tilde{c}^\top \Phi'(\langle \theta, x_i \rangle))(\tilde{c}^\top \Phi(\langle \theta, x_i \rangle) - y_i)x_i - (\tilde{c}^\top \Phi'(\langle \tilde{\theta}, x_i \rangle))(\tilde{c}^\top \Phi(\langle \tilde{\theta}, x_i \rangle) - y_i)x_i \right\|$$

$$\leq 2 \sup_{\tilde{c},\theta} \left\| (\tilde{c}^\top \Phi'(\langle \theta, x_i \rangle))(\tilde{c}^\top \Phi(\langle \theta, x_i \rangle) - y_i)x_i \right\|$$

$$\leq 2 \sup_{\tilde{c},\theta} \|\tilde{c}\| \left( \|\tilde{c}\| \, \|\Phi(\langle \theta, x_i \rangle)\| + \mathrm{Lip}(f_*)\|x_i\| + |\xi_i| \right) \|x_i\|$$

$$\lesssim \sup_{\tilde{c}} \left( \|\tilde{c}^2\| \left( \|x_i\| + \tau(1 + \sqrt{\log(4/\delta)/N}) \right) + \|\tilde{c}\| \left( \mathrm{Lip}(f_*)\|x_i\| + |\xi_i| \right) \right) \|x_i\|$$

$$\lesssim r(r + \mathrm{Lip}(f_*))\|x_i\|^2 + \left( r^2 \tau(1 + \sqrt{\log(4/\delta)/N}) + r|\xi_i| \right) \|x_i\|$$

$$\lesssim Cr^2(\|x_i\|^2 + \|x_i\|(1 + |\xi_i|/\sigma))$$

$$\lesssim Cr^2(\|x_i\|^2 + \|x_i\|\sqrt{\log(n/\delta)}) . \tag{43}$$

Finally, we bound the remaining $n - q$ terms.

$$\left\| \nabla_\theta \ell(\tilde{c}, \theta; x_i, y_i) - \nabla_\theta \ell(\tilde{c}, \tilde{\theta}; x_i, y_i) \right\| = \left\| \tilde{c}^\top \Phi'(\langle \theta, x_i \rangle) \left( \tilde{c}^\top (\Phi(\langle \theta, x_i \rangle) - \Phi(\langle \tilde{\theta}, x_i \rangle)) \right) x_i \right\|$$

$$\leq r^2 \epsilon_\theta \|x_i\|^2 .$$

As a result, with probability at least $1 - \delta/3$,

$$\left\| \nabla_\theta L_n(c, \theta) - \nabla_\theta L_n(\tilde{c}, \tilde{\theta}) \right\| \lesssim Cr^2 \|x_i\| \left( \left( \frac{\epsilon_c}{r} + \frac{q}{n} \right) (\|x_i\| + \sqrt{\log(n/\delta)}) + \epsilon_\theta \|x_i\| \right)$$

$$\lesssim Cr^2 \cdot \frac{(d\log(n/\delta))^2}{n} .$$

where we set $q = 4d\log(1/(\epsilon_\theta \delta))$ and recall that $\epsilon_c = r/n^2$, $\epsilon_\theta = 1/(4n^2 N)$.

We now bound the second term $\left\| \nabla_\theta L_n(\tilde{c}, \tilde{\theta}) - \nabla_\theta L(\tilde{c}, \tilde{\theta}) \right\|$ using a standard concentration argument over the $(1/2)$-net on $\mathbb{S}^{d-1}$, which we denote by $\mathcal{N}_{1/2}$. By Fact B.6,

$$\sup_{\tilde{\theta} \in \mathcal{N}_\theta, \tilde{c} \in \mathcal{N}_c} \left\| \nabla_\theta L_n(\tilde{c}, \tilde{\theta}) - \nabla_\theta L(\tilde{c}, \tilde{\theta}) \right\| \leq 2 \sup_{\substack{v \in \mathcal{N}_{1/2}, \\ \tilde{\theta} \in \mathcal{N}_\theta, \tilde{c} \in \mathcal{N}_c}} \langle v, \nabla_\theta L_n(\tilde{c}, \tilde{\theta}) - \nabla_\theta L(\tilde{c}, \tilde{\theta}) \rangle. \tag{44}$$

By Lemma E.6, the $\psi_1$-norm of a sample gradient is bounded by $\|\nabla_\theta \ell(\tilde{c}, \tilde{\theta}; x_i, y_i)\|_{\psi_1} \lesssim Cr^2$, where we recall that $C = \max\{\mathrm{Lip}(f_*), \tau\sqrt{\log(1/\delta)}, \sigma, 1\}$. Since $\nabla_\theta L_n(\tilde{c}, \tilde{\theta}) = \frac{1}{n}\sum_i \nabla_\theta \ell(\tilde{c}, \tilde{\theta}; x_i, y_i)$ and $\nabla_\theta L(\tilde{c}, \tilde{\theta}) = \mathbb{E}[\nabla_\theta \ell(\tilde{c}, \tilde{\theta}; x, y)]$, and $v \in \mathbb{S}^{d-1}$, the right-hand side of equation (B.6) is the supremum of averaged sub-exponential random variables. Hence, the term can be bounded with by combining a union bound and Bernstein's inequality (Theorem B.4). Recall that $D = \max\{d, N\}$. For sufficiently large universal constants $C_0, C_1 > 0$, the following holds.

$$\mathbb{P}\left[ 2 \sup_{\tilde{\theta}, \tilde{c}, v} \langle v, \nabla_\theta L_n(\tilde{c}, \tilde{\theta}) - \nabla_\theta L(\tilde{c}, \tilde{\theta}) \rangle \geq C_0 \cdot Cr^2 \cdot \sqrt{\frac{D\log(nN/\delta)}{n}} \right]$$

$$\leq 6^d \left( \frac{3}{\epsilon_\theta} \right)^d \left( \frac{3r}{\epsilon_c} \right)^N \exp\left( -C_1 \cdot n \cdot \frac{D\log(nN/\delta)}{n} \right)$$

$$\leq \exp\left( d\log\left( \frac{18}{\epsilon_\theta} \right) + N\log\left( \frac{3r}{\epsilon_c} \right) - C_1 \cdot D\log(nN/\delta) \right)$$

$$\leq \frac{\delta}{3} ,$$

where we used the assumption $d \leq n$ in the last inequality.

We now bound third term. Note that this term involves only populational quantities, so discontinuity of the sample gradients is not an issue here.

$$\left\|\nabla_\theta L(c,\theta) - \nabla_\theta L(\tilde{c},\tilde{\theta})\right\| \leq \left\|\nabla_\theta L(c,\theta) - \nabla_\theta L(c,\tilde{\theta})\right\| + \left\|\nabla_\theta L(c,\tilde{\theta}) - \nabla_\theta L(\tilde{c},\tilde{\theta})\right\|$$

$$= \left\|\nabla_\theta L(c,\theta) - \nabla_\theta L(c,\tilde{\theta})\right\| + \left\|\mathbb{E}[\nabla_\theta \ell(c,\tilde{\theta};x,y) - \nabla_\theta \ell(\tilde{c},\tilde{\theta};x,y)]\right\| .$$

We upper bound the two terms individually. By Eq. (42), the second term is bounded as follows.

$$\left\|\mathbb{E}[\nabla_\theta \ell(c,\tilde{\theta};x,y) - \nabla_\theta \ell(\tilde{c},\tilde{\theta};x,y)]\right\| \lesssim Cr\epsilon_c \cdot d\sqrt{\log(n/\delta)} \lesssim Cr^2 \sqrt{\log(n/\delta)}/n .$$

For the first term, we use our regularity assumption on the target $f_*$ (Assumption 5.2), specifically the existence of continuous higher-order derivatives. Let $f_*(z) = \sum_{j=0}^\infty \alpha_j h_j(z)$ and $c^\top \Phi(z) = \sum_{j=0}^\infty \beta_j(c)h_j(z)$, be the Hermite expansion of the target and ReLU network, respectively. Define the univariate function $g : [-1,1] \to \mathbb{R}$ by

$$g(m;c) = \sum_{j=0}^\infty \alpha_j \beta_j(c)m^j .$$

Straightforward algebraic manipulation using orthonormality of the Hermite basis gives the following expression of the gradient with respect to $\theta \in \mathbb{S}^{d-1}$.

$$\nabla_\theta L(\theta) = g'(\langle\theta^*,\theta\rangle;c)\theta^* .$$

The following observation gives an upper bound on the Lipschitz constant of $g'(m;c)$, which depends only on $\|c\|$ and is thus constant if $r$ is fixed.

$$|g''(m;c)| \leq \sum_{j=2}^\infty j(j-1)|\alpha_j||\beta_j(c)| \leq \left(\sum_{j=1}^\infty j^2(j-1)^2\alpha_j^2\right)^{1/2} \left(\sum_{j=0}^\infty \beta_j(c)^2\right)^{1/2}$$

$$\leq \left(\sum_{j=1}^\infty j^2(j-1)^2\alpha_j^2\right)^{1/2} \|c^\top \Phi(z)\|_\gamma$$

$$\leq \left(\sum_{j=1}^\infty j^2(j-1)^2\alpha_j^2\right)^{1/2} Cr .$$

Lemma E.7 gives a simple expression for $\sum_j j^2(j-1)^2\alpha_j^2$. Its proof can be found in Section E.8.

**Lemma E.7.** *Let $f : \mathbb{R} \to \mathbb{R}$ be function such that its derivatives $f^{(1)}, \ldots, f^{(4)}$ are all in $L^2(\gamma)$. Let $f(z) = \sum_{j=0}^\infty \alpha_j h_j(z)$ be the Hermite expansion of $f$. Then,*

$$\sum_{j=1}^\infty j^2(j-1)^2\alpha_j^2 = \|f^{(4)}\|_\gamma^2 + 4\|f^{(3)}\|_\gamma^2 + 2\|f^{(2)}\|_\gamma^2 .$$

Let $C_{f_*} = \max\{\|f^{(1)}\|_\gamma, \ldots, \|f^{(4)}\|_\gamma, 1\}$, which is well-defined thanks to our regularity assumption on the target link function (Assumption 5.2). Then,

$$\left\|\nabla_\theta L(c,\theta) - \nabla_\theta L(c,\tilde{\theta})\right\| = |g'(\langle\theta^*,\theta\rangle) - g'(\langle\theta^*,\tilde{\theta}\rangle)| \leq \sup_{m\in[-1,1]} |g''(m)|\|\theta - \tilde{\theta}\| \lesssim C_{f_*}Cr\epsilon_\theta .$$

Putting everything together, we have that with probability at least $1 - \delta$,

$$\sup_{\theta,\|c\|\leq r} \|\nabla_\theta L_n(c,\theta) - \nabla_\theta L(c,\theta)\| \leq C_{f_*}Cr^2 \cdot \max\left(\sqrt{\frac{D\log(nN/\delta)}{n}}, \frac{(d\log(nN/\delta))^2}{n}\right) .$$

**Uniform convergence of $\nabla_c L_n$.** The proof is similar to the one for $\nabla_\theta L_n$.

**Lemma E.8** ($\nabla_c \ell$ is sub-exponential). *There exists a universal constant $C_0 > 0$ such that under the same assumptions as Lemma E.6, with probability at least $1 - \delta$ over the random features,*

$$\|\nabla_c \ell(c, \theta; x, f_*(\langle \theta^*, x \rangle) + \xi))\|_{\psi_1} \leq C^2 r \,,$$

*where we recall that $C = C_0 \cdot \max\{\mathrm{Lip}(f_*), \tau \sqrt{\log(1/\delta)}, \sigma, 1\}$.*

*Proof of Lemma E.8.* We first let

$$\nabla_c \ell(c, \theta; x, f_*(\langle x, \theta^* \rangle) + \xi) = 2\Phi(\langle x, \theta \rangle) \left( c^\top \Phi(\langle x, \theta \rangle) - f_*(\langle x, \theta^* \rangle) - \xi \right) := 2Y_2(Y_3 - Y_1) \,,$$

where $Y_1 = f_*(\langle x, \theta^* \rangle) + \xi$, $Y_2 = \Phi(\langle x, \theta \rangle)$ and $Y_3 = c^\top \Phi(\langle x, \theta \rangle)$. Following the proof of Lemma E.6, observe that those are sub-Gaussian random vectors, satisfying $\|Y_1\|_{\psi_2} \lesssim \mathrm{Lip}(f_*) + \sigma \lesssim C$, $\|Y_2\|_{\psi_2} \lesssim \tau \sqrt{\log(1/\delta)} \lesssim C$, and $\|Y_3\|_{\psi_2} \lesssim Cr$. Using again that the product of two sub-gaussian variables is sub-exponential and that the sum of two sub-exponential variables is sub-exponential (Fact B.3), we obtain the desired result. $\square$

Let $\epsilon_\theta = 1/(n^2)$ and $\epsilon_c = r/n^2$ and let $\mathcal{N}_\theta$ and $\mathcal{N}_c$ be $\epsilon_\theta$ and $\epsilon_c$-nets of $\mathbb{S}^{d-1}$ and $B_r^N$, respectively. Also, denote by $\tilde{\theta}$ and $\tilde{c}$ the closest element in $\mathcal{N}_\theta$ and $\mathcal{N}_c$ to $\theta \in \mathbb{S}^{d-1}$ and $c \in B_r^N$. Then,

$$\sup_{\theta \in \mathbb{S}^{d-1}, \|c\| \leq r} \|\nabla_c L_n(c, \theta) - \nabla_c L(c, \theta)\| \leq \sup_{\theta \in \mathbb{S}^{d-1}, \|c\| \leq r} \left\|\nabla_c L_n(c, \theta) - \nabla_c L_n(\tilde{c}, \tilde{\theta})\right\|$$

$$+ \sup_{\tilde{\theta} \in \mathcal{N}_\theta, \tilde{c} \in \mathcal{N}_c} \left\|\nabla_c L_n(\tilde{c}, \tilde{\theta}) - \nabla_c L(\tilde{c}, \tilde{\theta})\right\|$$

$$+ \sup_{\theta \in \mathbb{S}^{d-1}, \|c\| \leq r} \left\|\nabla_c L(c, \theta) - \nabla_c L(\tilde{c}, \tilde{\theta})\right\|.$$

We bound the first and third terms by bounding the discretization error for each samplewise gradient. First observe that

$$\|\nabla_c \ell(c, \theta; x, y) - \nabla_c \ell(\tilde{c}, \tilde{\theta}; x, y)\| \leq \|\nabla_c \ell(c, \theta; x, y) - \nabla_c \ell(\tilde{c}, \theta; x, y)\| + \|\nabla_c \ell(\tilde{c}, \theta; x, y) - \nabla_c \ell(\tilde{c}, \tilde{\theta}; x, y)\| \,.$$

We bound the first term on the RHS as follows.

$$\|\nabla_c \ell(c, \theta; x, y) - \nabla_c \ell(\tilde{c}, \theta; x, y)\| \leq \|\Phi(\langle \theta, x \rangle)\| \left|(\tilde{c} - c)^\top \Phi(\langle \theta, x \rangle)\right|$$

$$\leq \epsilon_c \|\Phi(\langle \theta, x \rangle)\|^2$$

$$\lesssim C\epsilon_c \|x\|^2 \,.$$

On the other hand,

$$\|\nabla_c \ell(\tilde{c}, \theta; x, y) - \nabla_c \ell(\tilde{c}, \tilde{\theta}; x, y)\| \leq \|\Phi(\langle \theta, x \rangle)\| |\tilde{c}^\top (\Phi(\langle \theta, x \rangle) - \Phi(\langle \tilde{\theta}, x \rangle))|$$

$$+ \|\Phi(\langle \theta, x \rangle) - \Phi(\langle \tilde{\theta}, x \rangle)\| |\tilde{c}^\top \Phi(\langle \tilde{\theta}, x \rangle))|$$

$$\lesssim C\epsilon_\theta \|\tilde{c}\| \|x\| \,.$$

Thus,

$$\|\nabla_c L(c, \theta) - \nabla_c L(\tilde{c}, \tilde{\theta})\| = \|\mathbb{E}[\nabla_c \ell(c, \theta; x, y) - \nabla_c \ell(\tilde{c}, \tilde{\theta}; x, y)]\|$$

$$\leq \mathbb{E}[\|\nabla_c \ell(c, \theta; x, y) - \nabla_c \ell(\tilde{c}, \tilde{\theta}; x, y)\|]$$

$$\lesssim C(\epsilon_c + \epsilon_\theta r)d \,.$$

$$\|\nabla_c L_n(c, \theta) - \nabla_c L_n(\tilde{c}, \theta)\| = \left\|\frac{1}{n} \sum_{i=1}^n \nabla_c \ell(c, \theta; x_i, y_i) - \nabla_c \ell(\tilde{c}, \tilde{\theta}; x_i, y_i)\right\|$$

$$= \frac{1}{n} \sum_{i=1}^n \left\|\nabla_c \ell(c, \theta; x_i, y_i) - \nabla_c \ell(\tilde{c}, \tilde{\theta}; x_i, y_i)\right\|$$

$$\lesssim C \cdot \epsilon_c \sup_i \|x_i\|^2 + \epsilon_\theta r \sup_i \|x_i\|$$

$$\lesssim C(\epsilon_c + \epsilon_\theta r) \cdot d \log(n/\delta) \,.$$

Taking $\epsilon_c = r/n^2$ and $\epsilon_\theta = 1/n^2$, we observe that with probability $1 - (2/3)\delta$,

$$\max \left\{ \left\| \nabla_c L_n(c, \theta) - \nabla_c L_n(\tilde{c}, \tilde{\theta}) \right\|, \left\| \nabla_c L(c, \theta) - \nabla_c L(\tilde{c}, \tilde{\theta}) \right\| \right\} \lesssim Cr \cdot \frac{\log(n/\delta)}{n} .$$

Note that the above upper bound is much smaller than $Cr\sqrt{d \log(n/\delta)/n}$. This indicates that most of the gradient deviation comes from the sub-exponential concentration of $\|\nabla_c L_n(\tilde{c}, \tilde{\theta}) - \nabla_c L(\tilde{c}, \tilde{\theta})\|$. The bound on this term is given by Bernstein's inequality (Theorem B.4) and the union bound over $\mathcal{N}_\theta \times \mathcal{N}_c$. Recall that $D = \max\{d, N\}$. Then,

$$\begin{aligned}
\mathbb{P} &\left[ \sup_{\tilde{\theta}, \tilde{c}} \left\| \nabla_c L_n(\tilde{c}, \tilde{\theta}) - \nabla_c L(\tilde{c}, \tilde{\theta}) \right\| \geq C^2 r \cdot \sqrt{\frac{D \log(n/\delta)}{n}} \right] \\
&\leq 6^d \left( \frac{3}{\epsilon_\theta} \right)^d \left( \frac{3r}{\epsilon_c} \right)^N \exp\left( -C_1 \cdot n \cdot \frac{D \log(n/\delta)}{n} \right) \\
&\leq \exp\left( d \log\left( \frac{18}{\epsilon_\theta} \right) + N \log\left( \frac{3r}{\epsilon_c} \right) - C_1 \cdot D \log(n/\delta) \right) \\
&\leq \frac{\delta}{3} .
\end{aligned}$$

□

### E.3 Proof of Lemma E.2

**Lemma E.2** (Local sharpness of the empirical landscape). *Let $d, n \in \mathbb{N}$ be such that $d \leq n$, let $\delta \in (0, 1/4)$, let $\tilde{s} \in \mathbb{N}$ be such that $\tilde{s} \geq s$, where $s \geq 1$ is the information exponent of $f_*$, let $\lambda \in (0, \lambda_{\tilde{s}}^*)$ where $\lambda_{\tilde{s}}^* \leq 1$ depends only on $\tilde{s}$, $f_*$, and $\tau$, and let $N \in \mathbb{N}$ be such that $N \geq \frac{C_0}{\lambda} \log \frac{1}{\lambda \delta}$, where $C_0 > 0$ is a universal constant. Furthermore, let $D = \max\{d, N\}$ and let $\epsilon \in (0, 1)$ be such that $\epsilon \leq \lambda^{-2} \sqrt{d/n}$. Then, there exists a universal constant $C_1 > 0$ such that for $C = C_1 \cdot \max\{\text{Lip}(f_*), \tau\sqrt{\log(1/\delta)}, \sigma\}$ and $\Delta = \max\left\{ \sqrt{\frac{D \log(n/\delta)}{n}}, \frac{(d \log(n/\delta))^2}{n} \right\}$ the following holds with probability at least $1 - \delta$ over the samples and random features.*

$$\Omega_n(\epsilon, \epsilon) = \Omega_n^{\text{bad}} \sqcup \Omega_n^{\text{good}} , \text{where} \tag{37}$$

$$\Omega_n^{\text{bad}} \subset \left\{ (c, \theta) \;\middle|\; |m| \leq \left( \frac{C_2 C^7}{\lambda^2} \cdot \Delta \right)^{\frac{1}{2\tilde{s}-1}} \wedge \|c\| \leq \frac{C^3}{\lambda} \right\},$$

$$\Omega_n^{\text{good}} \subset \left\{ (c, \theta) \;\middle|\; 1 - |m| \leq \frac{C_3 C^{14}}{\lambda^4} \cdot \Delta^2 \wedge \|c\| \leq \frac{C^3}{\lambda} \right\},$$

*and $C_2 = C_{f_*}/\tilde{s}\alpha_{\tilde{s}}^2$ and $C_3 = (2^{2\tilde{s}-1}C_2)^2$ in the above display. Moreover,*

$$\min_{(c, \theta) \in \Omega_n^{\text{bad}}} L_n(c, \theta) \geq \sigma^2 + \|f_*\|_\gamma^2 - 2 \max\{\|f_*\|_\gamma^2 C_2^2 C^{14}, C^8\} \cdot \Delta_{\text{crit}}/\lambda^2 , \tag{38}$$

*where*

$$\Delta_{\text{crit}} = \max \left\{ \sqrt{\frac{D \log(n/\delta)}{n}}, \left( \frac{d^2 \log^2(n/\delta)}{n} \right)^{\frac{2\tilde{s}}{2\tilde{s}-1}} \right\} . \tag{39}$$

The main idea behind the proof of Lemma E.2 is that the set of near-critical points of $L(c, \theta)$, which we denote here by $\Omega$ and define as the set of $(c, \theta)$ for which $\|\nabla L(c, \theta)\| \approx 0$, can be partitioned into two disjoint sets $\Omega^{\text{bad}}$ and $\Omega^{\text{good}}$, where $\Omega^{\text{bad}}$ is the set of points close to the equator (meaning $|\langle \theta^*, \theta \rangle| \approx 0$) and $\Omega^{\text{good}}$ is the set of points close to the poles (meaning $|\langle \theta^*, \theta \rangle| \approx 1$). Once this is established, the result follows from uniform convergence of the empirical landscape. To elaborate, with high probability over the samples and random features, any near-critical point of the empirical loss $L_n(c, \theta)$ is a near-critical point of the population loss $L(c, \theta)$ and vice versa by Lemma E.1. Thus, if $(c, \theta)$ is a near-critical point of the *empirical* loss $L_n(c, \theta)$, then it is a near-critical point of $L(c, \theta)$ as well, so topological properties of $L(c, \theta)$ dictate that either $|\langle \theta^*, \theta \rangle| \approx 0$ or $\approx 1$.

The key observation $\Omega = \Omega^{\mathrm{bad}} \cup \Omega^{\mathrm{good}}$ follows from Lemma E.9 and E.11. Lemma E.9 states that for near-critical points of the *projected* population loss $\bar{L}(\theta)$, a robust version of Theorem D.7 holds. That is, if $\|\nabla_\theta^{\mathbb{S}^{d-1}} \bar{L}(\theta)\| \approx 0$, then $|\langle \theta^*, \theta \rangle| \approx 0$ or $|\langle \theta^*, \theta \rangle| \approx 1$. Lemma E.11 states that if $(c, \theta)$ is a near-critical point of $L(c, \theta)$, then $\theta$ is a near-critical point of the projected population loss $\bar{L}(\theta)$. That is, if $\|\nabla_c L(c, \theta)\| \approx 0$ and $\|\nabla_\theta^{\mathbb{S}^{d-1}} L(c, \theta)\| \approx 0$, then $\|\nabla_\theta \bar{L}(\theta)\| \approx 0$. The formal proof is as follows.

*Proof of Lemma E.2.* Let $\epsilon \leq \lambda^{-2}\sqrt{d/n}$ and let $(c, \theta) \in \mathbb{R}^N \times \mathbb{S}^{d-1}$ be an $\epsilon$-approximate first-order critical point of the empirical loss $L_n$. We first show that with probability at least $1 - \delta$ over the samples and random features,

$$\|c\| \lesssim (\mathrm{Lip}(f_*) + \sigma)\tau \log(1/\delta)/\lambda \lesssim C^3/\lambda .$$

By definition, $L_n$ can be expressed as

$$L_n(c, \theta) = c^\top \hat{Q}_\lambda(\theta)c - 2\langle c, \hat{Y}(\theta)\rangle + C_{\mathrm{data}} , \tag{45}$$

where $C_{\mathrm{data}} > 0$ is some constant independent of $c$ and $\theta$, and

$$\hat{Q}_\lambda(\theta) = \frac{1}{n}\sum_{i=1}^n \Phi(\langle x_i, \theta\rangle)\Phi^\top(\langle x_i, \theta\rangle) + \lambda I , \quad \hat{Y}(\theta) = \frac{1}{n}\sum_{i=1}^n (f_*(\langle x_i, \theta^*\rangle) + \xi_i)\Phi(\langle x_i, \theta\rangle) .$$

Using the fact that $\nabla_c L_n(c, \theta) = \hat{Q}_\lambda(\theta)c - \hat{Y}(\theta), \|\nabla_c L_n(c, \theta)\| \leq \epsilon$, and $\hat{Q}_\lambda^{-1}(\theta) \preceq \lambda^{-1}I$ uniformly in $\theta$, we obtain

$$\|c\| \leq \|\hat{Q}_\lambda^{-1}(\theta)\| \left(\epsilon + \|\hat{Y}(\theta)\|\right)$$

$$\leq \lambda^{-1} \left(\epsilon + \|\hat{Y}(\theta)\|\right)$$

$$\leq \lambda^{-1} \left(\epsilon + \sup_{\theta \in \mathbb{S}^{d-1}} \|\hat{Y}(\theta)\|\right) .$$

Thus, it suffices to show that $\sup_\theta \|\hat{Y}(\theta)\|$ is upper bounded with high probability over the samples and random features. We achieve this using an $\epsilon$-net argument over $\mathbb{S}^{d-1}$ and Bernstein's inequality. Let $\mathcal{N}_{1/2}$ be a $1/2$-net of $\mathbb{S}^{d-1}$ and define the random vector $Y(\theta)$

$$Y(\theta) = (f_*(\langle x, \theta^*\rangle) + \xi)\Phi(\langle x, \theta\rangle) \in \mathbb{R}^N ,$$

where $x \sim \mathcal{N}(0, I_d)$ and $\xi \sim \mathcal{N}(0, \sigma^2)$.

For any fixed $\theta \in \mathbb{S}^{d-1}$, $Y(\theta)$ is subexponentially distributed with norm $\|Y(\theta)\|_{\psi_1} \lesssim (\mathrm{Lip}(f_*) + \sigma)\|\Phi(z)\|_{\psi_2} \lesssim (\mathrm{Lip}(f_*) + \sigma)\tau\sqrt{\log(1/\delta)}$ (see Corollary E.5). Hence, $\|\hat{Y}(\theta)\|$ concentrates around its expectation which is upper bounded as follows. Denote by $m = \langle \theta^*, \theta \rangle$ and let $z, z'$ be $m$-correlated Gaussian random variables. Then,

$$\mathbb{E}[\|Y(\theta)\|] = \mathbb{E}_{z,z'}[|f_*(z) + \xi|\|\Phi(z')\|]$$

$$\leq \mathbb{E}_{z,z'}[|f_*(z)|\|\Phi(z')\|] + \sigma \mathbb{E}_{z'}[\|\Phi(z')\|]$$

$$\lesssim \mathbb{E}_{z,z'}[\mathrm{Lip}(f_*)|z|(|z'| + \tau\sqrt{\log(1/\delta)})] + \sigma\tau\sqrt{\log(1/\delta)} \qquad \text{[Lemma E.5]}$$

$$\lesssim \mathrm{Lip}(f_*)(1 + \tau\sqrt{\log(1/\delta)}) + \sigma\tau\sqrt{\log(1/\delta)}$$

$$\lesssim (\mathrm{Lip}(f_*) + \sigma)\tau\sqrt{\log(1/\delta)} ,$$

where we used the fact that $\tau\sqrt{\log(1/\delta)} > 1$ in the last line.

By Bernstein's inequality (Theorem B.4) and the union bound over $\mathcal{N}_{1/2}$, the following holds with probability at least $1 - \delta$.

$$\sup_{\theta \in \mathbb{S}^{d-1}} \|\hat{Y}(\theta)\| \leq 2 \sup_{\tilde{\theta} \in \mathcal{N}_{1/2}} \|\hat{Y}(\tilde{\theta})\| \lesssim (\mathrm{Lip}(f_*) + \sigma)\tau\sqrt{\log(1/\delta)}(1 + \sqrt{d\log(1/\delta)/n})$$

$$\lesssim (\mathrm{Lip}(f_*) + \sigma)\tau \log(1/\delta) ,$$

where we used the assumption $d \leq n$ for the last inequality.

Thus, we may set $r \lesssim (\mathrm{Lip}(f_*) + \sigma)\tau \log(1/\delta)/\lambda \lesssim C^3/\lambda$ for the upper bound on $\|c\|$ in Lemma E.1. Recalling the notation $C = \max\{\mathrm{Lip}(f_*), \sigma, \tau\sqrt{\log(1/\delta)}\}$ and using the assumption $\epsilon \leq \lambda^{-1}\sqrt{D/n}$, we have

$$
\begin{aligned}
\|\nabla_c L(c,\theta)\| &\leq \|\nabla_c L(c,\theta) - \nabla_c L_n(c,\theta)\| + \|\nabla_c L_n(c,\theta)\| \\
&\lesssim C^2 r \sqrt{\frac{D\log(n/\delta)}{n}} + \epsilon \\
&\lesssim \frac{C^5}{\lambda}\sqrt{\frac{D\log(n/\delta)}{n}} \ .
\end{aligned}
\tag{46}
$$

and

$$
\begin{aligned}
\|\nabla_\theta^{\mathbb{S}^{d-1}} L(c,\theta)\| &\leq \|\nabla_\theta^{\mathbb{S}^{d-1}} L(c,\theta) - \nabla_\theta^{\mathbb{S}^{d-1}} L_n(c,\theta)\| + \|\nabla_\theta^{\mathbb{S}^{d-1}} L_n(c,\theta)\| \\
&\leq \|\nabla_\theta L(c,\theta) - \nabla_\theta L_n(c,\theta)\| + \|\nabla_\theta^{\mathbb{S}^{d-1}} L_n(c,\theta)\| \\
&\lesssim C_{f_*} C r^2 \cdot \max\left\{\sqrt{\frac{D\log(n/\delta)}{n}}, \frac{(d\log(n/\delta))^2}{n}\right\} + \epsilon \\
&\lesssim \frac{C_{f_*} C^7}{\lambda^2} \cdot \max\left\{\sqrt{\frac{D\log(n/\delta)}{n}}, \frac{(d\log(n/\delta))^2}{n}\right\} ,
\end{aligned}
\tag{47}
$$

where $C_{f_*} = \max\{\|f^{(1)}\|_\gamma, \ldots, \|f^{(4)}\|_\gamma\}$.

Let $\tilde{\epsilon}_c = C_1(C^5/\lambda)\sqrt{D\log(n/\delta)/n}$ and $\tilde{\epsilon}_\theta = C_1(C_{f_*} C^7/\lambda^2) \cdot \max\{\sqrt{D\log(n/\delta)/n}, d^2\log^2(n/\delta)/n\}$, where $C_1 > 0$ is an appropriately chosen universal constant for Eq. (46) and (47). Then, the above inequalities can be expressed as

$$
\|\nabla_c L(c,\theta)\| \leq \tilde{\epsilon}_c \quad \text{and} \quad \|\nabla_\theta^{\mathbb{S}^{d-1}} L(c,\theta)\| \leq \tilde{\epsilon}_\theta \ .
$$

Since we assumed $N \geq \frac{C}{\lambda}\log\frac{1}{\lambda\delta}$, the conditions of Lemma E.11 are satisfied. Hence,

$$
\|\nabla_\theta^{\mathbb{S}^{d-1}} \bar{L}(\theta)\| \lesssim C_{f_*}(\tau/\lambda)\tilde{\epsilon}_c + \epsilon_\theta \lesssim \frac{C_{f_*} C^7}{\lambda^2} \cdot \max\left\{\sqrt{\frac{D\log(n/\delta)}{n}}, \frac{(d\log(n/\delta))^2}{n}\right\} \ .
\tag{48}
$$

As a result, by Lemma E.9, which applies since $\lambda < \lambda^*$ and $N \geq \frac{C}{\lambda}\log\frac{1}{\lambda\delta}$, either one of the following must be true.

$$
|m| \lesssim \left(\frac{C_{f_*} C^7}{\lambda^2 \tilde{s}\alpha_{\tilde{s}}^2} \cdot \max\left\{\sqrt{\frac{D\log(n/\delta)}{n}}, \frac{(d\log(n/\delta))^2}{n}\right\}\right)^{\frac{1}{2\tilde{s}-1}} , \text{ or}
$$

$$
1 - |m| \lesssim \left(\frac{2^{2\tilde{s}-1}}{\tilde{s}\alpha_{\tilde{s}}^2}\right)^2 \frac{C_{f_*}^2 C^{14}}{\lambda^4} \cdot \max\left\{\sqrt{\frac{D\log(n/\delta)}{n}}, \frac{(d\log(n/\delta))^2}{n}\right\}^2 ,
$$

where we recall from Lemma E.9 that $s$ is the information exponent of $f_*$ and $\tilde{s} \in \mathbb{N}$ is any number satisfying $\tilde{s} \geq s$. Hence, Eq. (37) is established.

Let us now prove (38). By Lemma E.13, for any $(c,\theta) \in \Omega_n^{\mathrm{bad}}$,

$$
|L_n(c,\theta) - L(c,\theta)| \lesssim \frac{C^8}{\lambda^2} \cdot \sqrt{\frac{D\log(n/\delta)}{n}} \ .
$$

Denoting $\Delta = \max\{\sqrt{\frac{D \log(n/\delta)}{n}}, \frac{d^2 \log^2(n/\delta)}{n}\}$, and using the fact that $C > 1$ and $\frac{2\tilde{s}}{2\tilde{s}-1} \leq 2$,

$$\min_{\Omega_n^{\mathrm{bad}}} L_n(c,\theta) \geq \min_{\Omega_n^{\mathrm{bad}}} L(c,\theta) - \frac{C^8}{\lambda^2} \cdot \sqrt{\frac{D \log(n/\delta)}{n}}$$

$$\geq \sigma^2 + \|f_*\|_\gamma^2 \cdot \min_{m \in \Omega_n^{\mathrm{bad}}} (1 - m^{2\tilde{s}}) - \frac{C^8}{\lambda^2} \cdot \sqrt{\frac{D \log(n/\delta)}{n}}$$

$$\geq \sigma^2 + \|f_*\|_\gamma^2 \left(1 - \left(\frac{C_2 C^7 \Delta}{\lambda^2}\right)^{\frac{2\tilde{s}}{2\tilde{s}-1}}\right) - \frac{C^8}{\lambda^2} \cdot \sqrt{\frac{D \log(n/\delta)}{n}}$$

$$\geq \sigma^2 + \|f_*\|_\gamma^2 - 2 \max\{\|f_*\|_\gamma^2 C_2^2 C^{14}, C^8\} \cdot \frac{\Delta_{\mathrm{crit}}}{\lambda^2}, \tag{49}$$

where $C_2 = C_{f_*}/(\tilde{s}\alpha_{\tilde{s}}^2)$ and

$$\Delta_{\mathrm{crit}} = \max\left\{\sqrt{\frac{D \log(n/\delta)}{n}}, \left(\frac{d^2 \log^2(n/\delta)}{n}\right)^{\frac{2\tilde{s}}{2\tilde{s}-1}}\right\}.$$

This concludes the proof of the lemma. $\qquad\square$

**Lemma E.9** (Near-criticality of $\bar{L}(\theta)$). *Let $\delta \in (0, 1/4)$, $\tau > 1$, $\beta = \frac{1-1/\tau^2}{3+1/\tau^2}$, $\lambda \in (0,1)$, and let $C_{f_*} = \max\{\|f_*\|_\gamma, \|f_*'\|_\gamma, \|f_*''\|_\gamma\}$. There exist $C_1 > 0$ depending only on $f_*$ and $\tau$, and a universal constant $C_2 > 0$ such that for any $\tilde{s} \geq s$, where $s \geq 1$ is the information exponent of $f_*$, if*

$$C_1 \lambda^{\beta/2} < \min\{\tilde{s}\alpha_{\tilde{s}}^2, C_{f_*}^2/\tilde{s}\} \quad \text{and} \quad N \geq \frac{C_2}{\lambda} \log \frac{1}{\lambda\delta},$$

*then with probability at least $1-\delta$, the following holds for the projected loss $\bar{L}$. If $\|\nabla_\theta^{\mathbb{S}^{d-1}} \bar{L}(\theta)\| \leq \epsilon$, then $m = \langle \theta^*, \theta \rangle$ satisfies either*

$$|m| \leq \left(\frac{2\epsilon}{\tilde{s}\alpha_{\tilde{s}}^2}\right)^{\frac{1}{2\tilde{s}-1}} \quad \text{or} \quad 1 - |m| \leq \left(\frac{2^{2\tilde{s}-1}}{\tilde{s}\alpha_{\tilde{s}}^2}\right)^2 \cdot \epsilon^2.$$

**Remark E.10** (Choice of $\tilde{s}$). *Lemma E.9 gives us freedom over the choice of $\tilde{s}$ provided $\tilde{s} \geq s$, where $s$ is the information exponent of the target link function $f_*$. If we fix $\tilde{s} = s$, then the upper bound on $C_1 \lambda^{\beta/2}$ depends only on $f_*$ and $\tau$. Indeed, the choice $\tilde{s} = s$ implies the tightest upper bound on $|m|$ since its exponent is $1/(2\tilde{s} - 1)$. Yet, extra freedom over the choice of $\tilde{s}$ allows us to apply Lemma E.9 to more general settings. For example, when the target, which we denote by $f_n$, potentially changes with respect to $n$, but converges to some limit $f_*$ in $L^2(\gamma)$ as $n \to \infty$. In this case, the information exponent $s_n$ of $f_n$ is not necessarily the same as that of $f_*$ nor do the Hermite coefficients match exactly. However, we can ensure that if $f_n$ is sufficiently close to $f_*$, then $|\alpha_s(f_n) - \alpha_s(f_*)| \geq |\alpha_s(f_*)|/2$ and thus apply Lemma E.9 to $f_n$ using quantities related to $f_*$.*

*Proof of Lemma E.9.* We use the representation of the restricted population loss $\bar{L}(\theta)$ from Lemma D.7 Eq. (34), the notation $\rho_m = 2\langle \hat{P}_\lambda g_m, \bar{g}_m \rangle$, and the definition of the Riemannian gradient to obtain

$$\nabla_\theta^{\mathbb{S}^{d-1}} \bar{L}(\theta) = \nabla_\theta \bar{L}(\theta) - \langle \nabla_\theta \bar{L}(\theta), \theta \rangle \theta = -\rho_m(\theta^* - m\theta). \tag{50}$$

This yields an exact representation of the magnitude of the Riemannian gradient $\|\nabla_\theta^{\mathbb{S}^{d-1}} \bar{L}(\theta)\|^2 = \rho_m^2(1 - m^2)$ that depends only on $m \in [-1, 1]$. Following the proof of Theorem 5.3 for upper

bounding $\left|\langle(I - \hat{P}_\lambda)g_m, \bar{g}_m\rangle_\gamma\right|$, we observe that

$$
\begin{aligned}
\left|\langle(I - \hat{P}_\lambda)g_m, \bar{g}_m\rangle_\gamma\right| &\leq \|(I - \hat{P}_\lambda)g_m\|_\gamma \|\bar{g}_m\|_\gamma \\
&\leq 2\sqrt{A(g_m, \lambda)}\|\bar{g}_m\|_\gamma && \text{[Lemma C.2]} \\
&\leq 2\sqrt{C(\tau^{1+\beta}\tilde{K}^2\|g_m''\|_\gamma^2\lambda^\beta + \lambda C_{g_m}^2)}\|\bar{g}_m\|_\gamma && \text{[Lemma D.2]} \\
&\leq 2\lambda^{\beta/2}C_{g_m}\sqrt{2C\tau^{1+\beta}}\tilde{K}\|\bar{g}_m\|_\gamma \\
&\leq 4\lambda^{\beta/2}\sqrt{C\tau^{1+\beta}}\tilde{K} \cdot \sum_{j=s}^\infty j^2\alpha_j^2 m^{2j-1} && \text{[Corollary D.9]} \\
&\leq C'\lambda^{\beta/2} \cdot \sum_{j=s}^\infty j^2\alpha_j^2 m^{2j-1} \,, && (51)
\end{aligned}
$$

where $C' = 4\sqrt{C\tau^{1+\beta}}\tilde{K}$. Denoting $C_{f_*} = \max\{\|f_*\|_\gamma, \|f_*'\|_\gamma, \|f_*''\|_\gamma\}$, we observe that

$$
\sum_{j=\tilde{s}}^\infty j^2\alpha_j^2 m^{2j-1} \leq |m|^{2\tilde{s}-1}\sum_{j=\tilde{s}}^\infty j^2\alpha_j^2 = |m|^{2\tilde{s}-1}\left(\|f_*''\|_\gamma^2 + \|f_*''\|_\gamma^2\right) \leq 2C_{f_*}^2|m|^{2\tilde{s}-1} \,,
$$

Thus, for $\lambda$ satisfying $2C'C_{f_*}^2\lambda^{\beta/2} \leq \min\{\tilde{s}\alpha_{\tilde{s}}^2, C_{f_*}^2/\tilde{s}\}$, we have

$$
\begin{aligned}
|\rho_m| &\geq 2\left|\langle g_m, \bar{g}_m\rangle_\gamma\right| - 2|\langle(I - \hat{P}_\lambda)g_m, \bar{g}_m\rangle_\gamma| \\
&\geq 2\sum_{j=s}^\infty j\alpha_j^2|m|^{2j-1} - C'\lambda^{\beta/2}\left(\sum_{j=s}^{\tilde{s}} j^2\alpha_j^2|m|^{2j-1} + \sum_{j=\tilde{s}}^\infty j^2\alpha_j^2|m|^{2j-1}\right) \\
&\geq 2\sum_{j=s}^\infty j\alpha_j^2|m|^{2j-1} - \sum_{j=s}^{\tilde{s}} j\alpha_j^2|m|^{2j-1} - C'\lambda^{\beta/2}\sum_{j=\tilde{s}}^\infty j^2\alpha_j^2|m|^{2j-1} \\
&\geq 2\sum_{j=\tilde{s}}^\infty j\alpha_j^2|m|^{2j-1} - 2C'C_{f_*}^2\lambda^{\beta/2}|m|^{2\tilde{s}-1} \\
&\geq \tilde{s}\alpha_{\tilde{s}}^2|m|^{\tilde{s}-1} \,.
\end{aligned}
$$

We now assume that $\|\nabla_\theta^{\mathbb{S}^{d-1}}\bar{L}(\theta)\| = |\rho_m|\sqrt{1-m^2} \leq \epsilon$ and retrieve the claimed bounds on $|m|$. If $|m| \leq 1/2$, then $|\rho_m| \leq \sqrt{4/3}\epsilon$. Hence, our lower bound on $|\rho_m|$ implies that

$$
|m| \leq \left(\frac{2\epsilon}{\tilde{s}\alpha_{\tilde{s}}^2}\right)^{\frac{1}{2\tilde{s}-1}} \,.
$$

If $|m| > 1/2$, then $|\rho_m| \geq (1/2)^{2\tilde{s}-1}\tilde{s}\alpha_{\tilde{s}}^2$, thus

$$
1 - |m| \leq \frac{\epsilon^2}{|\rho_m|^2(1+|m|)} \leq \left(\frac{2^{2\tilde{s}-1}}{\tilde{s}\alpha_{\tilde{s}}^2}\right)^2 \cdot \epsilon^2 \,.
$$

$\square$

**Lemma E.11** (Near-criticality of $L$ and $\bar{L}$)**.** *There exists a universal constant $C > 0$ such that for any $\delta \in (0,1)$ and $N \in \mathbb{N}$ satisfying $N \geq C\log(1/\delta)$, the following holds with probability at least $1 - \delta$ over the random biases. For any $(c, \theta) \in \mathbb{R}^N \times \mathbb{S}^{d-1}$ such that $\|\nabla_\theta^{\mathbb{S}^{d-1}}L(c,\theta)\| \leq \epsilon_\theta$ and $\|\nabla_c L(c,\theta)\| \leq \epsilon_c$, it holds*

$$
\left\|\nabla_\theta^{\mathbb{S}^{d-1}}\bar{L}(\theta)\right\| \leq \frac{2m^{s-1}\sqrt{1-m^2}\tau\epsilon_c}{\lambda}\sqrt{\|f_*''\|_\gamma^2 + \|f_*'\|_\gamma^2} + \epsilon_\theta \lesssim C_{f_*}(\tau/\lambda)\epsilon_c + \epsilon_\theta \,,
$$

*where $C_{f_*} = \max\{\|f_*'\|_\gamma, \|f_*''\|_\gamma\}$.*

*Proof of Lemma E.11.* Recall from Corollary D.5 that

$$\nabla_c L(c,\theta) = 2(Q_\lambda c - \sum_j \alpha_j m^j \mathcal{T}_j) \text{ and } \nabla_\theta L(c,\theta) = -\langle c, \sum_j j\alpha_j m^{j-1} \mathcal{T}_j \rangle \theta^* .$$

Define $c_\theta = \arg\min_c L(c,\theta)$, so that $\bar{L}(\theta) = L(c_\theta, \theta)$. We first show that if $\|\nabla_c L(c,\theta)\| \leq \epsilon_c$, then $c$ and $c_\theta$ are nearby. Because $\nabla_c L(c_\theta, \theta) = 0$ and $Q_\lambda \succeq \lambda I_N$,

$$\|\nabla_c L(c,\theta)\| = \|\nabla_c L(c,\theta) - \nabla_c L(c_\theta, \theta)\| = 2\|Q_\lambda(c - c_\theta)\| \geq 2\lambda \|c - c_\theta\|.$$

Thus, $\|c - c_\theta\| \leq \frac{\epsilon_c}{2\lambda}$. We now recall the Riemannian gradient for $\theta$,

$$\nabla_\theta^{\mathbb{S}^{d-1}} L(c,\theta) = -\langle c, \sum_j j\alpha_j m^{j-1} \mathcal{T}_j \rangle \theta^* + \langle c, \sum_j j\alpha_j m^j \mathcal{T}_j \rangle \theta,$$

and use it to bound the norm of the projected gradient.

$$
\begin{aligned}
\left\| \nabla_\theta^{\mathbb{S}^{d-1}} \bar{L}(\theta) \right\| &\leq \left\| \nabla_\theta^{\mathbb{S}^{d-1}} L(c,\theta) \right\| + \left\| \nabla_\theta^{\mathbb{S}^{d-1}} L(c_\theta, \theta) - \nabla_\theta^{\mathbb{S}^{d-1}} L(c,\theta) \right\| \\
&\leq \epsilon_\theta + \left\| -\langle c_\theta - c, \sum_j j\alpha_j m^{j-1} \mathcal{T}_j \rangle \theta^* + \langle c_\theta - c, \sum_j j\alpha_j m^j \mathcal{T}_j \rangle \theta \right\| \\
&= \epsilon_\theta + \sqrt{1 - m^2} \left| \langle c_\theta - c, \sum_j j\alpha_j m^{j-1} \mathcal{T}_j \rangle \right| \\
&\leq \epsilon_\theta + \sqrt{1 - m^2} \|c_\theta - c\| \left\| \sum_j j\alpha_j m^{j-1} \mathcal{T}_j \right\| \\
&\leq \epsilon_\theta + \frac{\epsilon_c}{2\lambda} \sqrt{1 - m^2} \left\| \sum_j j\alpha_j m^{j-1} \mathcal{T}_j \right\| .
\end{aligned}
$$

We conclude by employing Lemmas D.8 and E.12 to obtain a bound on the final term that holds with probability at least $1 - \delta$.

$$
\begin{aligned}
\left\| \sum_j j\alpha_j m^{j-1} \mathcal{T}_j \right\|^2 &= \left\| \mathcal{T} \sum_j j\alpha_j m^{j-1} h_j \right\|^2 \\
&= \left\| \hat{\Sigma}^{1/2} \sum_j j\alpha_j m^{j-1} h_j \right\|^2 \leq \|\hat{\Sigma}\|_{\mathrm{op}} \left\| \sum_j j\alpha_j m^{j-1} h_j \right\|_\gamma^2 \\
&\leq \mathrm{Tr}(\hat{\Sigma}) \|\bar{g}_m\|_\gamma^2 \leq \left( \frac{1}{2} + \tau^2 \right) (\|f_*''\|_\gamma^2 + \|f_*'\|_\gamma^2) m^{2(s-1)} .
\end{aligned}
$$

$\square$

**Lemma E.12.** *There exists a universal constant $C > 0$ such that for any $\tau > 1$ and $\delta \in (0,1)$, if $N \geq C \log(1/\delta)$, then $\|\hat{\Sigma}\|_{\mathrm{op}} \leq \mathrm{Tr}(\hat{\Sigma}) \leq \tau^2 + 1/2$ with probability at least $1 - \delta$.*

*Proof of Lemma E.12.* By definition of $\hat{\Sigma}$, $\mathrm{Tr}(\hat{\Sigma}) = \frac{1}{N} \sum_{i=1}^N \left\| \phi_{b_i}^{\varepsilon_i} \right\|_\gamma^2$, with $\phi_b^\varepsilon(u) = \phi(\varepsilon u - b)$. We compute the expectation of $\|\phi_b^\varepsilon\|_\gamma^2$ for $b \sim \gamma_\tau$ and $\varepsilon \sim$ Rad, show that it is sub-exponential, and conclude that $\mathrm{Tr}(\hat{\Sigma})$ concentrates around its expectation. The computation of the expectation depends on elementary properties of the Gaussian distribution.

$$\mathop{\mathbb{E}}_{\substack{\varepsilon \sim \mathrm{Rad} \\ b \sim \gamma_\tau}} \left[ \|\phi_b^\varepsilon\|_\gamma^2 \right] = \frac{1}{2} \mathop{\mathbb{E}}_{\substack{b \sim \gamma_\tau \\ z \sim \gamma}} \left[ \phi(z - b)^2 \right] + \frac{1}{2} \mathop{\mathbb{E}}_{\substack{b \sim \gamma_\tau \\ z \sim \gamma}} \left[ \phi(-z - b)^2 \right] = \mathop{\mathbb{E}}_{u \sim \mathcal{N}(0, 1+\tau^2)} \left[ \phi(u)^2 \right] = \frac{1}{2}(1 + \tau^2).$$

Note that $\left\| \phi_b^1 \right\|_\gamma$ and $\left\| \phi_b^{-1} \right\|$ are $C_1 \tau^2$-subgaussian random variables for some constant $C_1$ because $b$ is $\tau^2$-subgaussian, and $\|\phi_b^\varepsilon\|_\gamma$ for fixed $\varepsilon$ is a 1-Lipschitz function of $b$: $\left| \|\phi_b^\varepsilon\|_\gamma - \|\phi_{b'}^\varepsilon\|_\gamma \right| \leq$

$\left\| \phi_b^\varepsilon - \phi_{b'}^\varepsilon \right\|_\gamma \leq |b - b'|$. Thus, $\left\| \phi_b^\varepsilon \right\|_\gamma = \frac{1+\varepsilon}{2} \left\| \phi_b^1 \right\|_\gamma + \frac{1-\varepsilon}{2} \left\| \phi_b^{-1} \right\|_\gamma$ is $C'\tau^2$-subgaussian by Fact B.3. As a result, $\left\| \phi_b^\varepsilon \right\|_\gamma^2$ is $C'\tau^2$-subexponential, and $\mathrm{Tr}(\hat{\Sigma})$ is $\frac{C\tau^2}{N}$-subexponential. Then,

$$\mathbb{P}\left[ \mathrm{Tr}(\hat{\Sigma}) \geq \mathbb{E}\left[ \mathrm{Tr}(\hat{\Sigma}) \right] + \frac{1}{2}\tau^2 \right] \leq \exp\left( -\frac{\tau^2/2}{C_2 \cdot \tau^2/N} \right) = \exp\left( -\frac{N}{C_2} \right).$$

We conclude by selecting a sufficiently large $N$. $\qquad\square$

Moreover, the empirical loss uniformly concentrated for $(c, \theta) \in B_N(r) \times \mathbb{S}^{d-1}$, as quantified in the following lemma, following the same strategy as our previous gradient concentration:

**Lemma E.13** (Uniform convergence of empirical loss). *Let $d, n, N \in \mathbb{N}$ be such $d \leq n$, let $D = \max\{d, N\}$, let $\delta \in (0, 1/4)$, $r \geq 1$, and let $\sigma^2 > 0$ $\tau^2 > 1$. Then, there exists a universal constant $C_0 > 0$ such that with probability at least $1 - \delta$ over samples and random features,*

$$\sup_{\theta \in \mathbb{S}^{d-1}, \|c\| \leq r} |L_n(c, \theta) - L(c, \theta)| \leq C^2 r^2 \sqrt{\frac{D \log(n/\delta)}{n}} \ ,$$

*where $C = C_0 \cdot \max\{\mathrm{Lip}(f_*), \tau\sqrt{\log(1/\delta)}, \sigma\}$.*

*Proof of Lemma E.13.* We use the same $\epsilon$-net proof as that of Lemma E.1 to prove that this bound holds. As before, we first bound the sub-exponential norm of $\ell(c, \theta; x, y) = (c^\top \Phi(\langle x, \theta \rangle) - y)^2$.

**Lemma E.14.** *Let $f_* : \mathbb{R} \to \mathbb{R}$ be a Lipschitz function, let $\delta \in (0, 1/4)$, let $r \geq 1$, and let $\tau^2 > 1$. Then there exists a universal constant $C' > 0$ such that the following holds with probability at least $1 - \delta$ over the random features.*

$$\left\| \ell(c, \theta; x, f_*(\langle x, \theta^* \rangle) + \xi) \right\|_{\psi_1} \leq C'C^2 r^2 \ ,$$

*where $C = \max\{\mathrm{Lip}(f_*), \tau\sqrt{\log(1/\delta)}, \sigma\}$.*

*Proof of Lemma E.14.* By Fact B.3, it suffices to bound $\left\| c^\top \Phi(\langle x, \theta \rangle) - f_*(\langle x, \theta_* \rangle) - \xi \right\|_{\psi_2}$. Note that this quantity identically equals $\|W\|_{\psi_2}$ for the random variable $W$ defined in the proof of Lemma E.6. Thus, with probability at least $1 - \delta$,

$$\|W\|_{\psi_2} \lesssim r\tau \left( 1 + \sqrt{\log(1/\delta)/N} \right) + \mathrm{Lip}(f_*) + \sigma \lesssim Cr \ .$$

$\qquad\square$

We consider two $\epsilon$-nets $\mathcal{N}_\theta$ and $\mathcal{N}_c$ of radii $\epsilon_\theta$ and $\epsilon_c$ covering $\mathbb{S}^{d-1}$ and $B_r^N$ respectively. We again denote by $\tilde{\theta}$ and $\tilde{c}$ the closest elements in the nets to $\theta$ and $c$. Then,

$$\sup_{\theta, c} |L_n(c, \theta) - L(c, \theta)| \leq \sup_{\theta, c} \left| L_n(c, \theta) - L_n(\tilde{c}, \tilde{\theta}) \right| + \sup_{\tilde{\theta}, \tilde{c}} \left| L_n(\tilde{c}, \tilde{\theta}) - L(\tilde{c}, \tilde{\theta}) \right| + \sup_{\theta, c} \left| L(c, \theta) - L(\tilde{c}, \tilde{\theta}) \right|.$$

We bound the first and last terms by considering the discretization error of samplewise loss.

$$\left| \ell(c, \theta; x, y) - \ell(\tilde{c}, \tilde{\theta}; x, y) \right| = \left| \left( \tilde{c}^\top \Phi(\langle x, \tilde{\theta} \rangle) - y \right)^2 - \left( c^\top \Phi(\langle x, \theta \rangle) - y \right)^2 \right|$$

$$= \left| \tilde{c}^\top \Phi(\langle x, \tilde{\theta} \rangle) - c^\top \Phi(\langle x, \theta \rangle) \right| \left| \tilde{c}^\top \Phi(\langle x, \tilde{\theta} \rangle) + c^\top \Phi(\langle x, \theta \rangle) - 2y \right|.$$

We bound the first factor, relying on the event of Corollary E.5 with probability at least $1 - (\delta/6)$.

$$\left| \tilde{c}^\top \Phi(\langle x, \tilde{\theta} \rangle) - c^\top \Phi(\langle x, \theta \rangle) \right| \leq \left| \tilde{c}^\top \left( \Phi(\langle x, \tilde{\theta} \rangle) - \Phi(\langle x, \theta \rangle) \right) \right| + \left| (\tilde{c} - c)^\top \Phi(\langle x, \theta \rangle) \right|$$

$$\leq r \left| \langle x, \tilde{\theta} - \theta \rangle \right| + \epsilon_c \left\| \Phi(\langle x, \theta \rangle) \right\|$$

$$\lesssim r\epsilon_\theta \|x\| + \epsilon_c \left( \|x\| + \tau\sqrt{\log(1/\delta)} \right)$$

$$\lesssim (r\epsilon_\theta + \epsilon_c) \|x\| + \epsilon_c \tau \sqrt{\log(1/\delta)} \ .$$

We use the same event to bound the second factor.

$$\left| \tilde{c}^\top \Phi(\langle x, \tilde{\theta} \rangle) + c^\top \Phi(\langle x, \theta \rangle) - 2y \right| \lesssim (r + \mathrm{Lip}(f_*)) \|x\| + r\tau\sqrt{\log(1/\delta)} + |\xi|$$
$$\lesssim rC(\|x\| + 1 + |\xi|/\sigma) \,,$$

where $C = \max\{\mathrm{Lip}(f_*), \tau\sqrt{\log(1/\delta)}, \sigma\}$, as defined in the Lemma statement.

Hence, by taking $\epsilon_c = r\epsilon_\theta = r/n$, we have

$$\left| \ell(c, \theta; x, y) - \ell(\tilde{c}, \tilde{\theta}; x, y) \right| \lesssim \frac{1}{n} \left( r^2 C \|x\|^2 + (r^2 C^2 + r^2 C) \|x\| + r^2 C^2 + r \|x\| |\xi|/\sigma + r^2 C^2 |\xi|/\sigma \right)$$
$$\lesssim \frac{r^2 C^2}{n} \left( \|x\|^2 + \|x\|(1 + |\xi|/\sigma) + 1 \right) \,.$$

By applying Fact E.4 on all $\xi_i$ and the fact that $\|x_i\|^2 \lesssim d \log(n/\delta)$ for all $i$ with overwhelming probability, we conclude that with probability at least $1 - \delta/3$

$$\sup_{\theta, c} \left| L_n(c, \theta) - L_n(\tilde{c}, \tilde{\theta}) \right| \lesssim C^2 r^2 \cdot \frac{d \log(n/\delta)}{n} \,.$$

Likewise, bounds on the expectations of $|\xi|$ and $\|x\|$ similarly give

$$\sup_{\theta, c} \left| L(c, \theta) - L(\tilde{c}, \tilde{\theta}) \right| \lesssim C^2 r^2 \cdot \frac{d}{n} \,.$$

We conclude by bounding the second term using Bernstein's inequality with the sub-exponential norm bound of Lemma E.14. Recall that $D = \max\{d, N\}$. Then, for sufficiently large $C_0$ (and thus sufficiently large $C_1$),

$$\mathbb{P}\left[ \sup_{\tilde{\theta}, \tilde{c}} \left| L_n(\tilde{c}, \tilde{\theta}) - L(\tilde{c}, \tilde{\theta}) \right| \ge C_0 C^2 r^2 \sqrt{\frac{D \log(n/\delta)}{n}} \right] \le \left( \frac{3}{\epsilon_\theta} \right)^d \left( \frac{3r}{\epsilon_c} \right)^N \exp\left( -C_1 n \cdot \frac{D \log(n/\delta)}{n} \right)$$
$$\le \exp\left( d \log(3n) + N \log(3n) - C_1 D \log(n/\delta) \right)$$
$$\le \delta/3 \,.$$

$\square$

## E.4 Proof of Lemma E.3

**Lemma E.3** (Gradient flow escapes the equator). *Assume $\sum_j (j + A)^k \alpha_j^2 \le C$ for $A \le s$ and $k \le 3$. With probability at least $1/2 - 2\delta$ over the initial condition, the draw of the data, and the draw of the random features, if $n = \tilde{\Omega}(\max\{\lambda^{-4}(d + N)d^{s-1}, \lambda^{-2} d^{(s+3)/2}\})$ and $N = \Theta\left( \lambda^{-1} \log(\lambda^{-1} \delta^{-1}) \right)$ then the first phase of gradient flow with a randomly initialised $c(0) \sim \mathrm{Unif}\{c \in \rho \mathbb{S}^{N-1}; \|c\|_0 = N_0\}$ with $\rho = \Theta(\sqrt{N} N_0^{-(2+s)/2}(\tau^2 + \lambda N/N_0)^{-1})$ and $N_0 = \Theta\left( \log \frac{1}{\delta} \right)$ escapes the equator in time $T_0 = \tilde{O}\left( d^{s/2 - 1} \right)$.*

Recall our gradient flow dynamics in the first phase:

$$\dot{\theta}(t) = -\nabla_\theta L_n(c(0), \theta(t)) \,,$$

where $c(0) \sim \mathrm{Unif}(\{c \in \mathbb{R}^N; \|c\|_2 = \rho; \|c\|_0 = N_0\})$, $\theta(0) \sim \mathrm{Unif}(\mathbb{S}^{d-1})$, where $\rho$ is another parameter determining the initial norm of $c$.

Our goal is to show that the gradient flow trajectory is likely to cross the energy barrier $B_{\mathrm{crit}} = \tilde{\Theta}\left( \lambda^{-2} \Delta_{\mathrm{crit}} \right)$ and therefore avoid the bad critical points (see Eq. (38)). Denote by $m(t) = \langle \theta(t), \theta_* \rangle$ the trajectory of the correlation. We will show that, from an initial correlation $m(0) \sim 1/\sqrt{d}$, the gradient flow dynamics yield $\dot{m}(t) > 0$ for long enough to guarantee that $m(t)$ grows substantially. This will ultimately be sufficient to ensure that the loss crosses the previous energy barrier, provided $c(0)$ has an appropriate norm $\rho$.

Thanks to the concentration results from Lemma E.13 and Lemma E.1, we can first compute the correlation trajectory $m(t)$ for the population loss, and then extend them to the empirical gradients.

Assume that $m(0)$ and $c(0)$ are such that that $\text{sign}(\alpha_s c(0)^\top \mathcal{T}_s) = \text{sign}(m(0))$, which occurs with probability $1/2$ over the randomness of $c(0)$ and $m(0)$. By symmetry, we will assume $m(0) > 0$ and $\alpha_s c(0)^\top \mathcal{T}_s > 0$ for the rest of the proof. Let us express the population objective without offset as

$$L(c, \theta) = \|f_*\|_\gamma^2 + c^\top Q_\lambda c - 2m^s \bar{R} , \text{ with} \tag{52}$$

$$\bar{R}(m) := \alpha_s \langle c, \mathcal{T}_s \rangle + m \sum_{j \geq 0} \alpha_{j+s+1} \langle c, \mathcal{T}_{j+s+1} \rangle m^j . \tag{53}$$

From Lemma B.7, we know that the correlation $m(0)$ at initialization cannot be too small. More precisely,

$$\mathbb{P}\left(|m(0)| \geq \delta/\sqrt{d}\right) \leq 1 - 4\delta . $$

Moreover, the change in correlation according to the population gradient is given by

$$-\langle \nabla_\theta^{\mathbb{S}^{d-1}} L(c, \theta(t)), \theta_* \rangle = (1 - m^2)m^{s-1}R , \tag{54}$$

where we have defined

$$R(m) := s\alpha_s \langle c, \mathcal{T}_s \rangle + m \sum_{j \geq 0} (j+s+1)\alpha_{j+s+1} \langle c, \mathcal{T}_{j+s+1} \rangle m^j . \tag{55}$$

The following lemma, proved below, shows there exists $\gamma = \gamma(c(0)) > 0$ and $\bar{\gamma}$ such that $R(m) > R(0)/2$ for $m \in [0, \gamma)$ and $\bar{R}(m) > \bar{R}(0)/2$ for $m \in [0, \bar{\gamma})$.

**Lemma E.15.** *Let* $C_{f*,\tau} = 2\tau \left(\sum_{j>0}(j+s)^2 j^2 \alpha_{j+s}^2\right)^{1/2}$ *and* $\bar{C}_{f*,\tau} = 2\tau \left(\sum_{j>0}(j)^2 \alpha_{j+s}^2\right)^{1/2}$. *Then*

1. $R(m) > \frac{1}{2}s\alpha_s c^\top \mathcal{T}_s$ *for* $m \in [0, \gamma)$, *where*

$$\gamma \geq \frac{s\alpha_s c^\top \mathcal{T}_s}{2\rho C_{f*,\tau}} ,$$

2. $\bar{R}(m) > \frac{1}{2}\alpha_s c^\top \mathcal{T}_s$ *for* $m \in [0, \bar{\gamma})$, *where*

$$\bar{\gamma} \geq \frac{\alpha_s c^\top \mathcal{T}_s}{2\rho \bar{C}_{f*,\tau}} .$$

In other words, the gradient flow under the population loss sees a monotonically increasing correlation $m$ (since its time derivative under the population gradient flow is positive), until $m(t)$ reaches a value $\gamma = C\frac{\alpha_s c^\top \mathcal{T}_s}{\rho}$.

Let $\gamma_* = \min(\gamma, \bar{\gamma})$ and $\rho_0 = \rho\sqrt{\frac{N_0}{N}}$. As the correlation reaches the value $m = \gamma_*$, using Lemma E.15 to lower bound $\bar{R}$, one can verify that the population loss obeys the following upper bound:

$$L_{\text{esc}} \leq \|f_*\|_\gamma^2 + \rho^2 \langle c/\|c\|, Q_\lambda c/\|c\| \rangle - \rho \gamma_*^s \alpha_s \langle c/\|c\|, \mathcal{T}_s \rangle \tag{56}$$

$$= \|f_*\|_\gamma^2 + \lambda \rho^2 + \rho_0^2 \langle \tilde{c}, \tilde{Q}\tilde{c} \rangle - \rho_0 \gamma_*^s \alpha_s \langle \tilde{c}, \tilde{\mathcal{T}}_s \rangle , \tag{57}$$

where, denoting by $\mathcal{S}$ the support of $c$, we defined

$$\tilde{Q} = \frac{1}{N_0} \left[\langle \phi(\cdot - b_j), \phi(\cdot - b_{j'}) \rangle_\gamma\right]_{j,j' \in \mathcal{S}} \in \mathbb{R}^{N_0 \times N_0} \tag{58}$$

$$\tilde{\mathcal{T}}_s = \frac{1}{\sqrt{N_0}} \left[\langle h_s, \phi(\cdot - b_j) \rangle_\gamma\right]_{j \in \mathcal{S}} \in \mathbb{R}^{N_0} \tag{59}$$

$$\tilde{c} = \frac{1}{\rho}[c_j]_{j \in \mathcal{S}} \in \mathbb{R}^{N_0}. \tag{60}$$

By Lemma E.12, we have $\|\tilde{Q}\|_{\text{op}} \le 2\tau^2$ w.p. $1 - \delta$ as soon as $N_0 \gtrsim \log(1/\delta)$, so that the bound above becomes:

$$L_{\text{esc}} \le \|f_*\|_\gamma^2 + \lambda\rho^2 + 2\tau^2\rho_0^2 - \rho_0\gamma_*^s\alpha_s\langle\tilde{c}, \tilde{\mathcal{T}}_s\rangle. \tag{61}$$

Let us now verify that the empirical correlation trajectory and loss have the same behavior. Observe that

$$
\begin{aligned}
\dot{m}(t) &= -\langle\nabla_\theta^{\mathbb{S}^{d-1}} L_n(c, \theta(t)), \theta_*\rangle \\
&= -\langle\nabla_\theta^{\mathbb{S}^{d-1}} L(c, \theta(t)), \theta_*\rangle + \langle\nabla_\theta^{\mathbb{S}^{d-1}} L(c, \theta(t)) - \nabla_\theta^{\mathbb{S}^{d-1}} L_n(c, \theta(t)), \theta_*\rangle \\
&= (1 - m^2)m^{s-1}R(m) + \widetilde{O}\left(\lambda^{-2}\max\left\{\sqrt{\frac{D}{n}}, \frac{d^2}{n}\right\}\right).
\end{aligned} \tag{62}
$$

From the anti-concentration Lemma B.7, it follows that whenever $n = \tilde{\Omega}(\max\{\lambda^{-4}Dd^{s-1}, \lambda^{-2}d^{\frac{s+3}{2}}\})$, with probability greater than $1 - \delta$

$$(1 - m(0)^2)(m(0))^{s-1}R(m(0)) \gg \widetilde{O}\left(\lambda^{-2}\max\left\{\sqrt{\frac{D}{n}}, \frac{d^2}{n}\right\}\right) \tag{63}$$

and therefore from Lemma E.15 we deduce that $\dot{m}(0) > 0$, and $m(t)$ keeps increasing at least until it reaches $\gamma_*$. From Lemma E.13, the empirical loss at this correlation level is with probability greater than $1 - \delta$

$$
\begin{aligned}
L_{n,\text{esc}} &\le \|f_*\|_\gamma^2 + \sigma^2 + \lambda\rho^2 + 2\tau^2\rho_0^2 - \rho_0\gamma_*^s\alpha_s\langle\tilde{c}, \tilde{\mathcal{T}}_s\rangle + \widetilde{O}\left(\lambda^{-2}\sqrt{\frac{D}{n}}\right) \\
&= \|f_*\|_\gamma^2 + \sigma^2 + \lambda\rho^2 + 2\tau^2\rho_0^2 - C\rho_0(\alpha_s\langle\tilde{c}, \tilde{\mathcal{T}}_s\rangle)^{s+1} + \widetilde{O}\left(\lambda^{-2}\sqrt{\frac{D}{n}}\right)
\end{aligned}
$$

In order to ensure that this initial training phase escapes the 'bad' empirical points near the equator $|m| \approx 0$, by Eq. (38), it is sufficient to show that

$$\lambda\rho^2 + 2\tau^2\rho_0^2 - C\rho_0(\alpha_s\langle\tilde{c}, \tilde{\mathcal{T}}_s\rangle)^{1+s} \ll -\widetilde{O}\left(\lambda^{-2}\Delta_{\text{crit}}\right), \tag{64}$$

with $\Delta_{\text{crit}} := \max\left\{\sqrt{\frac{D}{n}}, \left(\frac{d^2}{n}\right)^{\frac{2s}{2s-1}}\right\}$.

Let us now study the term $\langle\tilde{c}, \tilde{\mathcal{T}}_s\rangle$ for the choice of sparsity $N_0$ we picked for $c$. Let $\mu_s = \langle h_s, \Sigma h_s\rangle$. Observe that $\mu_s > 0$ since the kernel is universal. We have the following anti-concentration result:

**Lemma E.16** (Anticoncentration of $|\langle\tilde{c}, \tilde{\mathcal{T}}_s\rangle|$). *We have*

$$\mathbb{P}\left(|\langle\tilde{c}, \tilde{\mathcal{T}}_s\rangle| \ge \frac{\mu_s\delta}{8\sqrt{N_0}}\right) \ge 1 - 2e^{\frac{N_0 c\mu_s^2}{4\tau^4}} - \delta, \tag{65}$$

*where the probability is over both the initial draw of $c$ and the draw of the random features.*

We obtain that $\alpha_s\langle\tilde{c}, \tilde{\mathcal{T}}_s\rangle \ge \frac{\alpha_s\mu_s\delta}{8\sqrt{N_0}}$ holds with probability close to $1/2$. Then, the condition (64) becomes

$$\lambda\rho^2 + 2\tau^2\rho_0^2 - C'\rho_0 N_0^{-\frac{1+s}{2}} = (\lambda N/N_0 + 2\tau^2)\rho_0^2 - C'\rho_0 N_0^{-\frac{1+s}{2}} \ll -\widetilde{O}(\lambda^{-2}\Delta_{\text{crit}}), \tag{66}$$

with $C'$ a positive constant.

Taking $\rho_0 = \frac{C'N_0^{-\frac{1+s}{2}}}{2(2\tau^2 + \lambda N/N_0)}$ yields the new condition

$$\frac{C''N_0^{-(1+s)}}{2\tau^2 + \lambda N/N_0} = \widetilde{\Omega}\left(\lambda^{-2}\Delta_{\text{crit}}\right). \tag{67}$$

In particular, since we assume $\lambda^{-2}\Delta_{\mathrm{crit}} \ll 1$, we may take $N_0 = \Theta(1)$ and

$$N = \widetilde{O}\left(\lambda\Delta_{\mathrm{crit}}^{-1}\right)$$

to ensure (67), and consequently (64).

Finally, let us upper bound the escape time $T$ needed to reach $m(T) = \gamma_*$. Denote $\Delta_n := \lambda^{-2}\max\left\{\sqrt{\frac{D}{n}}, \frac{d^2}{n}\right\}$. Observe that for $t \leq T$,

$$
\begin{aligned}
\dot{m}(t) &\geq \frac{1}{2}(1-m^2)m^{s-1}s\alpha_s c^\top \mathcal{T}_s - \widetilde{O}\left(\Delta_n\right) \\
&\geq A(1-\gamma_*^2)m(t)^{s-1} - \widetilde{O}\left(\Delta_n\right) \\
&:= G(m(t)) - \widetilde{O}\left(\Delta_n\right) ,
\end{aligned}
\tag{68}
$$

where $G(u) = \tilde{A}u^{s-1}$ is convex in $[0,1]$ with

$$\tilde{A} = \frac{1}{2}(1-\gamma_*^2)s\alpha_s c^\top \mathcal{T}_s .$$

A crude bound is therefore

$$
\begin{aligned}
\dot{m}(t) &\geq G(m(0)) + G'(m(0))(m(t) - m(0)) - \widetilde{O}\left(\Delta_n\right) , \\
&:= A + Bm(t) ,
\end{aligned}
\tag{69}
$$

with

$$
\begin{aligned}
A &= G(m(0)) - G'(m(0))m(0) - \widetilde{O}\left(\Delta_n\right) = \tilde{A}m(0)^{s-1}(2-s) - \widetilde{O}\left(\Delta_n\right) \\
B &= G'(m(0)) = \tilde{A}(s-1)m(0)^{s-2} ,
\end{aligned}
\tag{70}
$$

which leads to a Gronwall-type inequality of the form

$$m(t) \geq \frac{A}{B-1}\left(e^{Bt} - 1\right) + m(0) ,\tag{71}$$

and therefore

$$T \leq B^{-1}\log\left(\frac{\gamma_*(B-1)}{A}\right) = \tilde{A}^{-1}\widetilde{O}\left(d^{s/2-1}\right) = \widetilde{O}\left(d^{s/2-1}\right) ,\tag{72}$$

since $\tilde{A} = \Theta(|c^\top \mathcal{T}_s|) = \Theta\left(N_0^{-\frac{2+s}{2}}\right) = \Theta(1)$. This concludes the proof. $\square$

*Proof of Lemma E.15.* Recall that $\mathcal{T} : L^2(\gamma) \to \mathbb{R}^N$ is the operator $\mathcal{T}f = (\langle f, \phi_{b_i}^{\varepsilon_i}\rangle_\gamma)_{i=1\ldots n}$. Its operator norm is bounded by Lemma E.12 with probability $1-\delta$ over the random features: $\|\mathcal{T}\|_{\mathrm{op}} \leq \|\hat{\Sigma}\|_{\mathrm{op}}^{1/2} \leq 2\tau$. Since $R(0) = s\alpha_s\langle c, \mathcal{T}_s\rangle$ and

$$
\begin{aligned}
R'(m) &= \left\langle c, \sum_{j>0}(j+s)j\alpha_{j+s}\mathcal{T}_{j+s}m^{j-1}\right\rangle \\
&= \left\langle c, \sum_{j>0}(j+s)j\alpha_{j+s}m^{j-1}\mathcal{T}h_{j+s}\right\rangle \\
&= \left\langle c, \mathcal{T}\left(\sum_{j>0}(j+s)jm^{j-1}\alpha_{j+s}h_{j+s}\right)\right\rangle
\end{aligned}
\tag{73}
$$

satisfies

$$
\begin{aligned}
\sup_{m\in[0,1]}|R'(m)| &\leq \|\mathcal{T}\|\rho\left\|\sum_{j>0}(j+s)jm^{j-1}\alpha_{j+s}h_{j+s}\right\|_\gamma \\
&\leq 2\tau\rho\sqrt{\sum_{j>0}(j+s)^2j^2m^{2(j-1)}\alpha_{j+s}^2} \\
&\leq \rho C_{f_*,\tau} .
\end{aligned}
\tag{74}
$$

Thus, if we assume that $m \leq \gamma$ as specified in the theorem statement,

$$R(m) \geq R(0) - |R(m) - R(0)| \geq s\alpha_s \langle c, \mathcal{T}_s \rangle - \sup_{m \in [0,1]} |R'(m)| \, |m - 0|$$

$$\geq s\alpha_s \langle c, \mathcal{T}_s \rangle - m\rho C_{f_*} \geq \frac{1}{2} s\alpha_s \langle c, \mathcal{T}_s \rangle.$$

The derivation for $\bar{R}(m)$ is analogous. $\qquad\square$

*Proof of Lemma E.16.* Observe that the dot product $\langle \tilde{c}, \tilde{\mathcal{T}}_s \rangle$ only depends on a subset of $N_0$ random features. Let us denote by

$$Z_s = \langle h_s, \sigma(\cdot - Z) \rangle_\gamma, \text{ where } Z \sim \gamma_\tau .$$

Observe that $Z_s = \psi(Z)$ with $\psi(x) = \langle h_s, \sigma(\cdot - x) \rangle_\gamma$ satisfying

$$|\psi'(x)| = |\langle h_s, \sigma'(\cdot - x) \rangle_\gamma| \leq \|h_s\|_\gamma \|\sigma'(\cdot - x)\|_\gamma \leq 1 ,$$

which shows that $Z_s$ is $\frac{1}{2\tau^2}$-subgaussian, and thus that the random vector

$$\tilde{\mathcal{T}}_s = \frac{1}{\sqrt{N_0}} (\langle h_s, \sigma(\cdot - b_j) \rangle_\gamma; j \in \mathrm{supp}(c)) \in \mathbb{R}^{N_0}$$

has independent $\frac{N_0}{2\tau^2}$-subgaussian entries. Therefore, by Bernstein concentration [83, Theorem 3.1.1], the Euclidean norm $\|\tilde{\mathcal{T}}_s\|$ concentrates around its expectation $\mu_s = \sqrt{\langle h_s, \Sigma h_s \rangle}$ as

$$\mathbb{P}_\Phi(|\|\tilde{\mathcal{T}}_s\| - \mu_s| \geq t) \leq 2e^{-\frac{cN_0 t^2}{\tau^4}} .$$

Finally, using again the anticoncentration of the correlation of a uniform direction with a fixed direction (Lemma B.7), we obtain with a union bound that

$$\mathbb{P}_{c,\Phi}\left(|\langle \tilde{c}, \tilde{\mathcal{T}}_s \rangle| \geq \frac{\mu_s \delta}{4\sqrt{N_0}}\right) \geq 1 - \delta - 2e^{-\frac{cN_0 \mu_s^2}{4\tau^4}} , \tag{75}$$

as claimed. $\qquad\square$

## E.5 Proof of Corollary 6.2

We restate Corollary 6.2 here for convenience.

**Corollary 6.2** (Excess risk of Algorithm 1). *Under the assumptions of Theorem 6.1, and further assuming $n \gtrsim d^3$, an appropriate choice of $\lambda$ yields an excess risk guarantee of the form*

$$\|\hat{F} - F^*\|_{\gamma_d}^2 = \widetilde{O}\left(\left(\frac{d}{n}\right)^{\frac{\beta}{\beta+4}} + \left(\frac{1}{n}\right)^{\frac{\beta}{\beta+5}}\right) , \tag{12}$$

*where $\beta$ is defined as in Lemma 4.4.*

*Proof.* Let $\hat{F}(x) = \hat{f}(\langle x, \hat{\theta} \rangle)$, and $G_{m,\hat{\theta}}(x) = g_m(\langle \hat{\theta}, x \rangle) = \sum_j \alpha_j m^j h_j(\langle \hat{\theta}, x \rangle)$, where $m = \langle \hat{\theta}, \theta^* \rangle$. We have

$$\|\hat{F} - F^*\|_{\gamma_d}^2 \leq 2\|\hat{F} - G_{m,\hat{\theta}}\|_{\gamma_d}^2 + 2\|G_{m,\hat{\theta}} - F^*\|_{\gamma_d}^2 \tag{76}$$

$$= 2\|\hat{f} - g_m\|_\gamma^2 + 2\|G_{m,\hat{\theta}} - F^*\|_{\gamma_d}^2 . \tag{77}$$

Denoting $c_\theta = Q_\lambda^{-1} \mathcal{T} g_m$, and considering $N = \Theta(\frac{1}{\lambda} \log \frac{1}{\lambda})$, recall that we have

$$\nabla_c L(\hat{c}, \hat{\theta}) = 2Q_\lambda(\hat{c} - c_\theta), \qquad \|\nabla_c L(\hat{c}, \hat{\theta})\| \leq \widetilde{O}\left(\lambda^{-1}\sqrt{\frac{d+N}{n}}\right) = \widetilde{O}\left(\sqrt{\frac{d}{\lambda^2 n} + \frac{1}{\lambda^3 n}}\right) ,$$
$$\tag{78}$$

Thus, we obtain

$$\|\hat{f} - \hat{P}_\lambda g_m\|_\gamma^2 = \|Q(\hat{c} - c_\theta)\|^2 \leq \|Q_\lambda(\hat{c} - c_\theta)\|^2 = \frac{1}{4}\|\nabla_c L(\hat{c}, \theta)\|^2 \leq \widetilde{O}\left(\frac{d}{\lambda^2 n} + \frac{1}{\lambda^3 n}\right) . \quad (79)$$

As a consequence, using Lemmas C.2 and 4.4 we obtain

$$\|\hat{f} - g_m\|_\gamma^2 \leq 2\left(\|\hat{f} - \hat{P}_\lambda g_m\|_\gamma^2 + \|(I - \hat{P}_\lambda)g_m\|_\gamma^2\right) \leq \widetilde{O}\left(\frac{d}{\lambda^2 n} + \frac{1}{\lambda^3 n} + 2\lambda^\beta\|f_*''\|_\gamma^2\right) . \quad (80)$$

On the other hand, we also have

$$\|G_{m,\hat{\theta}} - F_*\|_{\gamma_d}^2 = \sum_j \alpha_j^2 m^{2j} + \sum_j \alpha_j^2 - 2\sum_j m^{2j}\alpha_j^2 \qquad (81)$$

$$= \sum_j (1 - m^{2j})\alpha_j^2 \qquad (82)$$

$$\leq (1 - |m|)\sum_j 2j\alpha_j^2 \qquad (83)$$

$$= O\left(1 - |m|\right) = \widetilde{O}\left(\lambda^{-4}\frac{d+N}{n}\right) = \widetilde{O}\left(\frac{d}{\lambda^4 n} + \frac{1}{\lambda^5 n}\right) . \qquad (84)$$

where the $\widetilde{O}$ follows from Lemma E.2. We thus obtain

$$\|\hat{F} - F^*\|_{\gamma_d}^2 \leq \widetilde{O}\left(\frac{d}{\lambda^2 n} + \frac{d}{\lambda^4 n} + \frac{1}{\lambda^3 n} + \frac{1}{\lambda^5 n} + \lambda^\beta\right) = \widetilde{O}\left(\frac{d}{\lambda^4 n} + \frac{1}{\lambda^5 n} + \lambda^\beta\right) .$$

Setting

$$\lambda = \max\left\{\left(\frac{1}{n}\right)^{\frac{1}{\beta+5}}, \left(\frac{d}{n}\right)^{\frac{1}{\beta+4}}\right\},$$

we then have

$$\|\hat{F} - F^*\|_{\gamma_d}^2 = \widetilde{O}\left(\max\left\{\left(\frac{d}{n}\right)^{\frac{\beta}{\beta+4}}, \left(\frac{1}{n}\right)^{\frac{\beta}{\beta+5}}\right\}\right) , \qquad (85)$$

which establishes the desired rate. It remains to check that the upper bound $N = \tilde{O}(\lambda\Delta_{\text{crit}}^{-1})$ required by Theorem 6.1 holds with this choice of $\lambda$. Note that we have

$$\frac{N\Delta_{\text{crit}}}{\lambda} = \widetilde{O}\left(\frac{1}{\lambda^2}\sqrt{\frac{d + \frac{1}{\lambda}}{n}}\right)$$

$$= \widetilde{O}\left(\sqrt{\frac{d}{\lambda^4 n} + \frac{1}{\lambda^5 n}}\right) = o(1),$$

where we control the quantity inside the square root in the same way as we obtained (85). The calculation above also shows that $\frac{\Delta_{\text{crit}}}{\lambda^2} = o(1)$, so that the requirement $\lambda = \Omega(\sqrt{\Delta_{\text{crit}}})$ is also satisfied. $\qquad \square$

### E.6  Proof of Proposition 6.3

**Proposition 6.3** (Excess risk of fine-tuning). *Let $\delta \in (0, 1/4)$. Let $m = \langle\theta^*, \hat{\theta}\rangle$, where $\hat{\theta}$ is obtained from the previous gradient descent phase, and let $\hat{c}$ be the ridge regression estimator obtained from a fresh dataset $\mathcal{D}'$ of $n'$ samples, $N$ random features, and regularization parameter $\lambda_{n'} := (\sigma^2\tau^2/\|f_*''\|_\gamma^2 n')^{1/(\beta+1)}$, and let $\hat{F}(x) = \hat{c}^\top\Phi(\langle\hat{\theta}, x\rangle)$. Assume*

$$n' \gtrsim \max\left\{\sigma^2\tau^2/\|f_*''\|_\gamma^2, (\|f_*''\|_\gamma^2/\sigma^2\tau^2)^{1/\beta}, \|f_*\|_\infty^2/(\sigma^2\tau^2)^{\beta/(\beta+1)}\right\} , \text{ and}$$

$$N \gtrsim C_\tau \left(n'\|f_*''\|_\gamma^2/\sigma^2\tau^2\right)^{\frac{1}{\beta+1}} \log\left(n'^{1/(\beta+1)}\delta^{-1}\right) .$$

*Then with probability at least $1 - \delta$ over the random features, we have*

$$\mathop{\mathbb{E}}_{\mathcal{D}'}[\|\hat{F} - F^*\|^2_{\gamma_d}|\hat{\theta}] \lesssim \|f''_*\|^{\frac{2}{\beta+1}}_{\gamma} \left(\frac{\sigma^2 \tau^2}{n'}\right)^{\frac{\beta}{\beta+1}} + \|f'_*\|^2_{\gamma}(1 - |m|), \tag{14}$$

*where the expectation is over the $n'$ fresh samples, and is conditioned on the previously obtained $\hat{\theta}$.*

*Proof.* Let $\hat{\kappa}_\theta(x, x') = \Phi(\langle\theta, x\rangle)^\top \Phi(\langle\theta, x'\rangle)$ be the random feature kernel on $\mathbb{R}^d$ and denote by $\hat{\mathcal{H}}_\theta$ the corresponding RKHS. Let $\kappa_\theta$ and $\mathcal{H}_\theta$ be their infinite-width counterparts. Note that $\kappa_\theta(x, x') = \langle\varphi(\langle\theta, x\rangle), \varphi(\langle\theta, x'\rangle)\rangle_\mathcal{H}$, where $\varphi(u) = \kappa(u, \cdot)$ denotes the kernel mapping of $\mathcal{H}$. Then, one can easily show, e.g. using Theorem B.8, that $\mathcal{H}_\theta = \{F = f(\langle\theta, \cdot\rangle) : f \in \mathcal{H}\}$, with $\|F\|_{\mathcal{H}_\theta} = \|f\|_\mathcal{H}$ when $F(x) = f(\langle\theta, x\rangle)$.

Considering fresh samples $(x_i, y_i)$, $i = 1, \ldots, n'$, with $y_i = F^*(x_i) + \epsilon_i$, $\mathbb{E}[\epsilon_i|x_i] = 0$, $\mathrm{Var}[\epsilon_i|x_i] \leq \sigma^2$, we now assume

$$c = \left(\frac{1}{n'}\sum_i \Phi(\langle\theta, x_i\rangle)\Phi(\langle\theta, x_i\rangle)^\top + \lambda I\right)^{-1} \frac{1}{n'}\sum_i y_i \Phi(\langle\theta, x_i\rangle)$$

Define

$$Q = \mathop{\mathbb{E}}_{x\sim\gamma_d}[\Phi(\langle\theta, x\rangle)\Phi(\langle\theta, x\rangle)^\top] = \mathop{\mathbb{E}}_{z\sim\gamma}[\Phi(z)\Phi(z)^\top] \in \mathbb{R}^{N\times N},$$

and

$$\hat{Q} = \frac{1}{n'}\sum_i \Phi(\langle\theta, x_i\rangle)\Phi(\langle\theta, x_i\rangle)^\top.$$

Assume for now that $F^*$ belongs to $\hat{\mathcal{H}}_\theta$ and takes the form $F^*(x) = c_*^\top \Phi(\langle\theta, x\rangle) = \tilde{f}(\langle\theta, x\rangle)$. Then, we may use a variant of [7, Prop 7.2], which holds for bounded data, to our unbounded setting, by adapting the covariance concentration step [7, Proposition 7.1].

**Lemma E.17** (Concentration for covariance operators, sub-exponential case). *For $n' \geq \frac{R^2}{2\lambda}\log\frac{R^2}{\lambda}$, where $R$ is a universal constant, with probability greater than $1 - 7\mathrm{Tr}(QQ_\lambda^{-1})\exp\left(-\frac{n'}{8R^2+2R}\right)$ it holds*

$$-\frac{1}{2}I \preceq (Q + \lambda I)^{-1/2}(Q - \hat{Q})(Q + \lambda I)^{-1/2} \preceq \frac{1}{2}I. \tag{86}$$

Then, we have for $n' \geq \frac{R^2}{2\lambda}\log\frac{R^2}{\lambda}$ and $\lambda \leq R^2$, following [7, Proposition 7.1]

$$\mathbb{E}[\|F_{c,\theta} - F^*\|^2_{\gamma_d}] \leq 16\frac{\sigma^2}{n'}\mathrm{Tr}(Q(Q+\lambda I)^{-1}) + 16\inf_{F\in\hat{\mathcal{H}}_\theta}\left\{\|F - F^*\|^2_{\gamma_d} + \lambda\|F\|^2_{\hat{\mathcal{H}}_\theta}\right\} + \frac{24}{n'^2}\|F^*\|^2_\infty, \tag{87}$$

where the expectation is over the $n'$ fresh samples.

We have the following upper bound on the first (variance) term

$$\mathrm{Tr}(Q(Q + \lambda I)^{-1}) \leq \frac{\mathrm{Tr}(Q)}{\lambda} = \frac{\mathrm{Tr}(\hat{\Sigma})}{\lambda} \leq \frac{\frac{1}{2} + \tau^2}{\lambda} \leq \frac{2\tau^2}{\lambda}, \tag{88}$$

where we used Lemma E.12.

The approximation error may be controlled as follows:

$$
\begin{aligned}
\inf_{F\in\hat{\mathcal{H}}_\theta}\left\{\|F - F^*\|^2_{\gamma_d} + \lambda\|F\|^2_{\hat{\mathcal{H}}_\theta}\right\} &= \lambda\langle c_*, Q(Q + \lambda I)^{-1}c_*\rangle \\
&= \lambda\langle\tilde{f}, (\hat{\Sigma} + \lambda I)^{-1}\tilde{f}\rangle_\gamma \\
&\leq 4\lambda\langle\tilde{f}, (\Sigma + \lambda I)^{-1}\tilde{f}\rangle_\gamma \quad\text{(by Lemma C.3)} \\
&= 4\lambda\inf_{f\in\mathcal{H}}\left\{\|f - \tilde{f}\|^2_\gamma + \|f\|^2_\mathcal{H}\right\} \\
&= 4\lambda\inf_{F\in\hat{\mathcal{H}}_\theta}\left\{\|F - F^*\|^2_{\gamma_d} + \|F\|^2_{\hat{\mathcal{H}}_\theta}\right\},
\end{aligned}
$$

where we assume $N \geq C d_{\max}(\tau, \lambda) \log(d_{\max}(\tau, \lambda)/\delta)$ in order to apply Lemma C.3.

We thus obtain

$$\mathbb{E}[\|F_{c,\theta} - F^*\|_{\gamma_d}^2] \lesssim \frac{\sigma^2 \tau^2}{n'\lambda} + \inf_{F \in \mathcal{H}_\theta} \left\{ \|F - F^*\|_{\gamma_d}^2 + \lambda\|F\|_{\mathcal{H}_\theta}^2 \right\} + \frac{1}{n'^2}\|F^*\|_\infty^2. \tag{89}$$

By limiting arguments, we may show that this holds for any $F^*$ in the closure of $\mathcal{H}_\theta$. Now consider the true $F^*(x) = f_*(\langle\theta^*, x\rangle)$. When $\theta^* \neq \theta$, $F^*$ does not belong to the closure of $\mathcal{H}_\theta$, but we may consider the projection $F^*_{\mathcal{H}_\theta}$ on this closure. Then, following the arguments of [7, Section 7.6.4], we obtain

$$\mathbb{E}[\|F_{c,\theta} - F^*\|_{\gamma_d}^2] \lesssim \frac{\sigma^2 \tau^2}{n'\lambda} + \inf_{F \in \mathcal{H}_\theta} \left\{ \|F - F^*_{\mathcal{H}_\theta}\|_{\gamma_d}^2 + \lambda\|F\|_{\mathcal{H}_\theta}^2 \right\} + \|F^*_{\mathcal{H}_\theta} - F^*\|_{\gamma_d}^2 + \frac{1}{n'^2}\|F^*\|_\infty^2 . \tag{90}$$

We may take $F^*_{\mathcal{H}_\theta}(x) = g(\langle\theta, x\rangle)$ for some $g$, since all functions in $\mathcal{H}_\theta$ and its closure take this form. Then, we may consider $g$ of the form $g = \sum_j b_j h_j$, since such functions are dense in the closure of $\mathcal{H}_\theta$. Optimizing the approximation error $\|F^*_{\mathcal{H}_\theta} - F^*\|_{\gamma_d}$ over such $g$ yields $b_j = \alpha_j m^j$, so that the approximation error becomes

$$\|F^*_{\mathcal{H}_\theta} - F^*\|_{\gamma_d}^2 = \|F^*_{\mathcal{H}_\theta}\|_{\gamma_d}^2 + \|F^*\|_{\gamma_d}^2 - 2\langle F^*_{\mathcal{H}_\theta}, F^*\rangle_{\gamma_d}$$

$$= \sum_j \alpha_j^2 m^{2j} + \sum_j \alpha_j^2 - 2\sum_j \alpha_j^2 m^{2j}$$

$$= \sum_j \alpha_j^2(1 - m^{2j})$$

$$\leq 2(1 - |m|)\sum_j j\alpha_j^2 = (1 - |m|)C_{f_*},$$

with $C_{f_*} = 2\|f'_*\|_\gamma^2$, by using the bound $1 - m^{2j} \leq (1-|m|)(1+|m|+\cdots+|m|^{2j-1}) \leq 2j(1-|m|)$. We also have

$$\inf_{F \in \mathcal{H}_\theta} \left\{ \|F - F^*_{\mathcal{H}_\theta}\|_{\gamma_d}^2 + \lambda\|F\|_{\mathcal{H}_\theta}^2 \right\} = A(g, \lambda) \lesssim \lambda^\beta \|g''\|_\gamma^2 \leq \lambda^\beta \|f''_*\|_\gamma^2 . \tag{91}$$

The final bound becomes

$$\mathbb{E}[\|F_{c,\theta} - F^*\|_{\gamma_d}^2] \lesssim \frac{\sigma^2 \tau^2}{n'\lambda} + \lambda^\beta \|f''_*\|_\gamma^2 + C_{f_*}(1 - |m|) + \frac{1}{n'^2}\|f_*\|_\infty^2 . \tag{92}$$

Setting $\lambda = \left(\frac{\sigma^2\tau^2}{n'\|f''_*\|_\gamma^2}\right)^{\frac{1}{\beta+1}}$ yields

$$\mathbb{E}[\|F_{c,\theta} - F^*\|_{\gamma_d}^2] \lesssim \|f''_*\|_\gamma^{\frac{2}{\beta+1}} \left(\frac{\sigma^2\tau^2}{n'}\right)^{\frac{\beta}{\beta+1}} + C_{f_*}(1 - |m|) + \frac{1}{n'^2}\|f_*\|_\infty^2. \tag{93}$$

The condition $n' \gtrsim R^2/\lambda$ is satisfied when

$$n' \gtrsim \left(\frac{\|f''_*\|_\gamma^2}{\sigma^2\tau^2}\right)^{1/\beta},$$

while the condition $\lambda \leq R^2$ is satisfied when

$$n' \gtrsim \frac{\sigma^2\tau^2}{\|f''_*\|_\gamma^2}.$$

The last term is negligible when

$$n' \gtrsim \frac{\|f_*\|_\infty^2}{(\sigma^2\tau^2)^{\beta/(\beta+1)}} .$$

Finally, the requirement on $N$ scales as

$$N \geq \frac{C_\tau}{\lambda} \ln \frac{1}{\lambda\delta} \gtrsim C_\tau \left(\frac{n'\|f''_*\|_\gamma^2}{\sigma^2\tau^2}\right)^{\frac{1}{\beta+1}} \ln\left(\frac{n'^{1/(\beta+1)}}{\delta}\right) . \tag{94}$$

$\square$

*Proof of Lemma E.17.* We establish this matrix concentration result using a dimension-independent matrix Bernstein inequality for subexponential and potentially unbounded random matrices, by adapting arguments of Minsker [62, Eq. (3.9)] and Tropp [80, Theorem 6.2]. The sub-exponential tail assumption is established next, in Lemma E.19.

**Lemma E.18** (Dimension-independent matrix Bernstein bound)**.** *Let $X_1, \ldots, X_n$ be random i.i.d. self-adjoint operators with sub-exponential tails, in the sense that there exist self-adjoint operators $A_i$ and $R > 0$ such that*

$$\mathbb{E}[X_i] = 0 \qquad and \qquad \mathbb{E}[X_i^p] \preceq \frac{p!}{2} R^{p-2} A_i^2 \quad for \ p = 2, 3, 4, \ldots$$

*Defining the variance parameter*

$$\sigma^2 := \left\| \sum_{i=1}^n A_i^2 \right\|,$$

*we have the following for all $t \geq \sqrt{R^2 + 4\sigma^2}$:*

$$\mathbb{P}\left[ \left\| \sum_{i=1}^n X_i \right\|_{\mathrm{op}} \geq t \right] \leq \frac{7 \sum_i \mathrm{Tr}(A_i^2)}{\sigma^2} \exp\left( \frac{-t^2/2}{\sigma^2 + Rt} \right) . \tag{95}$$

Next we show that the sub-exponential bound needed for Lemma E.18 holds under our setting.

**Lemma E.19** (Sub-exponential tail for covariance concentration)**.** *For $X \sim N(0, I)$, let*

$$M = Q_\lambda^{-1/2}(\Phi_\theta(X)\Phi_\theta(X)^\top - Q)Q_\lambda^{-1/2},$$

*and define $B_\lambda = Q^{1/2}Q_\lambda^{-1/2}$. Then $\mathbb{E}[M] = 0$ and $\mathbb{E}[M^p] \preceq (R)^p \, p! B_\lambda B_\lambda^\top$ for $p \geq 2$ and some universal constant $R$.*

Let us finally establish (86). By defining $X_i = Q_\lambda^{-1/2}(\Phi_\theta(x_i)\Phi_\theta(x_i)^T - Q)Q_\lambda^{-1/2}$, Lemma E.19 guarantees subexponential tails, satisfying $\mathbb{E}[X_i^p] \preceq R^p p! B_\lambda B_\lambda^\top$ for $p \geq 2$ and a universal constant $R$, and where $B_\lambda = Q^{1/2}Q_\lambda^{-1/2}$. Therefore, defining $A_i^2 := 2R^2 B_\lambda B_\lambda^\top$, we have

$$\mathbb{E}[X_i^p] \preceq \frac{p!}{2} R^{p-2} A_i^2 .$$

We can now apply Lemma E.18. In that case, $\sigma^2 = 2R^2 n$, and by choosing $t = n/2$ in (95), we obtain

$$\begin{aligned}
\mathbb{P}\left[ \left\| (Q + \lambda I)^{-1/2}(Q - \hat{Q})(Q + \lambda I)^{-1/2} \right\|_{\mathrm{op}} \geq \frac{1}{2} \right] &\leq \frac{7 \sum_i \mathrm{Tr}(A_i^2)}{2R^2 n} \exp\left( -\frac{n^2/4}{n(2R^2 + R/2)} \right) \\
&\leq 7\mathrm{Tr}(Q^{1/2}Q_\lambda^{-1}Q^{1/2}) \exp\left( -\frac{n}{8R^2 + 2R} \right) ,
\end{aligned}$$

proving (86). $\qquad \square$

*Proof of Lemma E.18.* Let $S_n := \sum_{i=1}^n X_i$ and $\psi(\theta) := e^\theta - \theta - 1$. Following the argument of Theorem 3.1 of [62], we apply Markov's inequality and the monotonicity of $\psi$ to upper-bound the probability that the $\| \sum_i X_i \|_{\mathrm{op}}$ is large for any fixed $\theta > 0$ and $t > 0$.

$$\begin{aligned}
\mathbb{P}\left[ \|S_n\|_{\mathrm{op}} \geq t \right] = \mathbb{P}\left[ \psi\left( \theta \|S_n\|_{\mathrm{op}} \right) \geq \psi(\theta t) \right] &= \mathbb{P}\left[ \|\psi\left( \theta S_n \right)\|_{\mathrm{op}} \geq \psi(\theta t) \right] \\
&\leq \mathbb{P}\left[ \mathrm{Tr}(\psi\left( \theta S_n \right)) \geq \psi(\theta t) \right] \leq \frac{\mathbb{E}\left[ \mathrm{Tr}(\psi(\theta S_n)) \right]}{\psi(\theta t)} .
\end{aligned}$$

We continue to adapt the argument of [62] in order to bound the numerator, taking advantage of the fact that $\mathbb{E} X_i = 0$ (and hence, $\mathbb{E} S_i = 0$). We additionally apply Jensen's inequality and Lieb's

concavity theorem (Fact 5 of [62]).

$$\mathbb{E}\left[\text{Tr}(\psi(\theta S_n))\right] = \mathbb{E}\left[\text{Tr}\left(\exp(\theta S_n) - S_n - I\right)\right] = \mathbb{E}\left[\text{Tr}\left(\exp(\theta S_{n-1} + \log(\exp(X_n))) - I\right)\right]$$

$$\leq \mathop{\mathbb{E}}_{X_1,\ldots,X_{n-1}}\left[\text{Tr}\left(\exp\left(\theta S_{n-1} + \log\left(\mathop{\mathbb{E}}_{X_n}\left[\exp(\theta X_n)\right]\right)\right) - I\right)\right]$$

$$\leq \text{Tr}\left(\exp\left(\sum_{i=1}^{n}\log\left(\mathbb{E}\left[\exp(\theta X_i)\right]\right)\right) - I\right)$$

According to Lemma 6.8 of [80], by scaling $X_i$ appropriately and taking $\theta \in (0, \frac{1}{R})$, the assumptions in the theorem statement guarantee that

$$\mathbb{E}\left[\exp(\theta X_i)\right] \preccurlyeq \exp\left(\frac{(\theta R)^2}{2(1 - \theta R)} \cdot \frac{1}{R^2} A_i^2\right) = \exp\left(\frac{\theta^2}{2(1 - \theta R)} \cdot A_i^2\right),$$

for all $i \in [n]$. Let $B_n := \sum_{i=1}^{n} A_i^2$. As a result,

$$\mathbb{E}\left[\text{Tr}(\psi(\theta S_n))\right] \leq \text{Tr}\left(\exp\left(\sum_{i=1}^{n}\frac{\theta^2}{2(1 - \theta R)} \cdot A_i^2\right) - I\right) = \text{Tr}\left(\sum_{k=1}^{\infty}\frac{1}{k!}\left(\frac{\theta^2}{2(1 - \theta R)}B_n\right)^k\right)$$

$$= \text{Tr}\left(\frac{\theta^2}{2(1 - \theta R)}B^{1/2}\sum_{k=1}^{\infty}\frac{1}{k!}\left(\frac{\theta^2}{2(1 - \theta R)}B_n\right)^{k-1}B^{1/2}\right)$$

$$\leq \text{Tr}\left(\frac{\theta^2}{2(1 - \theta R)}B^{1/2}\sum_{k=1}^{\infty}\frac{1}{k!}\left(\frac{\theta^2}{2(1 - \theta R)}\|B_n\|_{\text{op}}\right)^{k-1}B^{1/2}\right)$$

$$= \frac{\theta^2}{2(1 - \theta R)}\text{Tr}(B)\sum_{k=1}^{\infty}\frac{1}{k!}\left(\frac{\theta^2}{2(1 - \theta R)}\sigma^2\right)^{k-1}$$

$$= \frac{\theta^2}{2(1 - \theta R)}\text{Tr}(B)\frac{\exp\left(\frac{\theta^2}{2(1 - \theta R)}\sigma^2\right) - 1}{\frac{\theta^2}{2(1 - \theta R)}\sigma^2} \leq \frac{\text{Tr}(B)}{\sigma^2}\exp\left(\frac{\theta^2}{2(1 - \theta R)}\sigma^2\right).$$

We conclude by putting the terms together to simplify the expression (continuing to borrow from [62]) while letting $\theta := \frac{t}{\sigma^2 + Rt}$ and requiring that $t$ be sufficiently large:

$$\mathbb{P}\left[\|S_n\|_{\text{op}} \geq t\right] \leq \frac{\text{Tr}(B)}{\sigma^2}\exp\left(\frac{\theta^2}{2(1 - \theta R)}\sigma^2\right) \cdot \frac{1}{\psi(\theta t)}$$

$$\leq \frac{\text{Tr}(B)}{\sigma^2}\exp\left(\frac{\theta^2}{2(1 - \theta R)}\sigma^2 - \theta t\right) \cdot \frac{\exp(\theta t)}{\psi(\theta t)}$$

$$\leq \frac{\text{Tr}(B)}{\sigma^2}\exp\left(\theta t\left(\frac{\theta \sigma^2}{2t(1 - \theta R)} - 1\right)\right)\left(1 + \frac{6}{(\theta t)^2}\right)$$

$$= \frac{\text{Tr}(B)}{\sigma^2}\exp\left(\frac{t^2}{\sigma^2 + Rt}\left(\frac{\sigma^2}{2(\sigma^2 + Rt)} \cdot \frac{\sigma^2 + Rt}{\sigma^2} - 1\right)\right)\left(1 + \frac{6(\sigma^2 + Rt)^2}{t^4}\right)$$

$$= \frac{7\text{Tr}(B)}{\sigma^2}\exp\left(-\frac{t^2/2}{\sigma^2 + Rt}\right). \qquad \square$$

*Proof of Lemma E.19.* $\mathbb{E}[M] = 0$ is clear. For $p \geq 2$, we bound the moments of $M$ by considering the subgaussianity of inner products.

We define $\Psi := Q^{-1/2}\Phi_\theta(X)$ and note that it is isotropic and subgaussian, i.e. $v^\top(\Psi - I_N)v$ is $C_1$-subgaussian for universal constant $C_1$ and any $v \in \mathbb{S}^{N-1}$. We also define $B_\lambda := Q^{1/2}Q_\lambda^{-1/2}$, so that $M = B_\lambda^\top(\Psi\Psi^\top - I_N)B_\lambda$.

Observe first that $v^\top(\Psi\Psi^\top - I_N)v$ is $C_2$-subexponential, i.e. $\mathbb{E}[\exp(C_2 v^\top(\Psi\Psi^\top - I_N)v)] \leq 2$, for any fixed $v \in \mathbb{S}^{N-1}$ because $\Psi$ is subgaussian and $I_N = \mathbb{E}[\Psi\Psi^\top]$. As a result, $\mathbb{E}[\exp(C_2(\Psi\Psi^\top - I_N))] \preccurlyeq 2I_N$.

Let us now write $M^p = B_\lambda^\top \tilde{M}_p B_\lambda$. We claim that $\|\mathbb{E}\, \tilde{M}_p\|_{\text{op}} \leq 4p!/C_2^p$ for any $p \geq 2$. Indeed, observe that $\tilde{M}_p$ is a product of matrices of the form $T_1 = (\Psi\Psi^\top - I)$ and $T_2 = B_\lambda B_\lambda^\top$. Since $\|T_2\|_{\text{op}} \leq 1$, we have

$$\|\mathbb{E}[\tilde{M}_p]\|_{\text{op}} \leq \|\mathbb{E}[(\Psi\Psi^\top - I)^p]\|_{\text{op}} .$$

Now, because any positive semi-definite matrix $A$ satisfies $A^p \preceq p!(e^A + e^{-A})$, we bound the moment $\mathbb{E}[v^\top(\Psi\Psi^\top - I)^p v]$ following the simple argument made in [83, Proposition 2.71].

$$\mathbb{E}[v^\top(\Psi\Psi^\top - I_N)^p v] \leq \frac{p!}{C_2^p} \cdot v^\top (\mathbb{E}[\exp(C_2(\Psi\Psi^\top - I_N))] + \mathbb{E}[\exp(-C_2(\Psi\Psi^\top - I_N))])v$$

$$\leq \frac{4p!}{C_2^p},$$

which shows that $\|\mathbb{E}[\tilde{M}_p]\|_{\text{op}} \leq \frac{4p!}{C_2^p}$. Therefore, we have

$$\mathbb{E}[v^\top M^p v] = \mathbb{E}[(B_\lambda v)^\top \tilde{M}_p (B_\lambda v)]$$

$$\leq \frac{4p!}{C_2^p}\|B_\lambda v\|^2 ,$$

which shows that $\mathbb{E}[M^p] \preceq C_p p! B_\lambda B_\lambda^\top$ for $p \geq 2$. The conclusion is immediate for a proper choice of constant $R$. $\qquad\square$

### E.7 Proof of Corollary 6.4

**Corollary 6.4** (Excess risk of Algorithm 2). *Let $\delta \in (0, 1/4)$. As in Theorem 6.1, let $\mu_s = \langle h_s, \Sigma h_s \rangle > 0$, and let $f_*$ satisfy Assumption 5.2. Let $\lambda = \Theta(1)$, and assume the following on the sample sizes and number of random features for the first phase $(n, N, N_0)$ and fine-tuning phase $(n', N')$:*

$$N = N_0 = \Theta\left(\frac{1}{\lambda}\log\frac{1}{\lambda\delta}\right), \quad n = \widetilde{\Omega}\left(\max\{d^s, d^{(s+3)/2}\}\right), \quad N' = \widetilde{\Omega}\left(n'^{\frac{1}{\beta+1}}\right).$$

*and let $\rho$ be as in Theorem 6.1. With probability at least $1/2 - 2\delta$ over the initial $n$ samples, initialization, random features, we have*

$$\mathbb{E}_{\mathcal{D}'}[\|\hat{F} - F_{\theta^*}\|_{\gamma_d}^2] \leq \widetilde{O}\left(\max\left\{\frac{d}{n}, \frac{d^4}{n^2}\right\} + \left(\frac{1}{n'}\right)^{\frac{\beta}{\beta+1}}\right), \tag{15}$$

*where the constants in $\widetilde{O}$ do not depend on $d$ other than through logarithmic factors.*

*Proof.* The result is immediate by applying Proposition 6.3 and using the following bound from Lemma E.2:

$$1 - |m| \leq \widetilde{O}\left(\lambda^{-4}\max\left\{\frac{d+N}{n}, \frac{d^4}{n^2}\right\}\right),$$

where $\lambda$ is a constant as given in the statement. Note that with a constant $\lambda$ as in the statement, the choice $N_0 = N \sim \lambda^{-1}$ for the first phase is sufficient for satisfying the assumptions of Theorem 6.1. $\qquad\square$

### E.8 Omitted proofs from Section E

**Lemma E.7.** *Let $f : \mathbb{R} \to \mathbb{R}$ be function such that its derivatives $f^{(1)}, \ldots, f^{(4)}$ are all in $L^2(\gamma)$. Let $f(z) = \sum_{j=0}^\infty \alpha_j h_j(z)$ be the Hermite expansion of $f$. Then,*

$$\sum_{j=1}^\infty j^2(j-1)^2\alpha_j^2 = \|f^{(4)}\|_\gamma^2 + 4\|f^{(3)}\|_\gamma^2 + 2\|f^{(2)}\|_\gamma^2 .$$

*Proof.* The statement follows from straightforward, albeit tedious, algebraic manipulation.

$$\sum_{j=1}^{\infty} j\alpha_j^2 = \|f^{(1)}\|_\gamma^2$$

$$\sum_{j=1}^{\infty} j^2\alpha_j^2 = \sum_{j=1}^{\infty} j(j-1)\alpha_j^2 + \sum_{j=1}^{\infty} j\alpha_j^2 = \|f^{(2)}\|_\gamma^2 + \|f^{(1)}\|_\gamma^2$$

$$\sum_{j=1}^{\infty} j^3\alpha_j^2 = \sum_{j=1}^{2} j^3\alpha_j^2 + \sum_{j=3}^{\infty} j(j-1)(j-2)\alpha_j^2 + 3\sum_{j=3}^{\infty} j^2\alpha_j^2 - 2\sum_{j=3}^{\infty} j\alpha_j^2$$

$$= \sum_{j=1}^{2} j^3\alpha_j^2 + \sum_{j=3}^{\infty} j(j-1)(j-2)\alpha_j^2 + 3\left(\sum_{j=1}^{\infty} j^2\alpha_j^2 - \sum_{j=1}^{2} j^2\alpha_j^2\right) - 2\left(\sum_{j=3}^{\infty} j\alpha_j^2 - \sum_{j=1}^{2} j\alpha_j^2\right)$$

$$= \|f^{(3)}\|_\gamma^2 + 3\|f^{(2)}\|_\gamma^2 + \|f^{(1)}\|_\gamma^2 + \sum_{j=1}^{2}(j^3 - 3j^2 + 2j)\alpha_j^2$$

$$= \|f^{(3)}\|_\gamma^2 + 3\|f^{(2)}\|_\gamma^2 + \|f^{(1)}\|_\gamma^2$$

$$\sum_{j=1}^{\infty} j^4\alpha_j^2 = \sum_{j=1}^{3} j^4\alpha_j^2 + \sum_{j=4}^{\infty} j(j-1)(j-2)(j-3)\alpha_j^2 + 6\sum_{j=4}^{\infty} j^3\alpha_j^2 - 11\sum_{j=4}^{\infty} j^2\alpha_j^2 + 6\sum_{j=4}^{\infty} j\alpha_j^2$$

$$= \sum_{j=1}^{3} j^4\alpha_j^2 + \|f^{(4)}\|_\gamma^2$$

$$+ 6\left(\sum_{j=1}^{\infty} j^3\alpha_j^2 - \sum_{j=1}^{3} j^3\alpha_j^2\right) - 11\left(\sum_{j=1}^{\infty} j^2\alpha_j^2 - \sum_{j=1}^{3} j^2\alpha^2\right) + 6\left(\sum_{j=4}^{\infty} j\alpha_j^2 - \sum_{j=1}^{3} j\alpha_j^2\right)$$

$$= \|f^{(4)}\|_\gamma^2 + 6(\|f^{(3)}\|_\gamma^2 + 3\|f^{(2)}\|_\gamma^2 + \|f^{(1)}\|_\gamma^2) - 11(\|f^{(2)}\|_\gamma^2 + \|f^{(1)}\|_\gamma^2) + 6\|f^{(1)}\|_\gamma^2$$

$$+ \sum_{j=1}^{3}(j^4 - 6j^3 + 11j^2 - 6j)\alpha_j^2$$

$$= \|f^{(4)}\|_\gamma^2 + 6\|f^{(3)}\|_\gamma^2 + 7\|f^{(2)}\|_\gamma^2 + \|f^{(1)}\|_\gamma^2 \ .$$

Using the above expressions for series of the form $\sum_{j=1}^{\infty} j^p\alpha_j^2$ for $p = 1, 2, 3, 4$, we conclude

$$\sum_{j=1}^{\infty} j^2(j-1)^2\alpha_j^2 = \sum_{j=1}^{\infty}(j^4 - 2j^3 + j^2)\alpha_j^2$$

$$= \|f^{(4)}\|_\gamma^2 + 4\|f^{(3)}\|_\gamma^2 + 2\|f^{(2)}\|_\gamma^2 \ .$$

$\square$

## F   Gradient Flow on Non-smooth Landscapes

As mentioned in Section 2, for our purposes we only require 1) the *existence* of a curve $z : [a, b] \to \mathbb{R}^p$ satisfying the subgradient dynamics (what we have conveniently referred to as "gradient flow" in earlier Sections) and 2) the *descent property*, which requires that the (empirical) loss $L$ be *non-increasing* along any such curve. We first introduce basic terminology and concepts used in non-smooth optimization.

For non-smooth objective functions defined on Euclidean domains, a subdifferential *set* $\partial L(\theta)$ is used in place of the gradient $\nabla L(\theta)$. We restrict our attention to locally Lipschitz objectives which enjoy the property that they are differentiable a.e. [16, Theorem 9.1.2]. Formally,

**Definition F.1** (Clarke Subdifferential). *For any locally Lipschitz function $L : \Omega \to \mathbb{R}$ with an open domain $\Omega \subseteq \mathbb{R}^p$, the* Clarke subdifferential *of $L$ at $\theta \in \Omega$ is defined by*

$$\partial L(\theta) = \text{conv}\left\{\lim_{i\to\infty} \nabla L(\theta_i) \ \middle| \ x_i \in \Omega, \ \nabla L(\theta_i) \text{ exists}, \ \lim_{i\to\infty} \theta_i = \theta\right\} \ .$$

*We denote by $\bar{\partial}L(\theta)$ the unique min-norm element of $\partial L(\theta)$.*

A curve satisfying the subgradient dynamics of a locally Lipschitz objective function $L$ is any absolutely continuous function $z : [a, b] \to \Omega$ which satisfies the following differential *inclusion* almost everywhere.

$$\dot{z}(t) \in -\partial L(z(t)) . \tag{96}$$

Closely related to curves satisfying the subgradient dynamics are *near-steepest descent* curves, which can be defined in any locally convex metric space (see [3, 32]). For any locally convex metric space $(\Omega, d)$ and any lower semicontinuous objective function $L : \Omega \to \mathbb{R}$ satisfying very weak continuity conditions, the existence of a near-steepest curve for $L$ emanating from any starting point $z_0 \in \Omega$ is guaranteed [32, Theorem 3.4] and along the curve the objective is non-increasing. Furthermore, if $L$ admits the *chain rule* (Definition F.2), then these two notions of curves coincide; near-steepest descent curves satisfy the subgradient dynamics a.e. and vice versa [32, Proposition 4.10]. Thus, the chain rule guarantees the *descent property* for any curve satisfying the subgradient dynamics [54, 29, 49].

For our purposes, it suffices to show that the empirical squared loss on any ReLU network satisfies the chain rule. Previous work by [29, 49] show that the chain rule holds for the class of functions *definable on some o-minimal structure* [81]. We simply write "$L$ is definable" in place of "$L$ is definable in some o-minimal structure". Notably, empirical squared loss functionals on ReLU networks, which can be viewed as real-valued functions w.r.t. the network parameters, are definable. We refer to [49, Appendix B] for further technical definitions and detailed proofs, but reproduce the formal statements here for convenience (See also [29, Theorem 5.8]).

**Definition F.2** (Chain rule [32, Definition 4.9])**.** *Consider a lower semicontinuous function $L : \mathbb{R}^p \to \mathbb{R}$. We say that $L$ admits a chain rule if for every absolutely continuous function $z : [a, b] \to \mathbb{R}^p$ for which $L \circ z$ is non-increasing and $L$ is subdifferentiable along $z$, the following equation holds for a.e. $t \in (a, b)$*

$$(L \circ z)'(t) = \langle z^*(t), \dot{z}(t) \rangle \text{ for all } z^*(t) \in \partial L(z(t)) .$$

**Lemma F.3** ([49, Lemma B.2])**.** *Any empirical squared loss functionals of any ReLU network (as a function w.r.t. the network parameters $\theta$) is definable.*

**Lemma F.4** (Chain rule adapted from [49, Lemma B.9])**.** *Given locally Lipschitz definable $L : \Omega \to \mathbb{R}$ with an open domain $\Omega \subseteq \mathbb{R}^p$, for any absolutely continuous function $z : [a, b] \to \Omega$, it holds for a.e. $t \in [a, b]$ that*

$$(L \circ z)'(t) = \langle z^*(t), \dot{z}(t) \rangle , \text{ for all } z^*(t) \in \partial L(z(t)) .$$

*Moreover, for the gradient inclusion*

$$\dot{z}(t) \in -\partial L(z(t)) ,$$

*it holds for a.e. $t \geq 0$ that $\dot{z}(t) = -\bar{\partial}L(z(t))$ and $dL(z(t))/dt = -\|\bar{\partial}L(z(t))\|_2^2$ and therefore*

$$L(z(a)) - L(z(b)) = \int_a^b \|\bar{\partial}L(z(\tau))\|_2^2 d\tau .$$

**Remark F.5** (Riemannian gradients)**.** *We also need to show existence and the descent property for* Riemannian *gradient flows on the unit sphere, in which the subdifferentials in the differential inclusion* (96) *are projected onto the tangent space of $z(t)$. This is because we take spherical gradients for the (shared) first layer weights. However, the desired results follow from the same theorems since the existence of a near-steepest descent curve only requires the objective function to be lower semi-continuous and satisfy some very weak continuity conditions. We can enforce any near-steepest descent curve to be contained in $\mathcal{S}^{p-1}$ by modifying the objective to $\tilde{L}(z) = L \cdot \delta_{\mathcal{S}^{p-1}}(z)$, where $\delta_{\mathcal{S}^{p-1}}(z)$ is 1 for $z \in \mathcal{S}^{p-1}$ and $\infty$ otherwise. By Lemma F.4 and Claim F.6 (see below), $\tilde{L}$ satisfies the chain rule for all curves $z : [a, b] \to \Omega$ contained entirely in $\mathcal{S}^{p-1}$. More precisely, the following observation implies that the chain rule holds for $\tilde{L}$ and any curve $z : [a, b] \to \mathbb{R}^p$ contained entirely in $\mathcal{S}^{p-1}$:*

$$\langle (I - z(t)z(t)^\top)z^*(t), \dot{z}(t) \rangle = \langle z^*(t), \dot{z}(t) \rangle - \langle z(t), z^*(t) \rangle \langle z(t), \dot{z}(t) \rangle = \langle z^*(t), \dot{z}(t) \rangle .$$

**Claim F.6.** *Let $z : [a, b] \to \mathbb{R}^p$ be an absolutely continuous function satisfying $\|z(t)\|_2 = 1$ for all $t \in [a, b]$. Then, the derivative $\dot{z}(t)$ exists almost all $t \in [a, b]$ and satisfies $\langle z(t), \dot{z}(t) \rangle = 0$.*

*Proof.* Since $[a, b]$ is compact and $z$ is absolutely continuous, $z$ is differentiable a.e. on $[a, b]$ by Rademacher's Theorem [16, Theorem 9.1.2]. Now consider the derivative of the *constant* function $\|z\|_2^2$. For any $t \in [a, b]$ such that $\dot{z}(t)$ exists, we have

$$\frac{d}{dt}\|z(t)\|_2^2 = 2\langle z(t), \dot{z}(t) \rangle = 0 \ .$$

$\square$

# G    Smooth Activation Functions

We discuss the impact of replacing the ReLU activation by a smooth activation $\phi$. This choice affects both approximation and optimization properties of the corresponding model. To illustrate this, we focus on Gaussian smoothing which we define using the Ornstein-Ulhenbeck semigroup.

**Definition G.1.** *For $\rho \in [0, 1]$, the Ornstein–Uhlenbeck noise operator $U_\rho$ is defined by*

$$U_\rho f(t) = \int f(\rho t + \sqrt{1 - \rho^2} u) d\gamma(u) \ .$$

**Assumption G.2** (Smoothed ReLU). *Given $\rho \in [0, 1]$ and $\phi(t) = \max(0, t)$, also known as the ReLU activation, we refer to $\phi_\rho = U_\rho \phi$ as the $\rho$-smoothed ReLU.*

The resulting activation is akin to the so-called Exponential Linear Unit (ELU) [25]. As will be shown next, we leverage hypercontractivity properties of the Gaussian measure defining $\phi_\rho$. From [44], our smoothing operator may be replaced by a more general one provided it satisfies a Log-Sobolev inequality, but such extensions are out of the present scope.

**Approximation properties.**    Let $\rho \in [0, 1]$ and let $\mathcal{H}_\rho$ be the RKHS associated with the kernel

$$\kappa_\rho(x, x') = \mathop{\mathbb{E}}_{b \sim \gamma_{\tilde{\tau}}, \varepsilon} [\phi_\rho(\varepsilon x - b)\phi_\rho(\varepsilon x' - b)] \ .$$

where $\tilde{\tau} = \rho\tau$, and for any $f \in L^2(\gamma)$, let

$$A(f, \lambda, \rho) := \inf_{h \in \mathcal{H}_\rho} \|f - h\|_\gamma^2 + \lambda \|h\|_{\mathcal{H}_\rho}^2 \ . \tag{97}$$

Recall the function space $\mathcal{F} = \{g \in H^2(\gamma) \mid g'' \in L^4(\gamma)\}$ (see Assumption 4.3). We define an alternate $\lambda$-regularized approximation error of $f$ with respect to the image of $\mathcal{F}$ under the operator $U_\rho$ by

$$B(f, \lambda, \rho) := \inf_{g \in \mathcal{F}} \|f - U_\rho g\|_\gamma^2 + \lambda(\|g\|_\gamma^2 + \|g''\|_4^2) \ . \tag{98}$$

The following proposition relates the approximation error achievable by $\mathcal{H}_\rho$ to that of $\mathcal{H}$.

**Proposition G.3** (Approximation error in $\mathcal{H}_\rho$). *Let $\tau > 1$ and $\beta = \frac{1 - 1/\tau^2}{3 + 1/\tau^2}$. Then, there exists a universal constant $C_0 > 0$ such that for any $\rho \in [0, 1]$ and any $f \in L^2(\gamma)$,*

$$A(U_\rho f, \lambda, \rho) \le A(f, \lambda) \ , \ \ \text{and} \ \ A(f, \lambda, \rho) \le C_0 \tau^{1+\beta} B(f, \lambda^\beta, \rho) \ .$$

*Proof.* We first consider target functions $f$ which satisfy the source condition $f = U_\rho f_0$, where $f_0 \in \mathcal{F}$. Consider $h^* = \arg\min_{h \in \mathcal{H}} \|f_0 - h\|_\gamma^2 + \lambda \|h\|_{\mathcal{H}}^2$. We verify from the definition that $\|f_0 - h^*\|^2 \le A(f_0, \lambda)$ and $\|h^*\|_{\mathcal{H}}^2 \le \lambda^{-1} A(f_0, \lambda)$. Now consider $h_\rho = U_\rho h^*$. Let $T_u$ be the translation operator $T_u f(t) = f(t - u)$. We verify that

$$U_\rho T_u f = \int f(\rho t + \sqrt{1 - \rho^2} z - u) d\gamma(z) = \int f(\rho(t - (u/\rho)) + \sqrt{1 - \rho^2} z) d\gamma(z) = T_{(u/\rho)} U_\rho f \ .$$

so $U_\rho T_u = T_{(u/\rho)} U_\rho$. From the RKHS representation of $h^*$.

$$h^*(t) = \int \phi(t-u) c(u) \gamma_\tau(u) du = \int T_u \phi(t) c(u) \gamma_\tau(u) du$$

with $\|c\|_\gamma^2 = \|h^*\|_{\mathcal{H}}^2$, we verify that

$$h_\rho(t) = U_\rho h^*(t) = \int U_\rho T_u \phi(t) c(u) \gamma_\tau(u) du$$

$$= \int T_{(u/\rho)} U_\rho \phi(t) c(u) \gamma_\tau(u) du = \int \phi_\rho(t - (u/\rho)) c(u) d\gamma_\tau(u)$$

$$= \int \phi_\rho(t-u) c(\rho u) \gamma_{\tilde\tau}(u) du ,$$

which shows that $h_\rho \in \mathcal{H}_\rho$ since

$$\|h_\rho\|_{\mathcal{H}_\rho}^2 \le \int c(\rho u)^2 \gamma_{\tilde\tau}(u) du = \int c(u)^2 \gamma_\tau(u) du = \|h^*\|_{\mathcal{H}}^2 < \infty .$$

Therefore, for any $\rho < 1$ and target $f$ satisfying the source condition $f = U_\rho f_0$, we have

$$A(f, \lambda, \rho) \le \|f - U_\rho h^*\|_\gamma^2 + \lambda \|U_\rho h^*\|_{\mathcal{H}_\rho}^2$$

$$\le \|U_\rho(f_0 - h^*)\|_\gamma^2 + \lambda \|h^*\|_{\mathcal{H}}^2$$

$$\le A(f_0, \lambda) ,$$

where we used the fact that $U_\rho$ is a contraction in $L^2(\gamma)$ for any $\rho \le 1$ [67, Theorem 11.23].

Let us now consider a general $f \in L^2(\gamma)$.

$$A(f, \lambda, \rho) = \inf_{h \in \mathcal{H}_\rho} \|f - h\|_\gamma^2 + \lambda \|h\|_{\mathcal{H}_\rho}^2$$

$$\le 2 \inf_{g \in \mathcal{F}} \left( \inf_{h \in \mathcal{H}_\rho} \|f - U_\rho g\|_\gamma^2 + \|U_\rho g - h\|_\gamma^2 + \lambda \|h\|_{\mathcal{H}_\rho}^2 \right)$$

$$\le 2 \inf_{g \in \mathcal{F}} \left( \|f - U_\rho g\|_\gamma^2 + \left( \inf_{h \in \mathcal{H}_\rho} \|U_\rho g - h\|_\gamma^2 + \lambda \|h\|_{\mathcal{H}_\rho}^2 \right) \right)$$

$$= 2 \inf_{g \in \mathcal{F}} \|f - U_\rho g\|_\gamma^2 + 2 A(U_\rho g, \lambda, \rho)$$

$$\le 2 \inf_{g \in \mathcal{F}} \|f - U_\rho g\|_\gamma^2 + 2 A(g, \lambda)$$

$$\le 2 \inf_{g \in \mathcal{F}} \|f - U_\rho g\|_\gamma^2 + C\lambda^\beta (\|g\|_\gamma^2 + \|g''\|_4^2)$$

$$\le \max\{2, C\} \tau^{1+\beta} \cdot B(f, \lambda^\beta, \rho) .$$

where we used $A(g, \lambda) \le C\tau^{1+\beta} \lambda^\beta (\|g\|_\gamma^2 + \|g''\|_4^2)$, where $C > 0$ is a universal constant satisfying Lemma 4.4, $\|g'\|_\gamma^2 \le \|g\|_\gamma^2 + \|g''\|_\gamma^2$, and $\|\cdot\|_\gamma \le \|\cdot\|_4$, which follows from Jensen's inequality. $\quad\square$

Proposition G.3 shows that approximation properties can be transferred from $\mathcal{F}$ to $\mathcal{H}_\rho$ for target functions satisfying a certain smoothness property, which is encoded in the source condition $B(f, \lambda, \rho)$. The choice of $U_\rho$ as the smoothing operator is motivated by its rich structure in $L_2(\gamma)$, in particular its (hyper-)contractivity. The source condition (98) can be explicitly controlled using the Hermite decomposition of $f$, though the $L^4(\gamma)$-norm penalty on the (weak) second derivative of the approximant $g \in \mathcal{F}$ imposes restrictions on the decay of its Hermite coefficients. We leave such analysis for future work.

Besides the RKHS approximation error, our results also require control of approximation error from using random features (Lemma C.3). We verify that the same argument (contained in Lemma C.4) can be directly applied to $\mathcal{H}_\rho$, leading to an analogous control in terms of degrees of freedom. That being said, one may be able to obtain better control of the degrees of freedom under smoothness, leading to smaller estimation error of the KRR estimator, which in general compensates for the worse approximation error via tuning the regularisation parameter $\lambda$ [7, Chapter 7].

**Optimization properties.** Using a smooth activation function for the student network simplifies the analysis of the empirical optimization landscape since we can readily adapt the tools developed in [60]. Moreover, since the empirical loss becomes a smooth function with Lipschitz gradients, our gradient flow analysis can be discretized and thereby yield guarantees for gradient *descent*. We now verify that for any $\rho \in (0, 1)$, $\phi'_\rho$ is $L$-Lipschitz with $L \leq \sup_t |\phi''_\rho(t)|$ which we now compute.

**Claim G.4** (Lipschitz constant of $\phi''_\rho$). *Let $\rho \in [0, 1)$ and let $\phi_\rho$ be the $\rho$-smoothed ReLU. Then,*

$$\sup_{t \in \mathbb{R}} |\phi''_\rho(t)| \leq \frac{4\rho^2}{\sqrt{1 - \rho^2}} \ .$$

*Proof.*

$$\phi_\rho(t) = \int \phi(\rho t + \sqrt{1 - \rho^2}u)\gamma(u)du$$

$$= \frac{1}{\sqrt{1 - \rho^2}} \int \phi(v)\gamma\left(\frac{v - \rho t}{\sqrt{1 - \rho^2}}\right) dv$$

$$= \frac{1}{\sqrt{1 - \rho^2}} \int_0^\infty v \cdot \gamma\left(\frac{v - \rho t}{\sqrt{1 - \rho^2}}\right) dv \ ,$$

using change of variables with $v = \rho t + \sqrt{1 - \rho^2}u$. Hence,

$$\phi''_\rho(t) = \frac{\rho^2}{(1 - \rho^2)^{3/2}} \int_0^\infty v \cdot \gamma''\left(\frac{v - \rho t}{\sqrt{1 - \rho^2}}\right) dv$$

$$= \frac{\rho^2}{1 - \rho^2} \int_{-\frac{\rho t}{\sqrt{1-\rho^2}}}^\infty (\rho t + \sqrt{1 - \rho^2}u)\gamma''(u)du$$

$$= \frac{\rho^2}{1 - \rho^2} \left( (\rho t + \sqrt{1 - \rho^2}u)\gamma'(u)|_{-\frac{\rho t}{\sqrt{1-\rho^2}}}^\infty - \sqrt{1 - \rho^2} \int_{-\frac{\rho t}{\sqrt{1-\rho^2}}}^\infty \gamma'(u)du \right)$$

$$= -\frac{\rho^2}{\sqrt{1 - \rho^2}} \int_{-\frac{\rho t}{\sqrt{1-\rho^2}}}^\infty \gamma'(u)du \ ,$$

Thus,

$$|\phi''_\rho(t)| \leq \frac{\rho^2}{\sqrt{1 - \rho^2}} \int_{-\infty}^\infty |\gamma'(u)|du$$

$$= \frac{\rho^2}{\sqrt{1 - \rho^2}} \int_{-\infty}^\infty |u|\gamma(u)du$$

$$= \frac{2\rho^2}{\sqrt{1 - \rho^2}} \left( \int_0^1 u\gamma(u)du + \int_{u \geq 1} u\gamma(u)du \right)$$

$$\leq \frac{2\rho^2}{\sqrt{1 - \rho^2}} \left( 1 - \gamma(1) + \int_{-\infty}^\infty u^2\gamma(u)du \right)$$

$$\leq \frac{2\rho^2}{\sqrt{1 - \rho^2}} \cdot (2 - \gamma(1)) \ .$$

$\square$

## H RKHS Approximation Beyond $\mathcal{F}$

We further discuss the approximation capability of the RKHS $\mathcal{H}$. Recall that Lemma 4.4 states that functions in $\mathcal{F} = \{f \in H^2(\gamma) \mid f'' \in L^4(\gamma)\}$ can be approximated by functions in $\mathcal{H}$ at

a polynomial-in-$\lambda$ rate. We show that ReLU as the *target* function admits a polynomial-in-$\lambda$ approximation rate (Proposition H.1). Note that $\text{ReLU} \notin \mathcal{F}$ since ReLU is not even in $H^2(\gamma)$. Thus, Proposition H.1 demonstrates that containment in $\mathcal{F}$ (Assumption 5.2) is sufficient, but not necessary for polynomial approximation rates.

**Proposition H.1** (RKHS approximation error for ReLU). *Let $\tau > 1$ and let $\phi(t) = \max(0, t)$. Then, for any $\lambda \in (0, \lambda^*)$, where $\lambda^* < 1$ depends only on $\tau$,*

$$A(\phi, \lambda) \leq (2 + \tau^2) \cdot \lambda^{2/3} .$$

*Proof.* We directly upper bound $A(\phi, \lambda)$ by the one-parameter family of functions $\phi_\rho = U_\rho \phi$, where we recall that $U_\rho$ is the Ornstein-Uhlenbeck operator. Define $\lambda^* \in (0, 1)$ by

$$\lambda^* = (1 - \sqrt{2/(2\tau^2 + 1)})^{3/2} .$$

We consider approximants $\phi_\rho$ such that $\rho > \sqrt{2/(2\tau^2 + 1)}$, which in turn satisfies $1 - \rho < (\lambda^*)^{2/3}$. We first show that for $\rho$ sufficiently close to 1, $\phi_\rho$ approximates $\phi$ well in $L^2(\gamma)$. Then, we show that $\phi_\rho \in \mathcal{H}$ for $\rho > \sqrt{1/(2\tau^2 + 1)}$, and further show that $\|\phi_\rho\|_\mathcal{H}$ is roughly upper bounded by $1/\sqrt{1 - \rho^2}$. From Corollary H.4, we know that the Hermite expansion of $\phi$ yields $\phi = \sum_j \alpha_j h_j$ with $|\alpha_j| \leq j^{-5/4}$ for $j \geq 2$. Since Hermite polynomials are eigenfunctions of the operator $U_\rho$, we immediately have

$$\|\phi - \phi_\rho\|_\gamma^2 = \sum_{j=1}^\infty |\alpha_j|^2 (1 - \rho^j)^2 \leq \sum_{j=1}^\infty |\alpha_j|^2 (1 - \rho^j)$$

$$= (1 - \rho) \sum_{j=1}^\infty |\alpha_j|^2 (1 + \cdots + \rho^{j-1})$$

$$\leq (1 - \rho) \sum_{j=1}^\infty j |\alpha_j|^2$$

$$\leq (1 - \rho) \sum_{j=1}^\infty j^{-3/2}$$

$$\leq (1 - \rho)\left(1 + \int_1^\infty j^{-3/2}\right)$$

$$\leq 2(1 - \rho) .$$

On the other hand, by definition and change-of-variables, we have

$$\phi_\rho(t) = \int \phi(\rho t + \sqrt{1 - \rho^2} u)\gamma(u)du$$

$$= \int \rho \cdot \phi\left(t + \frac{\sqrt{1 - \rho^2}}{\rho} u\right)\gamma(u)du$$

$$= \frac{\rho^2}{\sqrt{1 - \rho^2}} \int \phi(t + b)\gamma\left(\frac{\rho}{\sqrt{1 - \rho^2}} \cdot b\right)db$$

$$= \int \phi(t + b)c(b)\gamma_\tau(b)db ,$$

where

$$c(b) = \frac{\rho^2 \tau}{\sqrt{1 - \rho^2}} \cdot \exp\left(-\frac{1}{2}\left(\frac{\rho^2}{1 - \rho^2} - \frac{1}{\tau^2}\right)b^2\right) .$$

We thus have

$$\|\phi_\rho\|_{\mathcal{H}}^2 \leq \|c\|_{\gamma_\tau}^2$$

$$= \frac{\rho^4\tau}{\sqrt{2\pi}(1-\rho^2)} \int \exp\left(-\left(\frac{\rho^2}{1-\rho^2} - \frac{1}{2\tau^2}\right)b^2\right) db$$

$$= \frac{\rho^4\tau}{\sqrt{2\pi}(1-\rho^2)} \int \exp\left(-\frac{(1+2\tau^2)\rho^2-1}{2\tau^2(1-\rho^2)}b^2\right) db$$

$$= \frac{\rho^4\tau}{\sqrt{2\pi}(1-\rho^2)} \cdot \frac{\tau\sqrt{2\pi(1-\rho^2)}}{\sqrt{(1+2\tau^2)\rho^2-1}}$$

$$= \frac{\rho^4\tau^2}{\sqrt{1-\rho^2}\sqrt{(1+2\tau^2)\rho^2-1}} \,,$$

which implies that $\phi_\rho \in \mathcal{H}$ for any $\rho > 1/\sqrt{1+2\tau^2}$.

We balance the upper bounds of $\|\phi - \phi_\rho\|_\gamma^2$ and $\lambda\|\phi_\rho\|_{\mathcal{H}}^2$ to control $A(\phi, \lambda)$ in terms of $\lambda$. To this end, we set $\rho = 1 - \lambda^{2/3}$, where $\lambda < \lambda^*$. Then, we have

$$\|\phi - \phi_\rho\|_\gamma^2 \leq 2(1-\rho) = 2\lambda^{2/3} \,,$$

Moreover, using the fact that $1 - \rho^2 = (1-\rho)(1+\rho) \geq 1 - \rho$ and $(1+2\tau^2)\rho^2 > 2$, which follows from $\rho > 1 - (\lambda^*)^{2/3}$, we get

$$\lambda\|\phi_\rho\|_{\mathcal{H}}^2 \leq \lambda \cdot \frac{(1-\lambda^{2/3})^4\tau^2}{\lambda^{1/3}\sqrt{(1+2\tau^2)\rho^2-1}}$$

$$\leq \lambda^{2/3} \cdot \tau^2 \,.$$

Hence, for any $\lambda \in (0, \lambda^*)$,

$$A(\phi, \lambda) \leq \|\phi - \phi_\rho\|_\gamma^2 + \lambda\|\phi_\rho\|_{\mathcal{H}}^2 \leq (2 + \tau^2) \cdot \lambda^{2/3} \,.$$

$\square$

## H.1  ReLU Hermite coefficients

**Fact H.2.** *Let $\{H_j(z)\}_{j\in\mathbb{N}}$ be the unnormalized (probabilist's) Hermite polynomials. Then,*

$$H_j(0) = \begin{cases} 0 & \text{if } j \text{ odd} \\ (-1)^{j/2}\frac{j!}{(j/2)!2^{j/2}} & \text{if } j \text{ even} \,. \end{cases}$$

**Claim H.3** (ReLU Hermite coefficients [42, Claim 1])**.** *The Hermite coefficients of $\mathrm{ReLU}(z)$ are given by*

$$\alpha_j = \begin{cases} 1/\sqrt{2\pi} & \text{if } j = 0 \\ 1/2 & \text{if } j = 1 \\ \frac{1}{\sqrt{2\pi j!}}(H_j(0) + jH_{j-2}(0)) & \text{otherwise} \,. \end{cases}$$

**Corollary H.4** (ReLU coefficient bounds for $j \geq 2$)**.**

$$\alpha_j = \frac{1}{\sqrt{2\pi j!}} \cdot (-1)^{(j-2)/2}\frac{(j-2)!j}{(j/2)!2^{j/2}} \quad \text{and} \quad |\alpha_j| \leq \frac{1}{\sqrt{2\pi^{3/2}}} \cdot \frac{1}{j^{5/4}} \,.$$

*Proof.* Combining Fact H.2 and Claim H.3,

$$H_j(0) + jH_{j-2}(0) = (-1)^{j/2} \left( \frac{j!}{(j/2)!2^{j/2}} - \frac{(j-2)!2j(j/2)}{(j/2)!2^{j/2}} \right)$$

$$= (-1)^{j/2} \frac{(j-2)!}{(j/2)!2^{j/2}} \left( j(j-1) - j^2 \right)$$

$$= (-1)^{(j-2)/2} \frac{(j-2)!j}{(j/2)!2^{j/2}}$$

$$\alpha_j = \frac{1}{\sqrt{2\pi j!}} \cdot (-1)^{(j-2)/2} \frac{(j-2)!j}{(j/2)!2^{j/2}}$$

$$= \frac{(-1)^{(j-2)/2}}{\sqrt{2\pi}} \cdot \frac{\sqrt{(j-2)!}\sqrt{j}}{\sqrt{j-1}(j/2)!2^{j/2}}$$

$$= \frac{(-1)^{(j-2)/2}}{\sqrt{2\pi}} \cdot \frac{\sqrt{j}\sqrt{(j-2)!}}{\sqrt{j-1}j!!} ,$$

where we used the fact that $j!! = (j/2)!2^{j/2}$ for even $j$ in the last line. It remains to evaluate the RHS. We use the following facts on double factorials.

$$\frac{j!!}{(j-1)!!} = \frac{2^{j/2}(j/2)!}{j!/(2^{j/2}(j/2)!)} = 2^j \binom{j}{j/2}^{-1} \le \sqrt{\pi(j+1)/2} \le \sqrt{\pi j} \tag{99}$$

$$(j-2)! = (j-2)!!(j-3)!! = \frac{(j-1)!!j!!}{j(j-1)} \le \frac{(j!!)^2}{j(j-1)\sqrt{\pi j}} ,$$

where in Eq. (99), we used the fact that $\binom{2k}{k} \ge 4^k/\sqrt{\pi(k+1/2)}$. Thus,

$$|\alpha_j| \le \frac{1}{\sqrt{2\pi}} \cdot \frac{1}{(j-1)(\pi j)^{1/4}}$$

$$\le \frac{1}{\sqrt{2\pi^{3/2}}} \cdot \frac{1}{j^{5/4}} .$$

□