# OpenReview forum: "Learning single-index models with shallow neural networks"
_NeurIPS.cc/2022/Conference — NeurIPS 2022 Accept_

### Official Review · Reviewer_9Zca · 2022-07-04

**Rating:** 8
**Confidence:** 4
**Soundness:** 4 excellent
**Presentation:** 4 excellent
**Contribution:** 4 excellent

**Summary:**

In this paper, the authors study the problem of learning a single index model using a specific class of two-layer neural networks. The network structure is nonstandard, because its bias terms are random initialized and fixed (not trained), and weights are tied across all neurons. The first main result is a population landscape theorem, which states that the first-order critical points are either in the poles ($\pm \theta^*$) or on the equator ($\langle \theta, \theta^* \rangle = 0$). Building on this result, the authors then study the empirical loss. They prove that the (nonstandard) gradient flow converges to a solution with desired generalization error.

**Questions:**

- Q1: In the statement of Theorem 5.1, the probability is "greater than $1/4 - 2\delta$". Is it possible to strengthen this to a high probability statement? If not, why?
- Q2: The authors mention that they student the critical points of $L(c, \theta)$ through projected population loss. Based on that, I expected that during the initial warmup stage ($t \le T_0$) one trains $c$ first, but Eqn. (11) is the contrary.
- Q3: I do not understand why we cannot achieve the optimal generalization error without fine-tuning. Is this an artifact of the proof? If so, where is the bottleneck and how fine-tuning improves the rate? If not, what is the implication for the practice?

Some typos:
- Line 119: $\varphi_{b_i}$ not $\varphi_{b_j}$
- Line 133: "first the study" --> "first study"

**Limitations:**

An evident limitation is that the model is rather simplified, but I believe this is on par with the current level of neural network theories. There are tweaks such as certain nonstandard $\zeta$ scheduling, and these limitations are discussed in the paper.

**Strengths And Weaknesses:**

Overall, I find this paper very clear and solid. The model under study is simplified but more complicated than existing ones (i.e., random features model). While the proof strategy is mostly built upon existing ideas, the new challenge is the interaction between parameters $c$ and $\theta$. Below is a more detailed assessment.

**Pros**:
- The landscape result is very clear. It shows that under essentially minimal overparametrization the critical points are either the true directions or orthogonal to the true directions.
- It is proved that gradient flow (with certain warmup and fine-tuning) achieves the optimal generation error under optimal sample complexity $n \gtrsim \Omega(d^s)$.
- The proof seems to be a nice combination of the landscape analysis that is popular for analyzing many nonconvex problems and gradient flow analysis for lazy training.

**Cons**:
- The neural network structure is rather simplified. While I do not complain about useful simplification (which is standard in the literature), the title and abstract are rather misleading; for instance, it is not mentioned that the weights are tied, which is crucial for this work.
- It is not clear whether certain tweaks such as warmup and fine-tuning are necessary in practice. Perhaps the authors should discuss how these tweaks would affect the practical aspect of training neural nets.
- Some further questions about theorem statement (see below).

---

> ### Author Response · Authors · 2022-08-02
> **Author Response**
>
> ***“Overall, I find this paper very clear and solid ... While the proof strategy is mostly built upon existing ideas, the new challenge is the interaction between parameters $c$ and $\theta$ … The proof seems to be a nice combination of the landscape analysis that is popular for analyzing many nonconvex problems and gradient flow analysis for lazy training.“***
>
> We thank the reviewer for a thorough, insightful, and positive review. We particularly appreciate the detailed questions on our theorem statements. These questions give us the opportunity to further discuss some interesting technical aspects of our analysis.
>
> ***“The title and abstract are rather misleading; for instance, it is not mentioned that the weights are tied, which is crucial for this work.”***
>
> This is a valid point. We described the tied weights structure more explicitly in the revised abstract, which hopefully provides a more accurate description of our contribution.
>
> ***“It is not clear whether certain tweaks such as warmup and fine-tuning are necessary in practice. Perhaps the authors should discuss how these tweaks would affect the practical aspect of training neural nets.”***
>
> The warm-up and fine-tuning phases might indeed be an artifact of our theory. Nevertheless, in our experiments we found that having a large relative step-size on theta was crucial for successful optimization, and that a final closed-form fine-tuning of the second-layer weights c can also significantly improve the loss compared to gradient descent, likely due to poor conditioning.The situation could be quite different for neural network training in practice, where there is no spherical constraint on the first layer weights, and common optimizers, such as Adam, often use adaptive step-sizes for different parameters. We also note that operations similar to our fine-tuning step have been found to be beneficial empirically [Moreau & Audiffren, 2017].
>
> ***“Q1: In the statement of Theorem 5.1, the probability is "greater than $1/4-2\delta$". Is it possible to strengthen this to a high probability statement? If not, why?”***
>
> This is indeed an important question. We thank the reviewer for bringing this up. First, note that we improved this to 1/2 - delta in our updated manuscript. Improving to a probability close to 1 might be difficult in this case, since if we’re on the “wrong side of the equator”, then crossing it might take a very long time. A recent paper by Ben Arous et al. (2022) shows that these probabilities can be even smaller in certain setups. For instance, it is 3/32 for the (Gaussian) XOR problem using 2-layer networks with 4 neurons. We suspect that having multiple neuron directions instead of a single one could improve these probabilities, but leave this question to future work.
>
> ***“Q2: The authors mention that they student the critical points of $L(c,\theta)$  through projected population loss. Based on that, I expected that during the initial warmup stage ($t \le T_0$) one trains $c$ first, but Eqn. (11) is the contrary.”***
>
> Note that we use the projected population loss simply as an analytical tool for characterizing the critical points of the *population* loss L(c,theta). While optimizing c first might work as well, it can be problematic for the following reason: at initialization the correlation m is small, and the population loss scales with m^{2s} for an optimal c, instead of m^s for a randomly initialized c. These “curvatures” determine the sample complexity needed to escape the initial saddle point, and suggest that the latter strategy should be preferred (indeed, we require O(d^s) samples, while optimizing c first would increase this to O(d^{2s})).
>
> ***“Q3: I do not understand why we cannot achieve the optimal generalization error without fine-tuning. Is this an artifact of the proof? If so, where is the bottleneck and how fine-tuning improves the rate? If not, what is the implication for the practice?”***
>
> This is also a good question. Fine-tuning allows us to improve our analysis in two ways: (i) it removes dependence of theta on the previous data by using fresh samples, and (ii) uses a different regularization parameter lambda for the two phases. While (i) may be just a technical convenience, overcoming (ii) seems difficult, since ridge regression might require a different, possibly very small lambda when the sample size is large, while the first phase (for estimating the hidden direction theta) could suffer from having a small lambda. At least with our current empirical landscape analysis based on uniform gradient concentration, the worst-case deviations degrade for smaller lambda. This is why the rates are slower in our Corollary 5.2 compared to Corollary 5.4. We will clarify this in our updated version.
>
> ### References
> - T. Moreau, J. Audiffren. *Post Training in Deep Learning with Last Kernel*. 2017
> - G. Ben Arous, R. Gheissari, A. Jagannath. *High-dimensional limit theorems for SGD: Effective dynamics and critical scaling*. 2022

---

### Official Review · Reviewer_WRiK · 2022-07-09

**Rating:** 7
**Confidence:** 3
**Soundness:** 4 excellent
**Presentation:** 4 excellent
**Contribution:** 3 good

**Summary:**

The paper's focus is the learning of single-index models, in which the label $y$ of a datapoint $x\sim\mathcal N(0,I_d)\in\mathbb R^d$ is obtained as $y=f_\star(\langle \theta_\star,x\rangle)$, i.e., by projecting the data in a direction $\theta^\star\in\mathbb S^{d-1}$ and then feeding the result in a "link function" $f_\star\colon\mathbb R\to\mathbb R$. Both $\theta^\star$ and $f_\star$ have to be learned. The authors consider a shallow neural network of the type $$G(x;c,\theta)=\frac{1}{\sqrt N}\sum_{i=1}^Nc_i\phi(\langle\theta,x\rangle-b_i),\qquad \phi(u)=\max(0,u)$$
with frozen biases $b_i$ and depending on $c\in\mathbb R^N$ and $\theta\in\mathbb R^d$. They show that, in the considered setting, the critical points of the population landscape can be fully characterised: for example, for sufficiently small regularisation and sufficiently large width of the network, the critical points $\hat\theta$ of the population loss are such that $\langle\hat\theta,\theta^\star\rangle\in\\{-1,0,1\\}$.  They also provide a bound for the random feature approximation error in terms of the RKHS error.

The paper moves then to analyze the empirical landscape, extracting the sample complexity scaling required to escape from the uninformative equator $\\{\theta\colon \langle \theta,\theta^\star\rangle=0\\}$, showing that the gradient flow on the empirical loss finds an approximate minimizer for a number of samples $n=O(d^s)$, where $s$, called information exponent, is related to the Hermite polynomial expansion of the target function $f_\star$. The theoretical understanding of the (empirical) landscape properties leads to the proposal of a gradient descent algorithm for $c$ and $\theta$, whose flow efficiently finds an approximate minimizer of the population loss.

**Questions:**

As the authors themselves pointed out, and as mentioned above, the considered model might look somehow rigid in its parametrization. I would like the authors to further comment on the robustness of their results. In particular, the Gaussian dataset leads the authors to choose Hermite polynomials as a basis of their decomposition and, e.g., the expansion of $f_\star$ on this basis determines relevant quantities, such as the information exponent. The authors might further comment on what they expect in the case of different data set distributions, maybe expanding the discussion on lines 157-163.

If compatible with the length restrictions, I think it might be useful to add to the paper a brief section discussing the implementation of the algorithm (e.g., moving the material in the Appendix in the main text) and some details on the numerical results.

*Minor typographical comments.*
- On page 3 (and in many other points, in particular in Secion 5) $F^*$ is sometimes called $F_{\theta^*}$ (see Definition 3).
- On page 4, line 119 has $\phi_{b_j}$ in place of $\phi_{b_i}$. Also, I believe that the coefficients $\alpha_j$ missing are missing in the sum. In line 133, "we first the study optimization" > "we first study the optimization".
- On page 5, in Remark 4.4, $\hat P_\lambda$ misses a hat.
- On page 6 line 206 subscript d missing on $I_d$.
- On page 7 in Theorem 5.1, it is maybe more convenient to write "Algorithm 1" in place of "procedure 1" for reasons of clarity (similarly at a later stage in line 281 Procedure 1 refers to the box named Algorithm 1). Also, there is an extra parenthesis in eq. 12 and, in line 241, an exponent 'outside' the O-notation.
- In Appendix eq 15, $\sigma$ should probably be replaced by $\phi$.

*Additional references*
The authors might consider adding the reference
V Ros, G Ben Arous, G Biroli, C Cammarota, Physical Review X 9 (1), 011003 (2019)
in line 86, as in this paper the quenched entropy of the spiked tensor model is computed, complementing the rigorous results in [4].

**Limitations:**

The paper is of theoretical nature and there are no potential negative societal effects and/or limitations of a sort in this respect. The weaknesses of the contributions are discussed above.

**Strengths And Weaknesses:**

*Originality* The considered setting can be seen as a special one within the large literature on single- and multi-index models that the authors properly cite in the paper; the landscape analysis is very inspired by the one presented in [19] (where a similar landscape topology is found, as acknowledged by the authors). All results are supported by rigorous analysis built upon the available literature on random features and kernel methods. To my understanding, the originality of the paper consists in constructing a simple and completely characterizable example of a learning task performed by training both layers of a shallow network using gradient descent in a nonconvex yet "benign" landscape.

*Quality* All results are supported by rigorous analysis and precise analytical bounds to the performances are provided. On the other hand, the proposed algorithm is not explored extensively from the numerical point of view (very few numerical experiments are given in the Appendix, and without implementation of the second fine-tuning step which is discussed in the main text).

*Clarity* The paper is clearly written and proceeds in sequential order in studying the population landscape and the empirical landscape, up to the proposal of an efficient gradient descent algorithm for the solution of the task.

*Significance* The main result of the paper is the full characterisation of the performances of a shallow neural network architecture (of a specific type) in learning a single-index model by means of gradient descent. The main significance comes, in my understanding, from the hybrid nature of the learning task (aiming at obtaining both $\theta$ and, non-parametrically, the link function) via a gradient descent scheme whose performances are clarified in terms of the benign landscape. As somehow the authors themselves stress in the conclusion, a weakness of the contribution is the rigidity of the investigated setting: the architecture consists of a shallow neural network whose hidden layer consists of $N$ ReLU neurons parametrised by the same $\theta$, differing only by their quenched bias. Also, assumptions on the dataset seem to be crucial for the analysis (see below). These assumptions might look a little bit restrictive/unrealistic, and it is not very clear to me how much the results can be considered robust: in particular, I wonder how much the simple properties of the landscape, which are pivotal for the results, are due to this combination of assumptions. Despite these observations, I personally found the paper interesting and possibly the first step for the study of more complex set-ups to investigate both the breaking down and the persistence/generalization of the phenomenology observed here.

---

> ### Author Response · Authors · 2022-08-02
> **Author Response**
>
> ***“The main significance comes, in my understanding, from the hybrid nature of the learning task (aiming at obtaining both  and, non-parametrically, the link function) via a gradient descent scheme whose performances are clarified in terms of the benign landscape … I personally found the paper interesting and possibly the first step for the study of more complex set-ups to investigate both the breaking down and the persistence/generalization of the phenomenology observed here.”***
>
> We thank the reviewer for the positive feedback, and a detailed and insightful review.
>
> ***“I would like the authors to further comment on the robustness of their results. In particular, the Gaussian dataset leads the authors to choose Hermite polynomials as a basis of their decomposition and, e.g., the expansion of  on this basis determines relevant quantities, such as the information exponent. The authors might further comment on what they expect in the case of different data set distributions, maybe expanding the discussion on lines 157-163.”***
>
> We thank the reviewer for this very interesting remark. Indeed, the Gaussian distribution leads to a simple population landscape where the monomial coefficients can be simply obtained from the Hermite coefficients. With other distributions, one could have different landscapes which may be more difficult to analyze. For instance, for the uniform distribution on the sphere, one would obtain a description in terms of Legendre polynomials in the correlation m = <theta, theta*>, which are more difficult to analyze. That said, gradient flow on a shallow neural network is a procedure that is agnostic to the data distribution (although its analysis depends on it). Moreover, one could argue that the topological properties of the optimization landscape are ‘stable’ with respect to small perturbations of the input distribution. Therefore, we expect that data distributions that are close (in some appropriate sense) to the standard Gaussian would have similar guarantees. This is definitely an interesting future direction.
>
> ***“If compatible with the length restrictions, I think it might be useful to add to the paper a brief section discussing the implementation of the algorithm (e.g., moving the material in the Appendix in the main text) and some details on the numerical results.”***
>
> We updated our experiments section (Appendix D) to include more details about implementation and numerical simulations. We also have new plots showing the empirical benefit of the fine-tuning step. We will move the experiments section to the main text before the conclusion when an additional content page is allowed.
>
> ***“Minor typographical comments” and “Additional references”***
>
> We thank the reviewer for these detailed suggestions. We have incorporated these suggestions into our updated version.

---

### Official Review · Reviewer_z2Sa · 2022-07-11

**Rating:** 7
**Confidence:** 4
**Soundness:** 4 excellent
**Presentation:** 3 good
**Contribution:** 3 good

**Summary:**

This paper studies the problem of learning a single-index function with a shallow network trained via gradient descent. The function to be learned is of the form $f_{\star}\left(\theta^{\star}\cdot x\right)$, where both $f_{\star}:\mathbb{R}\rightarrow\mathbb{R}$ and $\theta^{\star}\in\mathbb{S}^{d-1}$ are unknown, and the data $x$ is Gaussian. The authors establish that the training landscape of this problem has no bad local minima, and then prove that $(f_{\star}, \theta^{\star})$ can be learned efficiently, modulo conditions on the width of the network and a two-part structure during training, where only $\theta$ is learned during the first part.

**Questions:**

I’m curious why the authors feel that the teacher-student terminology is necessary in this paper? Can $F_{\theta^{\star}}$ not simply be referred to as ‘ground truth’ or the ‘true’ data generating process, as is standard in learning theory? If they can, then I suggest eliminating nonstandard terminology.

I think it would be helpful to the reader to explicitly state the following implications of Theorem 4.2 for the learning problem right after the theorem is presented: (1) all minima are global minima and (2) gradient descent will have to either avoid or escape the minima corresponding to $\langle\theta^{\star}, \theta\rangle=0$.

A little more discussion of $T_0$ and $T_1$ in the main text would also be useful. At present, $T_1$ appears in the main text for the first time in Algorithm 1, and I think there isn’t enough information in the main text about it to understand the algorithm. With regards to $T_0$, perhaps the logical place to first introduce it might be line 234? This will help the reader really understand the meaning of $\zeta(t)$, as well, without first having to go through the proofs. This will also make it clear \textit{before} the math is presented that the algorithm constructed by the authors has two different learning phases.

**A few minor things:**

Typo in line 35. Remove the word ‘in’ between ‘incurring’ and ‘a’.

Line 116. Maybe insert the word ‘between’ between ‘relationship’ and ‘each’?

The first time the acronym RKHS (line 99?) is used, its full form should be given.

A little bit of attention needs to be paid to capitalization in the references. For example, in line 311, ‘sgd’ should be ‘SGD’, and so on.

**Limitations:**

Yes.

**Strengths And Weaknesses:**

This is a very nice paper. It provides a positive answer to the (previously open) question of whether or not a shallow neural net can learn single-index functions via a constructive proof. The result contributes to our knowledge of the class of functions that can be learned efficiently with shallow networks and is therefore relevant to a broad machine learning audience. The paper is clearly written.

I do not find significant weaknesses.

---

> ### Author Response · Authors · 2022-08-02
> **Author Response**
>
> ***“This is a very nice paper. It provides a positive answer to the (previously open) question of whether or not a shallow neural net can learn single-index functions via a constructive proof. The result contributes to our knowledge of the class of functions that can be learned efficiently with shallow networks and is therefore relevant to a broad machine learning audience. The paper is clearly written.”***
>
> We thank the reviewer for the positive feedback and a detailed review of our paper.
>
> ***“I’m curious why the authors feel that the teacher-student terminology is necessary in this paper? Can  not simply be referred to as ‘ground truth’ or the ‘true’ data generating process, as is standard in learning theory? If they can, then I suggest eliminating nonstandard terminology.”***
>
> Indeed, the mixture of terminology from different fields can be confusing. We have changed all occurrences of the term “teacher” to “target function” and removed other uses of the “teacher-student” terminology.
>
> ***“I think it would be helpful to the reader to explicitly state the following implications of Theorem 4.2 for the learning problem right after the theorem is presented: (1) all minima are global minima and (2) gradient descent will have to either avoid or escape the minima corresponding to $\langle \theta^{\*},\theta \rangle = 0$.”***
>
> We thank the reviewer for this helpful suggestion on our exposition. We added a discussion on the implications after Theorem 4.2 in our updated version.
>
> ***“A little more discussion of $T_0$ and $T_1$ in the main text would also be useful. At present,  $T_1$ appears in the main text for the first time in Algorithm 1, and I think there isn’t enough information in the main text about it to understand the algorithm. With regards to $T_0$, perhaps the logical place to first introduce it might be line 234?”***
>
> We thank the reviewer for this suggestion. We have updated our paper to include more details about the times T0 and T1 and the choice of zeta(t) in our updated version.
>
> ***“A few minor things: …”***
> Thanks for pointing out the typos in our initial submission. we have addressed them all in our updated version.

---

> > ### Comment · Reviewer_z2Sa · 2022-08-08
> > **Re: author response**
> >
> > I thank the reviewers for their response and for engaging with my suggestions with their edits to the paper.
> >
> > Re: my first suggestion (about referring to $f_{\star}$ as a teacher), I realize it is possible it came across as stronger than intended, so I'd like to stress that I believe you should stick to whatever framing seems clearest to you. Please see the exchange with reviewer uZAC above for context.

---

> ### Comment · Reviewer_uZAC · 2022-08-02
> **Nonstandard?**
>
> > *I’m curious why the authors feel that the teacher-student terminology is necessary in this paper? Can $F_{\theta_{\star}}$ not simply be referred to as ‘ground truth’ or the ‘true’ data generating process, as is standard in learning theory? If they can, then I suggest eliminating nonstandard terminology.*
>
> With all due respect to the reviewer, I am frankly surprised by this comment.
>
> First, let me remember the reviewer that although the "teacher-student" terminology might be less standard in statistical learning theory, it is **very** standard in statistical physics of learning, which is one of the "Theory" subject areas in NeuRIPS [1]. Indeed, this terminology has been part of NeurIPS since the *very beginning* of the conference [2].
>
> Second, the reviewer seems to fully understand the meaning of the terminology employed by the authors. At this point, the choice of terminology should be personal, and it is innapropriate for a reviewer to *"suggest eliminating nonstandard terminology"* when it is really not. Do you think it would be ok for me to encourage the authors to change "single-index" for "generalized linear model" or "perceptron", for example?
>
> NeurIPS has always praised itself as a venue fostering scientific diversity, where mathematicians, statisticians, computer scientistis, engineers, neuroscientists, physicists, information theorists (and many others!) meet to share their unique point of view on machine learning and neural networks. Let's keep it this way!
>
> [1] https://nips.cc/Conferences/2020/PaperInformation/SubjectAreas
>
> [2] A Krogh, J Hertz, *"Dynamics of Generalization in Linear Perceptrons"*,  Part of Advances in Neural Information Processing Systems 3 (NIPS 1990).

---

> > ### Comment · Reviewer_z2Sa · 2022-08-03
> > **Re: Nonstandard?**
> >
> > Thank you for your comment.
> >
> > Indeed, I am familiar with the canonical teacher-student framework in statistical mechanics. It seemed to me that the particular setting in this paper referenced that framework only in a weak sense, and so it wasn’t clear what the authors had in mind when they alluded to teacher-student models. Hence my question to them. I do maintain, however, that if the authors decide that referencing the teacher-student framework adds terminology without clearly adding sufficient context or technical machinery to compensate, then, in the interests of clarity, it would be best practice not to do it.
> >
> > Re-reading the paper and my review, I can see I could have been clearer that this particular suggestion you have picked up on was made in the interests of improving the clarity of the paper, as were the two others. Still, I would point out that I did not make a blanket suggestion to change terminology. I wrote an if-then statement, and left it to the authors to evaluate the 'if' and then decide whether to execute on the 'then'. I think it is over the top to characterize that as 'inappropriate'.
> >
> > Ultimately, we agree that it is the prerogative of the authors to frame their work as they choose. My (high) score of this paper was not and is not contingent on whether or not the authors entertain any of my suggestions.

---

> > > ### Comment · Reviewer_uZAC · 2022-08-05
> > > **Re: Re: Nonstandard?**
> > >
> > > I thank the reviewer for clarifying her/his position. It is reassuring to hear that your comment was not meant as a blanket suggestion to change terminology, and that it did not influence your scientific judgement of the paper. Generally speaking, I believe this is a subtle matter given the position of power of reviewers with respect to authors.

---

### Official Review · Reviewer_uZAC · 2022-07-11

**Rating:** 6
**Confidence:** 2
**Soundness:** 3 good
**Presentation:** 3 good
**Contribution:** 3 good

**Summary:**

This manuscript studies the supervised problem of learning a single-index model with a proxy model for a two-layer neural network where the first layer weights are tied and the biases are random. Different from previous works on single-index models, the link function of the target single-index model are not matched with the network architecture. Both layers are trained. The key result is to show that gradient flow on the empirical loss can achieve good approximation error of the target. More specifically, the main theoretical results are:

1. To show that the critical points of the population loss are benign, i.e. that all the local minimisers are global and correspond to the target weights.

2.  To show that gradient flow on the empirical loss with a particular choice of relative time-scaling for the update of the first and second layer weights finds an approximate minimiser of the population loss. Although recovery is optimal with respect to the first layer weights, it is sub-optimal in terms of the target function.

3. To show that the optimal statistical sample complexity for the recovery of the target can be achieved by re-training the last layer on fresh samples.

**Questions:**

- **[Q1]**: Can the authors clarify why they believe the setting considered is natural when learning a single-index model (discussion in L38-L40)?

- **[Q2]**: Why the variance of the bias needs to be larger than one in the analysis (L94)? What would change in the analysis for $\tau<1$?

- **[Q3]**: Why the $\ell_2$ penalty parameter $\lambda$ is restricted to $(0,1)$ (L47)? Is this always satisfied by the optimal $\lambda$ leading to corollary 5.2?

- **[Q4]**: What role does the choice of $\phi = relu$ plays in the results? Would it be possible to derive similar guarantees for other activations?

**Limitations:**

The authors provide a nice discussion of the limitations of their work, as well as a list of possible improvements and future directions (L296-L308).

**Strengths And Weaknesses:**

This manuscript adds to the increasing literature of theoretical works showing the separation between neural networks and random features model and the benefits of overparametrisation for optimisation in non-convex landscapes [1, 43, VSL+22, SVL22, BES+22]. Although the setting considered is artificial in many ways (tied weights, random bias, delayed gradient flow), the fact that  gradient flow on this network can approximate "well" a mismatched single-index model with is an interesting result.

**Strengths**: Despite the density of technical details, the paper is well written and the main thread is easy to follow. The main result is significant and timely.

**Weaknesses**: The setting considered is artificial in many ways, and therefore it is hard to say it is directly relevant to our understanding of the approximation and optimisation properties of two-layer neural networks. Rather, I see it as a constrained first step with desirable extensions.

**References** (numbered refs. are from the bibliography in the paper)

[VSL+22], R. Veiga, L. Stephan, B. Loureiro, F. Krzakala, L. Zdeborová, "Phase diagram of Stochastic Gradient Descent in high-dimensional two-layer neural networks", arXiv: 2202.00293 [stat.ML]

[SVL22] I. Safran, G. Vardi, J.D. Lee, On the Effective Number of Linear Regions in Shallow Univariate ReLU Networks: Convergence Guarantees and Implicit Bias

[BES+22] J. Ba, M.A. Erdogdu, T. Suzuki, Z. Wang, D. Wu, G. Yang, "High-dimensional Asymptotics of Feature Learning: How One Gradient Step Improves the Representation", arXiv: 2205.01445 [stat.ML].

---

> ### Author Response · Authors · 2022-08-02
> **Author Response**
>
> ***“Despite the density of technical details, the paper is well written and the main thread is easy to follow. The main result is significant and timely.”***
>
> We thank the reviewer for the positive assessment of our results and bringing up great clarification questions. We particularly appreciate the detailed questions on why our proposed architecture is “natural” [Q1] and our assumptions [Q2,Q4]. These questions have given us the opportunity to further clarify the motivation behind our constrained NN architecture and technical assumptions.
>
> ***“Q1: Can the authors clarify why they believe the setting considered is natural when learning a single-index model (discussion in L38-L40)?”***
>
> Our starting point was to think of weights in the first layer as learning some useful features/representations of the data, while the biases may be useful to capture “offsets” for modeling 1D functions. The simplification of random biases and tied neurons are mostly for technical convenience: they are sufficient for capturing the single-index models we consider and lead to a tractable landscape. The tied weights assumption may also be motivated by the fact that untied neurons tend to align in a few directions (related to the target function) throughout training in some cases (see e.g., Figure 2 in Chizat and Bach (2018)). Nevertheless, we do not feel strongly about using the word “natural” to describe our setting, and would be happy to change terminology in a revised version.
>
> ***“Q2: Why the variance of the bias needs to be larger than one in the analysis (L94)? What would change in the analysis for T<1?”***
>
> We need tau > 1 in order to control the approximation error (more specifically in Lemma B.6 in the appendix). Concretely, we want the random biases to be more spread out than the data (projected onto the relevant direction), so that the ReLU neurons can approximate the bulk of the data well.
>
> ***“Q3: Why is lambda restricted to (0,1) in L147? Is this always satisfied by the optimal lambda in Corollary 5.2?”***
>
> We thank the reviewer for spotting this inconsistency. We were assuming lambda < 1 for convenience, but will change this to lambda < C (constant independent of n and d), which is true in the corollaries, and only changes our results by a constant.
>
> ***“Q4: What role does the choice of phi=ReLU play in the results? Would it be possible to derive similar guarantees for other activations?”***
>
> We thank the reviewer for this great question. It is indeed possible to derive results for other activations, as long as similar approximation guarantees are established for the corresponding kernel. That said, the ReLU activation function enables an integral representation based on Sobolev norms, as already observed and exploited in previous works [Bach, 2017, Savarese et al., 2019.; Ongie et al., 2020], which will generally be more interpretable than with arbitrary activations.
>
> ### References
> - Lenaic Chizat, Francis Bach, *On the Global Convergence of Gradient Descent for Over-parameterized Models using Optimal Transport*. 2018
> - Francis Bach. *Breaking the Curse of Dimensionality with Convex Neural Networks*. 2017
> - Pedro Savarese, Itay Evron, Daniel Soudry, Nathan Srebro. *How do infinite width bounded norm networks look in function space?*. 2019
> - Greg Ongie, Rebecca Willett, Daniel Soudry, and Nathan Srebro. *A function space view of bounded norm infinite width relu nets: The multivariate case*. 2020.

---

### Official Review · Reviewer_5fyA · 2022-07-19

**Rating:** 7
**Confidence:** 4
**Soundness:** 4 excellent
**Presentation:** 4 excellent
**Contribution:** 3 good

**Summary:**

The authors consider the problem of regression in high dimensions, with Gaussian i.i.d. inputs and a target function that is an arbitrary function that only depends on the projection of the input along a special direction, θ*. They study the performance of a two-layer neural network with many neurons, which however all share the same weight θ (the biases are different for each neuron, but they are random and fixed). The second-layer weights are free and optimised.

The authors first show that at small regularisation, the only first-order critical points of the population loss of the model correspond to a magnetisation of the model m =〈 θ, θ*〉= 0 at initialisation, and m=1 at perfect recovery. (Theorem 4.2). In other words, the optimization landscape in the population limit does not contain spurious minima or other obstacles.

They then study the generalisation of a model trained on the empirical loss. Considering an algorithm where they first train the first layer, then fine-tune the second layer while keeping the direction θ fixed, they obtain a bound on the excess risk (eq. 17) that has two parts: one that depends on the dimension, and which relates to the cost of escaping the saddle-point at initialisation, and a second part which comes from the non-parametric fine-tuning.

**Questions:**

n/a

**Strengths And Weaknesses:**

This is a clean and well-written analysis of feature learning dynamics. The authors propose an original model for the two-layer network that allows for a precise analysis, and I believe this model can be a nice addition to the set of models where these questions can be studied, in addition to the teacher-student setup. The authors do a good job imho of putting their work into the context on other recent results, such as recent work on "information exponents" Ref. [3]. All in all, I believe this paper will be of interest and use to the theory community at NeurIPS and should therefore be published.

---

> ### Author Response · Authors · 2022-08-02
> **Author Response**
>
> We thank the reviewer for a very clear and concise summary of our results, and encouraging feedback.

---

### Meta-Review · Area_Chair_AqSa · 2022-08-25

**Recommendation:** Accept
**Confidence:** Certain

**Metareview:**

In this paper, the authors study the problem of learning a single index model using a specific class of two-layer neural networks. Interestingly, at variance with previous works on single-index models, the target single-index model is not matched with the network architecture and Both layers are trained. One of the key results is that gradient flow can achieve a good approximation error in this setting.

The consensus is that this theoretical paper constitutes a nice, and well-written analysis of feature learning dynamics, a topic of current interest in statistical learning theories. It definitely adds to the current literature on the separation between neural networks and lazy regime (NTK, etc...),  and on the benefits of over parametrization for optimization in non-convex landscapes (benign overfitting, etc...). All referees were clearly supporting acceptance.

**Award:**

No

---

### Decision · Program_Chairs · 2022-09-14

Accept